



**Morphology, mixing state, and hygroscopicity of primary biological**
**aerosol particles from a Chinese boreal forest**
Weijun Li[1], Lei Liu[1], Qi Yuan[1], Liang Xu[1], Yanhong Zhu[1], Bingbing Wang[2], Hua Yu[3], Xiaokun
Ding[4], Jian Zhang[1], Dao Huang[1], Dantong Liu[1], Wei Hu[5], Daizhou Zhang[6], Pingqing Fu[5],
Maosheng Yao[7], Min Hu[7], Xiaoye Zhang[8], Zongbo Shi[9,5]
[1]Department of Atmospheric Sciences, School of Earth Sciences, Zhejiang University, Hangzhou
310027, China
[2]State Key Laboratory of Marine Environmental Science, College of Ocean and Earth Sciences,
Xiamen University, Xiamen 361102, China.
[3]College of Life and Environmental Sciences, Hangzhou Normal University, 310036, Hangzhou, China
[4]Department of Chemistry, Zhejiang University, Hangzhou 310027, China
[5]Institute of Surface-Earth System Science, Tianjin University, 300072, Tianjin, China
[6]Faculty of Environmental and Symbiotic Sciences, Prefectural University of Kumamoto,
Kumamoto 862-8502, Japan
[7]State Key Joint Laboratory of Environmental Simulation and Pollution Control, College of
Environmental Sciences and Engineering, Peking University, Beijing 100871, China
[8]Key Laboratory of Atmospheric Chemistry, Chinese Academy of Meteorological Sciences, Beijing,
China
[9]School of Geography, Earth and Environmental Sciences, University of Birmingham,
Birmingham B15 2TT, UK
*Correspondence to*: Weijun Li (liweijun@zju.edu.cn)



**Abstract:**
Biological aerosols play an important role in atmospheric chemistry, clouds, climate, and public
health. Here, we studied the morphology and composition of primary biological aerosol particles
(PBAPs) collected in the Lesser Khingan Mountain boreal forest of China in summertime using
transmission electron microscopy and scanning electron microscopy. Of all detected particles >
100 nm in diameter, 13% by number were identified as PBAPs. In addition, 57% of the PBAPs
were identified as bacteria, followed by brochosomes (24%) and fungi (19%). The dominant size
of bacteria was 1-4 μm, fungi was 2-4 μm, and brochosomes was 300-500 nm. The number size
distribution of PBAPs coupled with the mass concentrations of $PM_{2.5}$ and $PM_{10}$ were used to
estimate the total mass concentration of PBAPs, which is approximately 1.9 μg m$^{-3}$ and accounts
for 47% of the in situ $PM_{2.5-10}$ mass. C, N, O, P, K, and Si are detected in all PBAP particles, and P
represented a major marker to identify PBAPs. Moreover, there is a higher frequency and
concentration of PBAPs at night compared with day. Bacterial and fungal particles displayed weak
hygroscopicity with a growth factor of ~1.09 at RH=94%. Electron microscopy shows that
approximately 20% of the bacterial particles were internally mixed with metal, mineral dust, and
inorganic salts in the boreal forest air. This work provides a database for both further
understanding physicochemical state of individual PBAP particles from natural sources and
expanding the scope of atmospheric implications.




**Key points**

- In a boreal forest, 57% of the PBAPs were identified as bacteria, 19% as fungi and 24% as brochosomal particles.

- Emissions of PBAPs tend to occur with high humidity at night rather than during the day.

- Hygroscopic experiments show that most of the primary PBAPs displayed weak hygroscopicity, and their growth factor was ~1.09 at RH=94%.



## 1. Introduction


Primary biological aerosol particles (PBAPs) (e.g., bacteria, spores, fungi, viruses, algae, and
pollen) are ubiquitous in the Earth's atmosphere and represent key elements in the life cycle of
many organisms and ecosystems (Poschl, 2005;Tunved et al., 2006). PBAPs are airborne
biological materials that prevail from the biosphere to the atmosphere (Huffman et al., 2010), and
they can account for a large proportion of the aerosol particle mass in pristine forest air as well as
in some rural and ocean environments (Elbert et al., 2007;Bauer et al., 2008;Hu et al., 2017;May
et al., 2018). Research interest in biological aerosol has been growing significantly in recent
decades (Després et al., 2012). Laboratory studies have shown that certain cell fragments in
biological aerosols may be active as both cloud condensation nuclei (CCN) and ice nuclei (IN)
(Morris et al., 2004;Ling et al., 2018). A recent study demonstrated that fungal spores emitted by
the forest contain abundant sodium salt particles in the central Amazon basin and significantly
influence the hygroscopicity and CCN of PBAPs (China et al., 2018). Furthermore, field
campaigns have found that abundant biological aerosols occur in cloud ice-crystals, fog/cloud,
rain, and snowfall (Amato et al., 2005;Möhler et al., 2007;Christner et al., 2008;Pratt et al.,
2009;Twohy et al., 2016;Hu et al., 2018). These studies addressed the hypothesis that PBAPs
indeed influence the hydrological cycle and climate by initiating the formation of clouds and
precipitation as CCN and IN. PBAPs in pristine regions significantly contribute to the particle
mass and number and have important implications for radiation budget estimates in the
atmosphere (Tunved et al., 2006;Martin et al., 2010;Tobo et al., 2013). Although PBAPs only have
a small contribution to particulate mass in polluted urban air, pollen and spores from plants can
induce human allergic symptoms worldwide (Denning et al., 2006;Zhou et al., 2019).



Previous studies have investigated the sampling, particle number concentration, shape, and
chemical characterization of primary biological aerosols (Wittmaack et al., 2005;Elbert et al.,
2007;Fröhlich-Nowoisky et al., 2009;Huffman et al., 2010;Després et al., 2012;Hu et al.,
2017;Therkorn et al., 2017;Zhang et al., 2017;Chen and Yao, 2018). For example, the contribution
of fungal spores to total organic carbon was estimated to be approximately 10% in clean and
polluted periods in Beijing (Yue et al., 2017) and 0.9% (up to 9.9% in coarse size) in the Austrian
Alps (Bauer et al., 2002). Elbert et al. (2007) reported that the mean mass concentration of PBAPs
was ~1 µg m$^{-3}$ and accounted for 20% of total coarse particle mass in central Europe. To obtain the
chemical composition of PBAPs, many studies tend to detect biochemical markers (e.g., proteins,
fatty acids, sugars) and nucleic acids (i.e., DNA and RNA) to determine their properties in the
atmosphere (Georgakopoulos et al., 2009;Chen and Yao, 2018;Hu et al., 2018;Ling et al., 2018).
These comprehensive and detailed studies of time- and size-resolved PBAPs and their biochemical
markers do not well explain the physical properties (e.g., morphology, phase, hygroscopicity, and
mixing state) of individual PBAPs in the atmosphere
A limited number of studies have provided detailed morphological and mixing state data on
PBAPs (Posfai et al., 2003;Wittmaack, 2005;Wittmaack et al., 2005;China et al., 2018).
Information on the morphology, size, and mixing state of different PBAPs allow for the
identification of biological particle types and provide insights into the actual state of individual
biological particles suspended in the atmosphere (Posfai et al., 2003;Wittmaack et al., 2005;Martin
et al., 2010;Després et al., 2012). Single particle analyses can characterize the physical and
chemical properties of individual particles from the nanoscale to microscale (Li et al., 2016), and
this approach can also indicate the optical and hygroscopic properties and possible sources of





these particles. Thus far, only a few studies have observed the morphology and size of some
biological aerosols via scanning electron microscopy (SEM) (Shi et al., 2003;Wittmaack et al.,
2005;Shi et al., 2009;Martin et al., 2010;Huffman et al., 2012;Valsan et al., 2015;Wu et al., 2019).
For example, fungal fragments sampled from Amazonia contain hygroscopic sodium salts based
on an environmental scanning electron microscopy (ESEM) analysis, and these salts significantly
influence the hygroscopic growth and light scattering of the fragments (China et al., 2016;China et
al., 2018). However, whether fungal spores emitted by boreal forests are similar to the fungal
spores in central Amazon forests, which contain sodium salts, has not been resolved. Therefore,
the morphology, elemental composition, and mixing state of individual PBAPs (nanometer to
micrometer size) collected from other global forests must be analyzed.

Forests are important contributors of primary biological aerosols in the atmosphere (Tunved et

al., 2006;Spracklen et al., 2008;Després et al., 2012). Aerosols in large forests contain abundant
biological particles from plants and lesser anthropogenic pollutants of long-range transport
(Tunved et al., 2006;Gabey et al., 2010;Martin et al., 2010). We chose the Lesser Khingan
Mountains in Northeast China, which is the second largest boreal forest in China. In this study,
integrated single-particle techniques are required to clearly observe individual PBAPs from the
nanoscale to microscale and further reveal their hygroscopicity in the atmosphere. Transmission
electron microscopy (TEM) and scanning electron microscopy (SEM) both have been employed to
characterize the morphology, size, and mixing state of various PBAPs collected over the boreal
forest. Furthermore, hygroscopic experiments on the primary biological particles have been
conducted.



**2.  Methods**
**2.1  Sampling site and sample collection**
The sampling site is at the Heilongjiang Liangshui National Nature Reserve (47.32$^{\circ}$ N,
128.54$^{\circ}$ E) in the center of the Lesser Khingan Mountains of Northeast China (Figure 1). The
boreal region is characterized by large seasonal variations in temperature, and the flora is
dominated by Korean pine and spruce species. There are no anthropogenic sources of pollutants,
such as villages, industries and vehicles within 80 km of the sampling site. Because boreal forests
play a key role in biological aerosol emissions during summer, we collected aerosol samples in
August.
Aerosol particles were collected on copper (Cu) grids coated with a carbon (C) film (carbon
type-B, 300-mesh copper, Tianld Co., China) and silicon wafer by a DKL-2 sampler (Jenstar
Electronic Technology, China) with a single-stage cascade impactor (Li and Shao, 2009) equipped
with a 0.5 mm diameter jet nozzle at a flow rate of 1.0 L/min at 9:00, 15:00, 21:00, and 2:00 a.m.
(midnight) local time every day. The sampling duration at each time varied from 10 min to 25 min
depending on the particle distribution on the substrate. After sample collection, we immediately
performed optical microscopy at 100 magnification to determine whether the aerosol distribution
on the substrate was suitable for electron microscopy analysis. The sampling procedure can
guarantee that the collected particles separated or did not overlap each other on the substrate (Li et
al., 2016). The collection efficiency of the impactor is 50% for particles with an aerodynamic
diameter of 0.1 μm when we assume an aerosol particle density of 2 g cm$^{-3}$. The Cu grids and
silicon wafers placed in a dry, clean, and airtight container were stored in a desiccator at 25 °C and
20±3% RH to minimize exposure to ambient air and preserve them for analysis.
The daily $PM_{2.5}$ and $PM_{10}$ samples were collected on quartz-fiber filters with a diameter at 90
mm through two medium-volume samplers (TH-150, Wuhan Tianhong, China) at a constant flow
rate of 100 L $min^{-1}$. The samples were changed at 08:00 a.m. each day. Our sampling and
monitoring instruments in the field experiment were installed on a building roof 15 m above
ground. The quartz filters were put in polyethylene boxes immediately after sampling and stored
at −5 °C. The filters were equilibrated at a constant temperature (20 ± 0.5 °C) and humidity (50 ±
2%) for over 24 h before being weighed with an electronic microbalance (Sartorius-ME5,
Germany). Meteorological data, including the relative humidity (RH), temperature, wind speed,
and wind direction, were measured and recorded every 5 min by an automated weather meter
(Kestrel 5500, USA). During the sampling period, the relative humidity (RH) and temperature
varied from 40-70% and 22-28 ℃ during the day and 90-100% and 10-15 ℃ during the night,
respectively. The wind speed was 1.5-7.6 m $s^{-1}$ during the day and 0-1 m $s^{-1}$ at night (Figure S1).
**2.2 Transmission electron microscopy analysis**
Individual aerosol particles collected on Cu grids were analyzed via transmission electron
microscopy (TEM, JEM-2100, JEOL Ltd., Japan) at a 200 kV accelerating voltage. The TEM
system is equipped with an energy-dispersive X-ray spectrometer (EDS, INCA X-Max$^{N}$ 80T,
Oxford Instruments, UK). EDS semiquantitatively detects the elemental composition of individual
particles with an atomic number greater than six ($Z > 6$). However, Cu peaks in the EDS spectra
were not considered because of interference from the copper substrate of TEM grids. We
determined the morphology, composition, and mixing state of individual particles through the
combination of TEM and EDS. To reduce the damage to particles under the electron beam, the
EDS collection duration was limited to 15 s. Particles in 3-5 grids of each sample were analyzed to



ensure their universality and representativeness. TEM can determine the internal mixing structure
of different aerosol components in fine particles and their specific composition.
**2.3 Scanning electron microscopy analysis**
Scanning electron microscopy (SEM) is performed using a type of electron microscope that
can determine the particle surface by scanning it with a high-energy beam of electrons in a raster
scan pattern. An SEM system (Zeiss Ultra 55) equipped with a field emission gun operating at 5–
20 kV was used to obtain detailed information on the surfaces of individual aerosol particles.
Moreover, the SEM was equipped with an energy-dispersive X-ray spectrometry (EDS), which
can analyze the chemical composition of individual particles. The SEM can efficiently obtain the
surface morphology, size, and composition of coarse particles without any coating process on the
substrate.
**2.4 Hygroscopic experiments**
A custom-made individual particle hygroscopic (IPH) system was used to observe the
hygroscopic properties of individual biological particles at different relative humidity (RH)
values. The IPH system involved three steps: (1) introducing $N_2$ gas with a mass flow
controller into a chamber; (2) setting a TEM grid or silicon wafer on the bottom of an
environmental microscopic cell (Gen-RH Mcell, UK), which can change the RH and maintain
the temperature at 20 °C; and (3) taking images at incremental RH values using an optical
microscope (Olympus BX51M, Japan) with a camera (Canon 650D). This IPH system has
successfully captured the hygroscopic growth of individual particles collected on either a
silicon wafer or TEM grid (Sun et al., 2018). In this study, one typical sample containing
biological particles was chosen to observe the hygroscopic growth of the bacterial and fungal





particles at RH values ranging from 5% to 94%. The particle growth factor (GF) is an
important parameter used to describe the hygroscopic growth of individual particles, and it is
defined as follows:

$$GF(RH) = \frac{D(RH)}{D_0}$$

where $D(RH)$ and $D_0$ are the diameters of particles at a given RH and at 5% RH, respectively.

**3.   Results and Discussion**
**3.1  Morphology and elemental composition of PBAPs**

Among the 4422 analyzed aerosol particles with diameters of 100 nm-10 μm, individual

particles are classified into five groups based on their morphology and composition: S-OM
(mixture of sulfate (S), organics (OM)), OM, mineral dust, and PBAPs (Figure 2). S can be used
to indicate secondary sulfates; abundant C and minor O with transparent color constitute the
coating of the sulfate core and represent secondary organic matter; and irregular particles
containing Si, Al, Ca and minor Fe, Ti normally indicate mineral dust particles. Moreover,
previous studies have stated that elemental P in individual particles and the associated unique
morphologies can be used to identify PBAPs via electron microscopy (Poschl, 2005;Wittmaack et
al., 2005). Thirteen percent of particles were PBAPs, and low magnification SEM and TEM
images both revealed that abundant PBAPs occurred in the samples (Figure 2a-d).

The number fractions of size-resolved aerosol particles show that secondary S-OM and OM

particles were the dominant particle groups in the fine mode (< 1 μm) while PBAPs and mineral
particles dominated the coarse mode (≥ 1 μm) (Figure 3a). Moreover, we noticed that the number
fractions of PBAPs in each sample collected at night were much higher than those collected



during the day. Abundant fine secondary sulfate and organic particles from photochemical
formation were observed during the day. Figure 3b shows that the average number fraction of
PBAPs was 2.5% in the samples collected during the day and as high as 30.0% at night. If we
further calculated the number concentration of PBAPs in Figure 3b, the PBAPs concentration
significantly increased by approximately 7 times from daytime to nighttime, although the
non-PBAPs concentration decreased.
The PBAPs were classified based on morphology into four types: bacterial, fungal,
brochosomal, and other biological particles. Pollen was not found in our samples, which may be
because large pollen emissions occur in spring and early summer instead of late summer (August).
Similarly, Wittmaack et al. (2005) did not find pollen in the boreal forest air in other locations in
late summer.
**Bacterial particles.** Figure 4 shows the typical morphology of the bacteria particles, which
have a rod-like shape and include several dark inclusions (Posfai et al., 2003). These bacteria
particles were stable under the electron beam during the TEM analysis, and they contained C, N,
O, P, and K with minor Mg, Si, S, Ca and Fe (Figure 4). EDS further showed that the bacterial
inclusions contained much higher P, Mg, and K while other parts contained much higher C, N, and
O (Figure 4). Bacterial inclusions resemble a nucleoid and plasmid and other parts of the
cytoplasm.
Figure 2b shows an SEM image of a bacterium particle and indicates its morphology
(although no information about the inner structure is obtained). The surface of the bacterial
particles is uneven and the surface contains clear wrinkles, which probably formed as the bacteria
dried on the substrate (Patterson et al., 2016). Most of the bacterial particles have a rod-like shape



(Figure 5), and some showed a near-spherical shape (Figure 5). A majority of the individual
bacteria particles is present as a single bacteria cell, although some form aggregates (Figure 2a, c).

**Fungal particles.** SEM images show that various fungal particles occurred in the boreal

forest air (Figure 2a). TEM observations show that the fungal particles generally display irregular
shapes and rough surfaces and that they mainly contain C, O and Si and following minor N, Mg, P,
S, K and Fe (Figure 6). Figure 6 shows that several typical fungal particles with diameters of
3.7-6.5 μm do not have well-defined shapes and their surfaces have regular strips or regular
bubble. Based on the classifications of fungal particles from Wittmaack et al. (2005), particles
shown in Figures 6a and 7a-d and in Figures 6b-c and 7e-f were considered as conidia and spores,
respectively.

We identified 19% of the primary biological particles as fungi in this study. Compared with

bacterial particles, fungal particles normally have a rougher surface (Figures 6-7) and contain
much higher Si and lower N. Moreover, a few fungal particles are found associated with fragments
of other unknown biological particles (Figures 7a, d, e).

**Brochosomal particles.** TEM observations show that brochosomal particles frequently

occurred in the samples and accounted for 24% of the analyzed primary biological particles.
Interestingly, the outline of each brochosome approximates a truncated icosahedron and the
brochosome particles likely have unique inner structures, such as C60 Buckminster fullerenes
(Figures 8a-b and 9). Compared with the bacterial and fungal particles, the chemical compositions
of the brochosomal particles show extremely high Si and low P in addition to major C and O and
minor N, Na, S, K and Fe.

**Other biological particles.** In this study, we observed only a few elongated particles among



the biological particles. TEM observations show that these particles mainly contain C, O, and Si.
It should be noted that P is not detectable in some of these biological particles as shown in Figures
10-11. Because of the low particle numbers, we could not statistically determine their size
distribution. The TEM and SEM images both show that these particles are quite large at 8-20 μm.
We speculate that these biological particles were fragments of plants or insects. For example,
Wittmaack et al. (2005) suggested that the spaghetti-type biological particles from Figure 10a-d
and Figure 11c are likely epicuticlar wax fragments of plants. The biological particles with
recognizable surface features from Figures 10e and 11a-b resemble part of insects. Because these
biological particles are large, the TEM and SEM analyses both easily identified them, although the
SEM analysis provided better and more detailed information of the large biological particles in the
samples.
**3.2  Relative abundance of PBAPs**
Bacterial particles range from 400 nm to 10 μm, with a peak diameter at 1-4 μm (Figure 12a).
Of the total analyzed primary biological particles, 57% were identified as bacterial particles
(Figure 12a). Most fungal particles occurred in the coarse mode and their size distribution
dominated at 2-4 μm with one peak at 3 μm. Brochosome particles dominate at 300-500 nm and
have one main peak at 300 nm. SEM observations show that brochosomal particle clusters were
distributed on the substrate (Figure 9a). This is because certain hygrophobic secretions of insects
(e.g., leafhoppers) are composed of brochosomal particles, and these secretions function in
keeping the insect cuticle dry (Wittmaack, 2005;Rakitov and Gorb, 2013).
Figure 12b shows the daily mass concentrations of $PM_{2.5}$ at ~6 μg m$^{-3}$ and $PM_{10}$ at ~10 μg m$^{-3}$
and the ratio of $PM_{2.5}/PM_{10}$ at ~0.6 at the sampling site. The results from the electron microscopy


analysis further estimated that PBAPs, mineral dust, and the remaining particles accounted for
50%, 25%, and 25% of the coarse mode, respectively (Figure 3a). Assuming a density of ~1 g
$cm^{-3}$ for PBAPs (Elbert et al., 2007), 2 g $cm^{-3}$ for mineral dust particles, and 1.4 g $cm^{-3}$ for the
remaining particles (e.g., S-OM, OM, and metal) (Rissler et al., 2006), mass concentrations of
three different types of particles with different size bins can be further calculated based on particle
number and geometrical diameter as shown in Figure 3a. Finally, we can estimate that the mass
concentration of PBAPs, mineral dust, and remaining particles accounted for 47%, 43%, and 10%
of $PM_{2.5-10}$, respectively. The results suggest that large boreal forests are significant sources of
PBAPs in summertime in Northeast China.

Thirteen percent of all detected particles collected from the boreal forest air are PBAPs. Such

a high fraction of PBAPs has not been reported in urban and rural air in China (Shi et al., 2003;Shi
et al., 2009;Li et al., 2016). The number concentration of PBAPs is higher at night than during the
day (Figure 3b). A shallower nocturnal boundary layer will lead to an increase in the number
concentration of coarse particles near the ground (Graham et al., 2003), although this increase
cannot explain the very large difference in the relative number fraction of PBAPs (12 times larger
at night than during the day) (Figure 3b). Therefore, this difference can only be explained by the
higher relative emission strength of PBAPs compared with non-PBAPs or the differential removal
of non-PBAPs. However, the latter is unlikely considering the usually larger size of non-PBAPs.

We compared the meteorological parameters (e.g., RH, temperature, and wind) and further

found that the high RH near 100% at night (Figure S1) could enhance the emissions of PBAPs.
This result is consistent with the conclusion of Elbert et al. (2007), who showed that PBAPs in a
boreal forest are generally most abundant in samples collected at night when the RH is close to


100%. Moreover, Troutt and Levetin (2001) found that the increase in PBAP concentration was
caused by the increase in basidiospores concentrations with RH, and they showed that a clear
diurnal rhythm occurs and peaks at 04:00-06:00 LT. Furthermore, the number ratio (4.6 at
nighttime and 4.0 at daytime) of bacterial vs fungal particles and their number concentrations
increased from daytime to nighttime (Figure 2S). These results might suggest that higher RH can
promote the emission of bacterial and fungal particles in boreal forests.

### 299    3.3  Mixing state of bacterial particles

Our study shows that bacterial particles are the most abundant PBAPs in the boreal forest air.

Figure 6 shows that the bacterial particles frequently occurred in fine and coarse modes. Although
approximately 80% of bacteria particles were externally mixed particles in the boreal forest air, we
still found that 20% of bacterial particles were internally mixed particles. TEM observations show
that bacterial particles were frequently internally mixed with mineral, metal, organics, and
inorganic salts. We noticed that irregular mineral dust particles significantly changed the shape of
the bacterial particles (Figure 13a-c). The EDS analysis shows that the internally mixed mineral
particles contain certain amounts of C, O, and P in addition to Si, Al, or Ca (Figure 13a-c),
suggesting that bacterial particles were coated with mineral dust particles. Patterson et al. (2016)
used cryo-TEM to observe soft bacterial structures in the atmosphere, and these irregular solid
mineral dust particles can transform the shape of the bacterial particles during their physical
coagulation processes.

In this study, we found that some nanoscale metal particles were internally mixed with

bacterial particles. Figure 13d-f further shows that these metals were spherical and contained Mn,



Si and/or Fe. As in previous studies, similar nanosize metal particles were emitted from industrial
emissions or power plants instead of natural soil (Li et al., 2017). TEM observations show that
these metallic particles were mainly attached to the surface of bacterial particles. Moreover, some
bacterial particles were coated by inorganic salts (e.g., K-P in Figure 13g and S-rich in Figure 13i)
and organics. The shape of the bacterial particles might change following the aging process during
long-range transport (Figure 13), although the elemental P or its associated ionic components
($H_2PO_4^-$ and $PO_3^-$) did not change (Pratt et al., 2009). Pratt et al. (2009) detected $H_2PO_4^-$ and $PO_3^-$
in individual cloud ice-crystal residues to identify PBAPs using aerosol time-of-flight mass
spectrometry. Although one study indicates that a few mineral dust or fly ash particles contain
trace inorganic P, these particles do not contain abundant organics and their number is low in the
air (Zawadowicz et al., 2017).

**3.4 Hygroscopicity of PBAPs**

In this study, we conducted a hygroscopic experiment to observe the hygroscopic growth of

fresh PBAPs. Before the hygroscopic experiment, an SEM analysis of the sample was performed,
and it showed that bacterial and fungal spores are dominant (Figure 2a). In the hygroscopic
experiment, primary bacterial and fungal spores all take up water and grow by up to 88% during
hydration, and they lose water and return to the dry particle size (reduction of 83%) during
dehydration (Figure 14). The growth factor of the bacterial and fungal spores is ~1.09 at 94%
based on the particle diameter change (Figure 14). These results show that fresh PBAPs have
extremely weak hygroscopicity.

Recent studies found that fungal fragments collected in Amazon forests displayed strong



hygroscopic properties (China et al., 2016;China et al., 2018) and were internally mixed with
certain amounts of sodium salts. However, we found weak hygroscopic growth at 1.09, whereas
this value was in the range of 1.05-1.3 for bacterial and fungal spores in previous studies
(Reponen et al., 1996;Lee et al., 2002). However, the result is much lower than the value of 2.31 at
RH 96% for sodium salt (China et al., 2016) and 1.60 at RH 94% for ammonium sulfate (Sun et
al., 2018). The comparison suggests that fresh PBAPs display extremely weak hygroscopicity and
do not contain any sodium salt in the boreal forest (Figure 2a). We integrated the morphological,
chemical composition and the low growth factor data of individual PBAPs and further concluded
that certain hydrophilic organic species might enhance the PBAP size at higher RH. Overall, our
results indicate that PBAPs from the substantial biological emissions from the Khingan Mountain
boreal forest are weakly hygroscopic in nature.

## 4. Atmospheric implications and conclusions

The TEM and SEM observations both showed that the morphology of PBAPs were unique
and different from that of the sulfate, mineral, soot, organics, and metal particles in continental air.
As a result, P derived from the particle EDS analysis coupled with the morphological features can
be used to identify the PBAPs. In this study, we establish one full database that includes the
morphology and composition of bacteria, fungi, and brochosomes, and it can be used to identify
primary biological particles using single particle techniques. We estimated that the mass
concentration of PBAPs, mineral dust, and remaining particles accounted for 47%, 43%, and 10%
of the $PM_{2.5-10}$ mass concentration, respectively, indicating that large boreal forests might
represent a major source of PBAPs in the atmosphere. The hygroscopic experiment shows that the



primary bacterial and fungal particles all take up water and grow by up at 88% during hydration,
and the particles lose water and return to the dry particle size (reduction of 83%) during
dehydration. The growth factor of the bacterial and fungal spores is ~1.09 at 94%, suggesting that
some hydrophilic organic species might enhance the size of PBAPs at higher RH.
PBAPs from the natural source may have an important role in precipitation and cloud
dynamics in the background areas (Prenni et al., 2009;Huffman et al., 2013). Field observations at
downwind areas of the Asian continent found substantial bacteria in dust plumes (Hara and Zhang,
2012). The mechanisms by which PBAPs influence mineral dust particles if they become
internally mixed particles, as shown in Figure 13a-c, remain unclear. Our results indicate that
significant amounts of PBAPs are emitted from the Khingan Mountain area acting as the "green
ocean" (Poschl et al., 2010) in Northeast Asia, and they may have an important impact on clouds
and climate in Northeast China and in the downwind North Pacific Ocean. Therefore, the
modelling work required to simulate how a large number of submicron primary biological
particles from boreal forests promote the atmospheric biogeochemical cycle and have a significant
impact on climate by acting as CCN and IN over the large boreal forest and the downwind areas.



**Author Contributions:** WL designed the study. WL, LL, QY, LX, and HY collected
aerosol particles. WL, LL, LX, YZ, BW, XD, and JZ contributed laboratory
experiments and data analysis. WL prepared the manuscript with contributions from
all the coauthors. BW, DH, DL, WH, DZ, PF, MY, MH, XZ, and ZS commented and
edited the paper.

**Competing interests:** The authors declare no competing financial interests

**Acknowledgments**
We appreciate Peter Hyde's comments and proofreading. This work was funded by the
National Natural Science Foundation of China (9184430003, 41622504 and 41575116),
Zhejiang Provincial Natural Science Foundation of China (LZ19D050001), and Zhejiang
University Education Foundation Global Partnership Fund. All the data are presented in the
paper.




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



*Figure Captions*

**Figure 1** Location of the sampling site in a boreal forest of the Lesser Khingan Mountain in Northeast China. The map source is Google Earth.

**Figure 2** Low magnification SEM and TEM images of individual particles collected from the forest air. (a) low magnification SEM image of bacterial (red arrow) and fungal particles (green color); (b) SEM image of a single bacterial particle; (c) low magnification TEM image of bacteria aggregations and single bacterial particles; (d) low magnification TEM image of single bacterial particles and secondary sulfate (S-rich) particles; (e) TEM image of mineral dust particle (f) TEM image of an organic matter (OM) particle; and (g) TEM image of S-OM coating internally mixed with a soot particle. The color in (a) (also in the following figures) was artificially painted on the original SEM images.

**Figure 3** Number fractions of different types of particles in different size bins and their total number fraction (a); and number fractions of primary biological aerosol particles (PBAPs) and non-PBAPs during the day and night (b). The number of analyzed particles is listed above each column.

**Figure 4** TEM image of one rod-like bacterial particle and EDS spectra of bacterial inclusions and other parts.

**Figure 5** TEM images showing different shapes of bacterial particles.

**Figure 6** TEM/EDS showing the morphology and composition of various fungal particles. (a) Rod-like fungi particle; (b) fungi particle with bubbles; (c) fungi particle with bubbles; and (d) EDS spectrum showing the composition of fungi particles.

**Figure 7** Color SEM images showing the shape, size, and surface properties of fungal particles. Size represents the diameter of fungal particles. (a-d) Surfaces of three rod-like fungal particles with a layer of strips. The green-colored particles are conidia, and the attached pink particles on the



conidia are fragments from other unknown biological particles. (e-f) Surfaces of two fungal particles

with bubbles. The green particles are fungi spores, and the attached red part on the spores is a fragment

from other unknown biological particles. The color is artificially modified through the original SEM.

**Figure 8** TEM images of brochosomes and the composition of (a) a single brochosome and

brochosome aggregations; (b) high-resolution TEM image showing the inner structure of one

brochosome; (c) EDS spectrum showing the chemical composition of the brochosomes.

**Figure 9** Color SEM images of brochosomes. (a) Single brochosome and their aggregations. Some

brochosomal particles are associated with primary biological species. (b) High-resolution SEM image

showing the surface properties of the brochosomal particles.

**Figure 10** TEM images showing the morphology of the primary biological particles. (a) One elongated

particle with thorns; (b) one circular particle; (c-d) two elongated particles; and (e) one spindle particle

**Figure 11** Color SEM image showing the morphology and surface properties of three elongated

biological particles.

**Figure 12** Size distribution of PBAPs and mass concentration of daily $PM_{2.5}$. (a) Number fraction

(right y-axis) and size distribution (left y-axis) of three types of primary biological particles. (b)

Daily mass concentrations of $PM_{2.5}$ and $PM_{10}$ and their ratio during the sampling period

**Figure 13** Internally mixed bacteria particles observed by TEM. (a-c) Internal mixture of mineral and

bacteria; (d-f) internal mixture of metal and bacteria; (g) internal mixture of inorganic salts and bacteria;

(h) internal mixture of organics and bacteria; and (i) internal mixture of S-rich salts and bacteria.

**Figure 14** Hygroscopic growth of the primary biological particles on the silicon wafer collected at

night. All the particles confirmed by SEM are bacteria and fungi. The up arrows (i.e., RH)

represent hydration, and the down arrows represent dehydration.



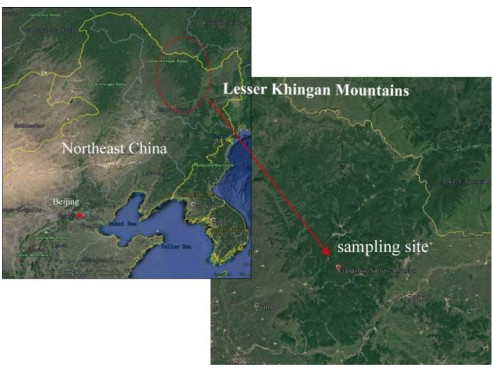

**Figure 1** Location of the sampling site in a boreal forest of the Lesser Khingan Mountain in Northeast China. The map source is Google Earth.

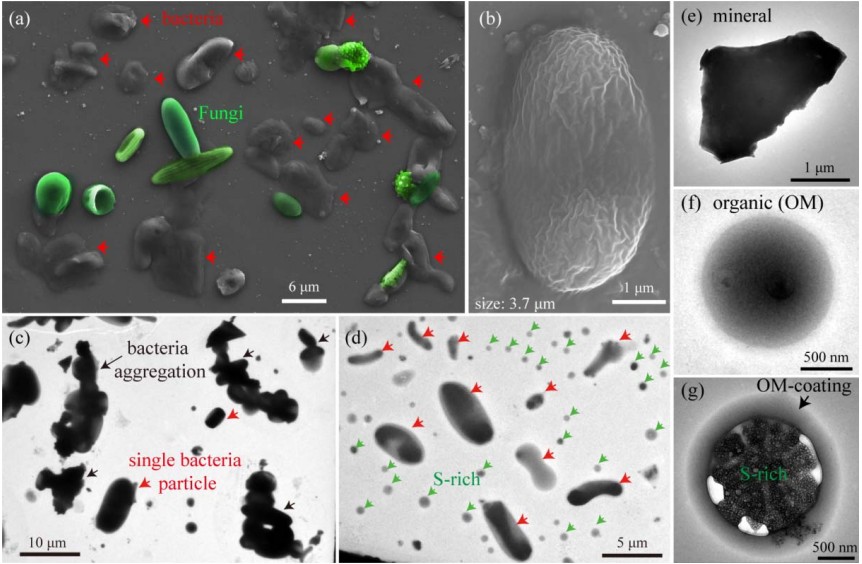

**Figure 2** Low magnification SEM and TEM images of individual particles collected from the forest air.

(a) low magnification SEM image of bacterial (red arrow) and fungal particles (green color); (b) SEM

image of a single bacterial particle; (c) low magnification TEM image of bacteria aggregations and

single bacterial particles; (d) low magnification TEM image of single bacterial particles and secondary

sulfate (S-rich) particles; (e) TEM image of mineral dust particle (f) TEM image of an organic matter


(OM) particle; and (g) TEM image of S-OM coating internally mixed with a soot particle. The color in

(a) (also in the following figures) was artificially painted on the original SEM images.

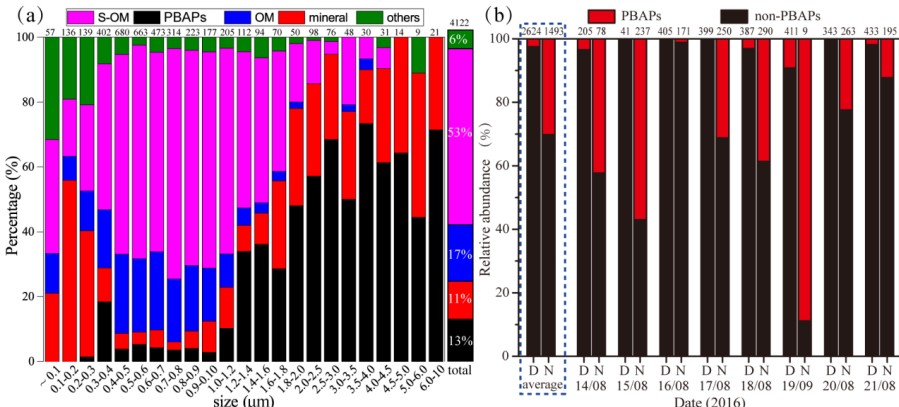

**Figure 3** Number fractions of different types of particles in different size bins and their total number

fraction (a); and number fractions of primary biological aerosol particles (PBAPs) and non-PBAPs

during the day and night (b). The number of analyzed particles is listed above each column.

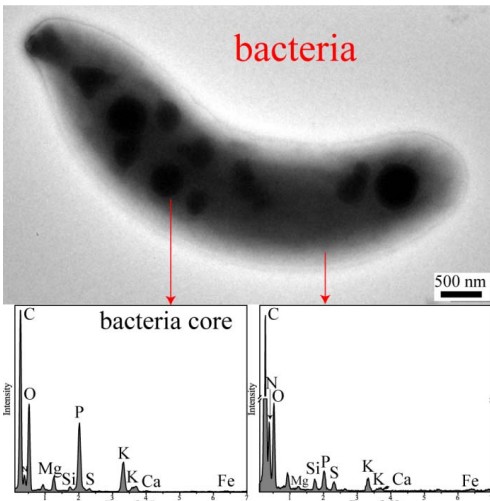

**Figure 4** TEM image of one rod-like bacterial particle and EDS spectra of bacterial inclusions and

other parts.





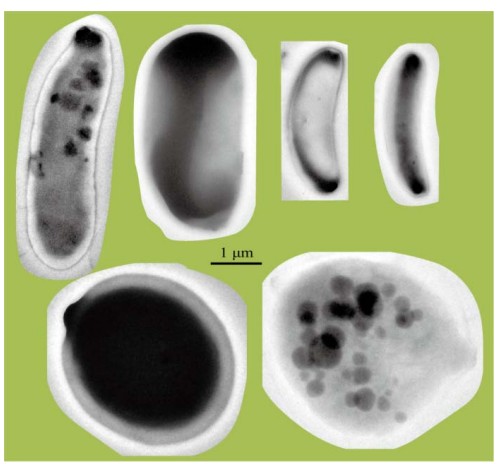

**Figure 5** TEM images showing different shapes of bacterial particles.

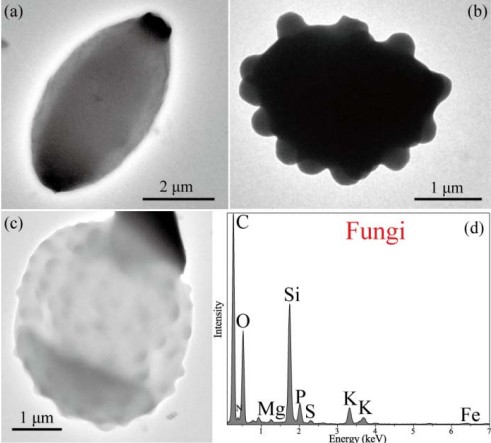

**Figure 6** TEM/EDS showing the morphology and composition of various fungal particles. (a)

Rod-like fungi particle; (b) fungi particle with bubbles; (c) fungi particle with bubbles; and (d)

EDS spectrum showing the composition of fungi particles.

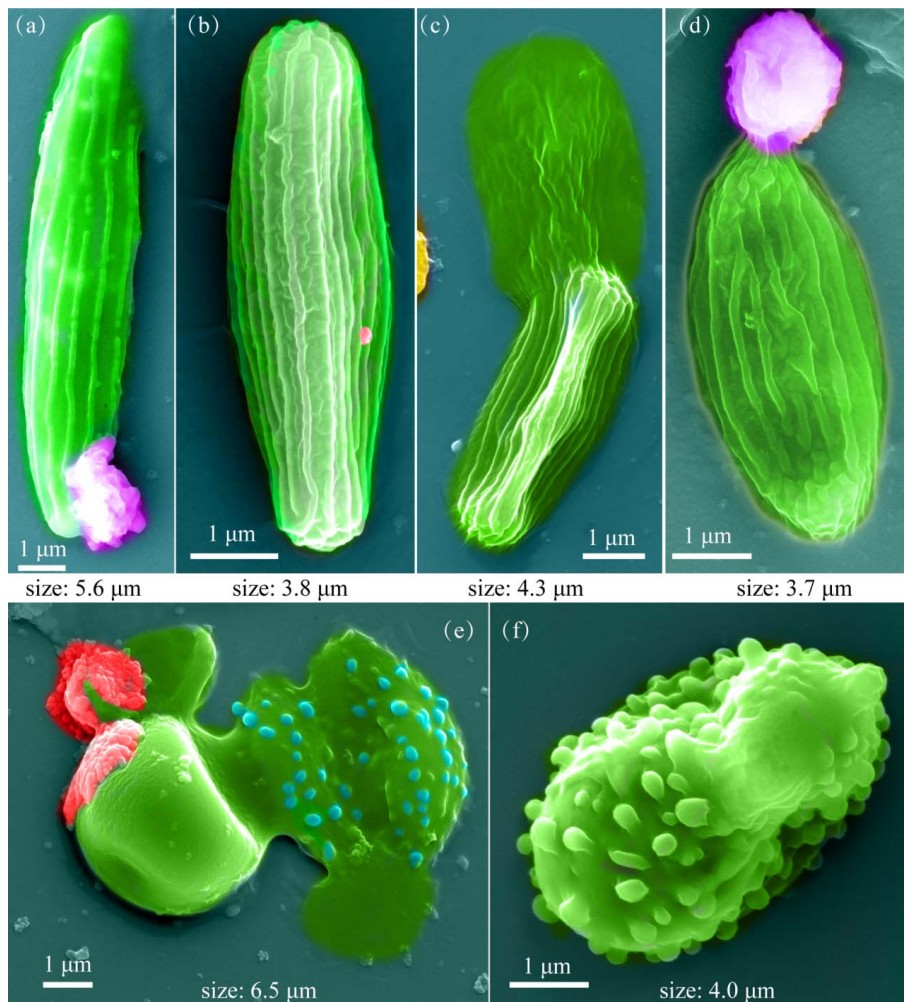

**Figure 7** Color SEM images showing the shape, size, and surface properties of fungal particles.

Size represents the diameter of fungal particles. (a-d) Surfaces of three rod-like fungal particles

with a layer of strips. The green-colored particles are conidia, and the attached pink particles on the

conidia are fragments from other unknown biological particles. (e-f) Surfaces of two fungal particles

with bubbles. The green particles are fungi spores, and the attached red part on the spores is a fragment

from other unknown biological particles. The color is artificially modified through the original SEM.





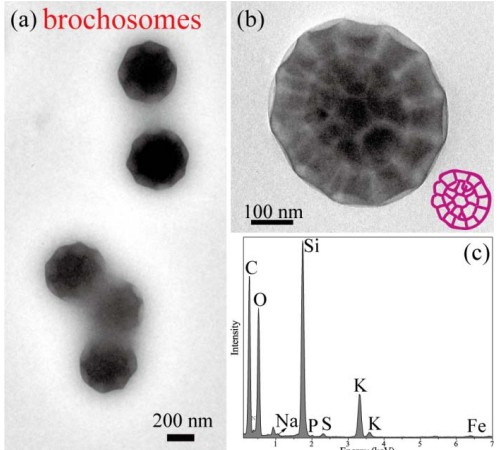

**Figure 8** TEM images of brochosomes and the composition of (a) a single brochosome and

brochosome aggregations; (b) high-resolution TEM image showing the inner structure of one

brochosome; (c) EDS spectrum showing the chemical composition of the brochosomes.

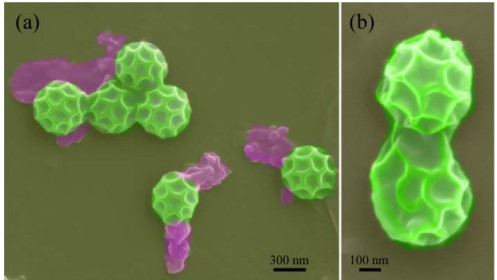

**Figure 9** Color SEM images of brochosomes. (a) Single brochosome and their aggregations. Some

brochosomal particles are associated with primary biological species. (b) High-resolution SEM image

showing the surface properties of the brochosomal particles.



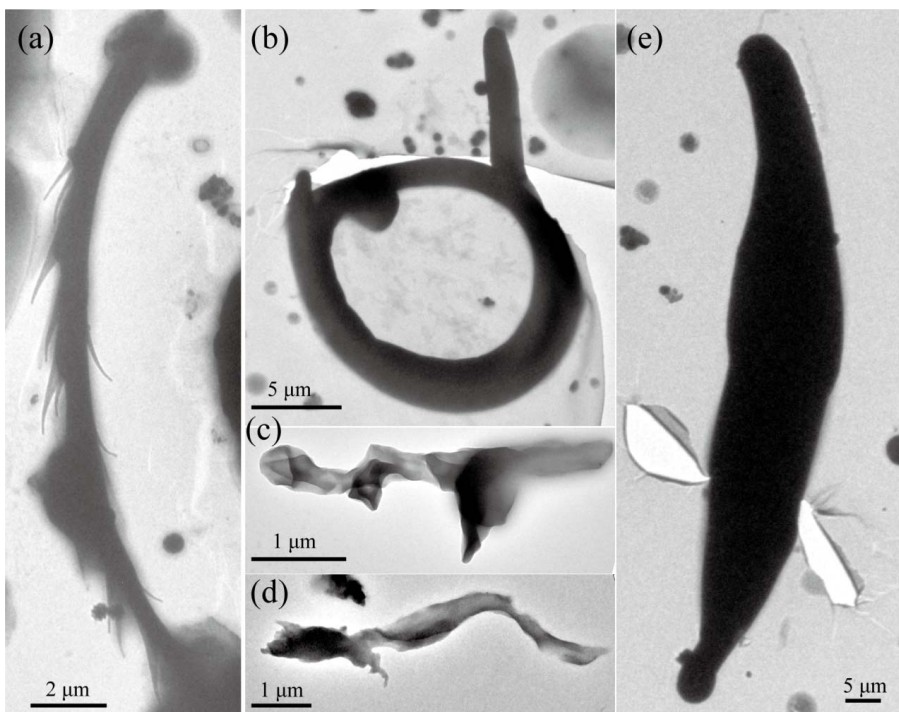

**Figure 10** TEM images showing the morphology of the primary biological particles. (a) One elongated

particle with thorns; (b) one circular particle; (c-d) two elongated particles; and (e) one spindle particle

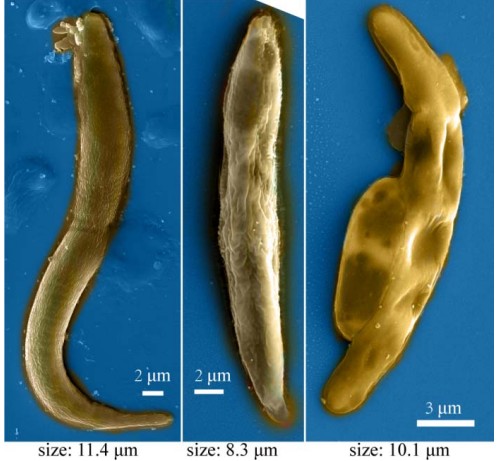

**Figure 11** Color SEM image showing the morphology and surface properties of three elongated

biological particles.

1



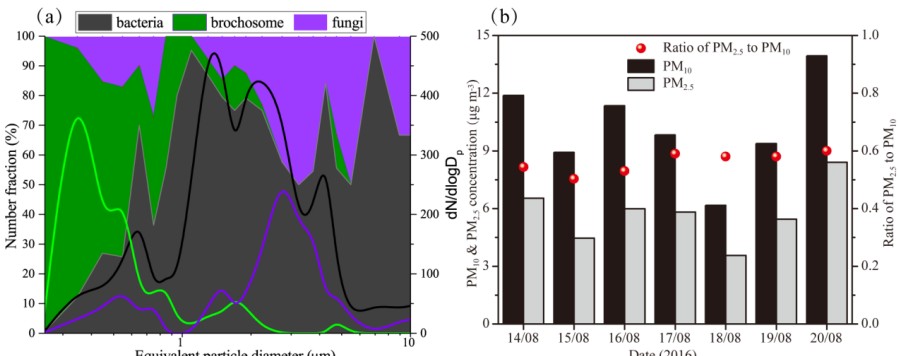

**Figure 12** Size distribution of PBAPs and mass concentration of daily PM$_{2.5}$. (a) Number fraction

(right y-axis) and size distribution (left y-axis) of three types of primary biological particles. (b)

Daily mass concentrations of PM$_{2.5}$ and PM$_{10}$ and their ratio during the sampling period

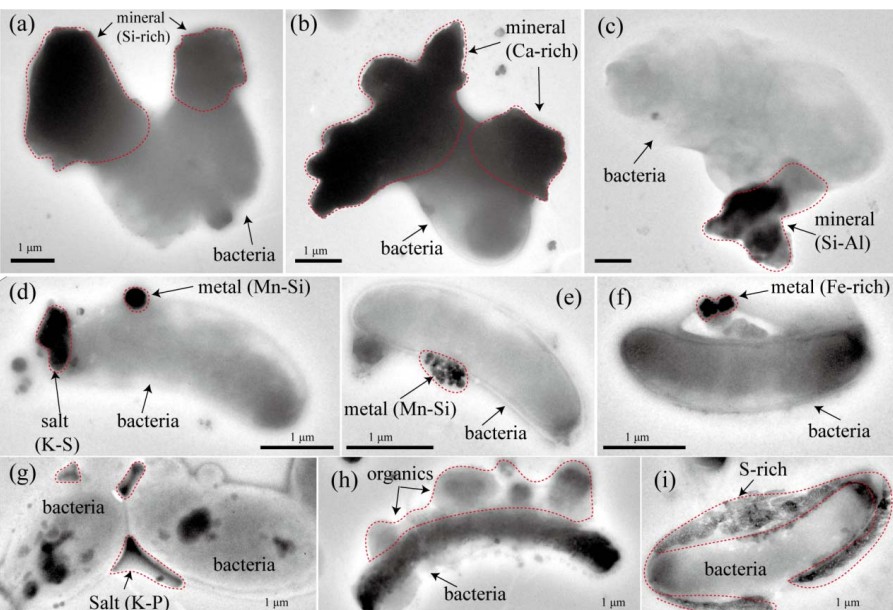

**Figure 13** Internally mixed bacteria particles observed by TEM. (a-c) Internal mixture of mineral and

bacteria; (d-f) internal mixture of metal and bacteria; (g) internal mixture of inorganic salts and bacteria;

(h) internal mixture of organics and bacteria; and (i) internal mixture of S-rich salts and bacteria.





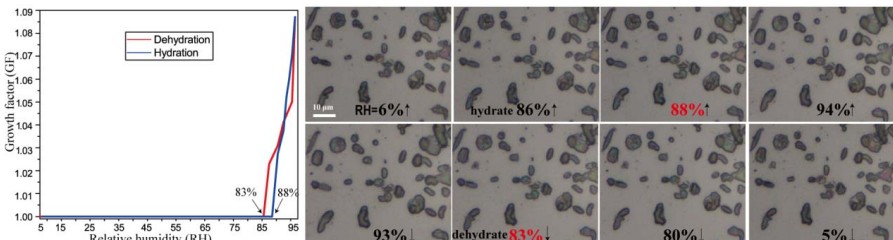

**Figure 14** Hygroscopic growth of the primary biological particles on the silicon wafer collected at

night. All the particles confirmed by SEM are bacteria and fungi. The up arrows (i.e., RH)

represent hydration, and the down arrows represent dehydration.