# Peer review of "Morphology, mixing state, and hygroscopicity of primary biological"

_Atmospheric Chemistry and Physics, 2019_

## Referee Comment (RC1) · Anonymous Referee #1 · 8 Aug 2019

This paper examines the composition of primary biological aerosol particles (PBAP) collected on various substrates for offline analysis, at a mountainous boreal forest site in China. Particles were classified optically based on morphology and composition was determined using a combination of TEM & EDS.

The authors report that PBAP were found to contain key, unique compositional markers (e.g., elemental P), which is consistent with previous studies performing similar analysis. A key result of this study was demonstrating that 20% of bacterial particles were internally mixed with non-PBAP, which may have a significant impact on the long-range transport of bacteria and aerosol budgets as well as mixed-phase aerosol-cloud

interactions. The authors also examined PBAP hygroscopicity, demonstrating that the sampled PBAP display small growth factors and subsequently weak hygroscopicity.

Overall the paper is reasonably well written and provides useful information to be absorbed into our general understanding of PBAP emissions and quantifying the fraction of PBAP which are internally mixed is a key result. My only significant criticism is that the paper lacks detail on the sample/substrate handling procedure employed, what procedures were in place to minimise contamination and how any contamination was dealt with during analysis. I would also have liked to have seen a short section examining any meteorological influence and perhaps some short scale back trajectory analysis to attempt to define source regions. I recommend publication after the following comments have been addressed.

Specific comments

L65: Please be cautious of overinterpreting these results. A major criticism of these findings is that it is not possible to separate nucleation processes from scavenging, which should be noted. You may also wish to mention the bioprecipitation hypothesis in this section too, e.g., Morris et al., (2014).

L105: Whitehead et al., (2016) demonstrated up to 90% of detected particles at a Brasilian rainforest site to be PBAP, and likely fungal spores. They also demonstrated a strong, RH driven, diurnal variation in PBAP, which is consistent with arguments you make later in the paper so I recommend citing this work here.

L119: Please include the altitude of the site.

L123: Please state the start and end time and dates of sampling.

L139: Please include a description of the sample handling procedure, including any steps taken to minimise contamination, e.g., as in Smith et al., (2018). Were substrate holders and the impactor assembly sterilised in any way prior to sampling? If so, how and with what frequency? I appreciate that you are not performing DNA extraction

analysis or any other methods that require strict handling/contamination protocols in this study, but I feel it is a significant weakness to not include this information as it is needed to assess the reliability of your results. Have attempts been made to screen out biological particles introduced by contamination? If so can you quantify the amount of contamination?

L202: Here you state that more PBAP were observed at night than during the day. This is not a particularly novel result so I would ask the authors to include some citations to previous studies to contextualise this. Strong, RH driven, diurnal variation in PBAP concentrations at forest sites has previously been demonstrated by Crawford et al., (2014,2015), Gosselin et al., (2016), Toprak and Schnaiter (2013) & Whitehead et al., (2016) for example.

L202/Fig.3: I would like to see some of the data from the images tabulated here. Would it be possible to provide statistics of the particle size and aspect ratio for each of the PBAP types observed?

L210: Can you comment on the possibility of particle misclassification and how this is handled in subsequent analysis.

L211: Can you please comment on how the inlet system used may have impacted your ability to detect pollen? If the inlet was fitted with a PM10 head then it would be expected that the majority of pollen would be too large to be sampled.

L296: Please contextualise this with other results in the literature as suggested earlier.

L298: A short section here examining the influence of other meteorological factors (e.g., wind speed/direction) and possibly short time scale back trajectory analysis would strengthen the paper as this would be useful to attempt to define source regions. Are higher counts observed at higher wind speeds or from specific wind sectors for example?

L38/L352: I feel that the term full database overstates the work presented here as

the particles are only broadly sub-classified and only a few select parameters are presented. Please scale this back. For me, a full database would require deeper classification with comprehensive statistics presented for each phyla or species as appropriate, which is lacking here.

Technical corrections

L45: Too general. Please rephrase. E.g., "At this boreal forest site...."

L139: "a diameter of..."

L233/Fig. 6: I'm not sure that bubble is the correct term. Suggest protrusion or protuberance.

L369: Rephase this sentence as it doesn't make sense as it is written. It may need splitting into two or more sentences.

Fig.3: Define day and night in the caption.

References

Crawford et al., (2014): Characterisation of bioaerosol emissions from a Colorado pine forest: results from the BEACHON-RoMBAS experiment, Atmos. Chem. Phys., 14, 8559-8578, https://doi.org/10.5194/acp-14-8559-2014

Crawford et al., (2015): Evaluation of hierarchical agglomerative cluster analysis methods for discrimination of primary biological aerosol, Atmos. Meas. Tech., 8, 4979-4991, https://doi.org/10.5194/amt-8-4979-2015

Gosselin et al., (2016): Fluorescent bioaerosol particle, molecular tracer, and fungal spore concentrations during dry and rainy periods in a semi-arid forest, Atmos. Chem. Phys., 16, 15165-15184, https://doi.org/10.5194/acp-16-15165-2016

Morris et al., (2016), Bioprecipitation: a feedback cycle linking earth history, ecosystem dynamics and land use through biological ice nucleators in the atmosphere. Glob

[Figure]

Chang Biol, doi: 10.1111/gcb.12447

Smith et al., (2018), Airborne Bacteria in Earth's Lower Stratosphere Resemble Taxa Detected in the Troposphere: Results From a New NASA Aircraft Bioaerosol Collector (ABC), Front. Microbiol., doi:10.3389/fmicb.2018.01752

Toprak and Schnaiter (2013).: Fluorescent biological aerosol particles measured with the Waveband Integrated Bioaerosol Sensor WIBS-4: laboratory tests combined with a one year field study, Atmos. Chem. Phys., 13, 225-243, https://doi.org/10.5194/acp-13-225-2013

Whitehead et al., (2016): Biogenic cloud nuclei in the central Amazon during the transition from wet to dry season, Atmos. Chem. Phys., 16, 9727-9743, https://doi.org/10.5194/acp-16-9727-2016

---

## Referee Comment (RC2) · Anonymous Referee #2 · 8 Aug 2019

Review for

**Morphology, mixing state, and hygroscopicity of primary biological aerosol particles from a Chinese boreal forest**

Author(s): Weijun Li et al.
Journal: ACP
MS No.: acp-2019-539

The manuscript "Morphology, mixing state, and hygroscopicity of primary biological aerosol particles from Chinese boreal forest" from Li et al. presents a physical and chemical characterisation of aerosol particles collected at a boreal forest site in China. The authors (i) derive an identification of large taxonomic classes (i.e., bacteria and fungi) from the particle's morphology and chemical composition, (ii) analyse the relative abundance of large particle classes as a function of day and night cycles, and (iii) analyse the hygroscopic growth of the collected particles. These results were obtained from transmission electron microscopy (TEM) and scanning electron microscopy (SEM) with energy-dispersive x-ray spectroscopy (EDS) analyses. Ultimately, the authors derive quantitative concentrations of certain bioaerosol classes and speculate on their potential roles in clouds and in precipitation formation.

Most of the paper is based on rather established concepts of bioaerosol cycling and techniques for bioaerosol analysis (i.e., SEM and TEM, hygroscopic growth studies, etc.). In my view, the really new aspects are the analysis of bioaerosol samples from this particular Chinese boreal forest site, which may allow interesting comparisons with other (boreal) forest sites worldwide as well as the quantification of bacteria and fungal spore concentrations. Thus, the aim and focus of the study is clearly a useful one.

However, I am very concerned about the overall quality of the manuscript – formally as well as scientifically. Formally, the paper is (i) not well structured, (ii) the introduction is just a loose collection of previous literature without really motivating the present work, (iii) the summary rather lists speculations than provides rigorous conclusions, and (iv) the language should be improved. Scientifically, crucial aspects of the analysis are poorly or even not at all explained. Moreover, I am sceptical if certain key results of the study are correct. My **major points of criticism** can be summarized as follows:

- The meaning and use of "bioaerosol identification" seems very problematic in this study. The authors state for example *"As a result, P derived from the particle EDS analysis coupled with the morphological features can be used to identify the PBAPs."* First of all, it is not clear what the authors exactly mean by "identification". In some case this seems to mean discrimination of biological and non-biological particles, whereas in other cases it seems to mean taxonomic determination. Moreover, a fundamental question of this work, which remains unanswered, is to what extent SEM/TEM analysis allows an identification of certain (taxonomic) groups within the total bioaerosol population and which uncertainty this involves. I don't doubt that several aerosol particles can be recognized as biological based on their morphology, surface texture and so on. Also, certain fungal spores (the characteristic ones) can be identified taxonomically based on their appearance as shows in previous studies. However, I am sceptical if any clear discrimination between bacteria and fungal spores (as stated in this study) can be obtained. In both classes, the morphological diversity is large. Many of the "bacteria" that the authors show (e.g., Fig. 2, 4, 5) are pretty large, which rather advocates for fungal spores. In fact, I have the impression that many fungal spores are 'sold' here as bacteria (i.e., see increase of bacteria fraction towards 10 µm in Fig. 12). To point out some specific examples: (i) Some particles in Fig. 2a, which are classified as "fungi", resemble *Bacillariophyceae* (algae). Note here that the potential presences of algae and archaea is not mentioned/considered at all in the study. Moreover, the terms fungal spores and fungi are not discriminated carefully. (ii) In Fig. 2d, particles that resemble bicellular fungal spores are classified as bacteria. (iii) Also many cells in Fig. 5 resemble – in my view – fungal spores rather than bacteria. I have been in frequent contact with mycologists, who use morphological features for fungal spore taxation. Their

procedures follow very careful, iterative, and conservative guidelines for taxonomic identifications/classifications. Diametrically, the approach I see in this work does not refer transparently to any guidelines at all and further appears to be quite 'spontaneous' and suspect. Since the discrimination of bacteria and fungal spores is a core piece of the entire study, I feel that the aforementioned deficits severely challenge the experimental basis of this work.

- The experimental section is intransparent in terms of central pieces of the analysis. Examples: (i) The "identification" and quantification procedures of the taxation remain unclear. What were the exact criteria/guidelines to discriminate bacteria, fungal spores, and "other biological particles"? What are the uncertainties involved here? (ii) How exactly were the brochosoms quantified? Brochosoms tend to occur in (often quite large) clusters. Did you count clusters or individual brochosom entities to obtain the brochosom number fraction of 24 %? (iii) Relevant information in the context of the hygroscopic growth experiments are missing – e.g. uncertainty of RH measurement; how exactly $D_0$ (the diameter of the bioaerosol particle) was obtained, which is not trivial for a rod-shaped particle; etc. (iv) Do we expect a deformation of the cells upon impaction, which may hamper the morphological characterization?

- The caption of Fig. 14 suggest an SEM analysis was conducted prior to the hygroscopic growth experiments. Do you expect to see an authentic/representative hygroscopic response after the harsh treatment with the electron beam and the beam damage involved? I have strong reservations here.

- The study does not translate the results obtained (though questionable) into any meaningful conclusions. The conclusions section is a summary of (i) established and partly trivial statements such as *"The TEM and SEM observations both showed that the morphology of PBAPs were unique and different from that of sulfate, mineral, soot, organics, and metal particles in continental air."* or *"PBAPs from the natural source may have an important role in precipitation and cloud dynamics in the background areas."* and (ii) grotesque overstatements such as *"In this study, we establish one full database that includes the morphology and composition of bacteria, fungi, and brochosomes, and it can be used to identify primary biological particles using single particle techniques."* or *"Our results indicate that significant amounts of PBAPs are emitted from the Khingan Mountain area acting as the "green ocean" […] in Northeast Asia, and they have an important impact on clouds and climate in Northeast China and in the downwind North Pacific Ocean."*.

In my view, the paper is not publishable in the current form and needs a pretty fundamental major revision.

In the following, I listed some **more specific and/or minor comments**:

1. **Introduction:**
- In general, the introduction contains some important points. However, the structure and flow of argumentation needs improvement. The text should be more structured from general to detailed information, finally leading to the guiding research question(s) of this work. This might help to highlight the targeted knowledge gap and to emphasize the importance of the study.
- Some information and reference should be placed in more appropriate location in the text. Currently, certain statements occur redundantly. The text should be more structured in content related segments. Resulting segments should be related.
- Linking thoughts between statements/sentences is often missing. Shortening sentences will improve clarity and the flow of reading.
- p.4/l. 53 "key elements", if this term is used, please briefly indicate in which way they are key elements in the life cycle (e.g. dispersal units).
- p.4/l. 56 "large proportion" is too imprecise. You can give some numbers here?
- p.4/l. 58-59 "Research interest in biological aerosol has been growing significantly in recent decades". To demonstrate the relevance of PBAPs, I suggest to relate this statement to other statements like the fact that bioaerosols can act as CCN or IN like you show in l. 59-60.
- P.4/l.68-72 Better structure needed. Try to summarize information and try to avoid redundancy. E.g.: You already gave some information about the abundance at distinct sites (l.56-57).
- P.4/l.68 "significantly contribution" – can you further specify this?
- P.5/l.73 "the sampling" What does that mean? Aerosol sampling methods?
- P.5/l.76-80 The information and mentioned studies in the two sentences again appear unrelated to the present study.
- P.5/l.81 "chemical composition" is a bit too specific. In my opinion you rather try to identify present kinds of organisms, domains up to species (plant or animal debris, bacteria, fungi, viruses, etc.) by means of biochemical markers or nucleic acids.
- P.5/l.84-86 "These comprehensive and detailed studies of time- and size-resolved PBAPs and their biochemical markers do not well explain the physical properties (e.g., morphology, phase, hygroscopicity, and mixing state) of individual PBAPs in the atmosphere" The sentence is hard to understand. In this context, "studies of time- and size-resolved PBAPs" is not clear.
- P.5/l.87-88 The sentence is nebulous.
- P.5/l.90 What means "actual state"?
- P.6/ l.98-100 The information about the sodium salt in this sentence is redundant (see p.4/l. 61-63). Also, "fungal fragments sampled from Amazonia contain hygroscopic sodium salts based on an environmental scanning electron microscopy" This sentence is not smooth.
- P.6/l. 100-101 "However, whether fungal spores emitted by boreal forests are similar to the fungal spores in central Amazon forests, which contain sodium salts, has not been resolved" Here you should define why it might be important to find out if the fungal spores are similar. Furthermore, you should point out why you think they might be similar, or even not. Is that important or does that lead to the research question of the current paper? You should make clear why that leads to the required analysis (connection to sentence, l.102-103 "Therefore, the morphology, elemental composition, and mixing state of individual PBAPs (nanometre to micrometre size) collected from other global forests must be analysed").

2. **Method:**
- If microscopic techniques are not introduced in more detail already in the introduction, it would be good to highlight the difference between the two techniques, as well as the respective advantages. For SEM you describe shortly the principle of the method (2.3). You should do that also for TEM (in 2.2) to point out the differences and the advantages. Why you are using two different methods? Also for EDS a short description would be nice. Moreover, you mentioned ESEM within the abstract, but you don't mention it in the method section again.
- It is not easy to understand the functional principle of the IPH system. An illustration of the setup might be helpful. Moreover, your experimental steps are not described clearly. The experimental procedure is described incompletely. More information is needed - e.g.: In which steps did your increase or decrease the RH? Which time was needed?

- Moreover, it would be interesting to learn more about the functional principle of the environmental chamber, too. I am wondering if you did some calibrations for the RH measurements?
- Finally, in the method section the analysis of the quartz-fibre filters is totally missing.
- P.7/l. 123-125 "Because boreal forests play a key role in biological aerosol emissions during summer, we collected aerosol samples in August.". What means "key role" here? The sentence states not clear enough why you chose August for sampling time. What did you expect to observe at this specific time period in contrast to other months?
- P.7/l.126-130 The first sentence is definitely too long. You can split the information for a better understanding. You are using two different types of collection substrate. What is the reason? "DKL-2 sample" Can you describe the sampler in more detail? Is it an abbreviation? The sampling times are listed in a confusing way (21:00 vs. 2:00 a.m.)! "every day" – What is the exact sampling period? How many days did you continue the sampling (dates)? Did you use both, copper grids and silicon waver, during each sample event? The size range of collected particles is missing.
- P.7/l. 132 "microscopy" Please mention the type of microscope.
- P.7/l. 133 "suitable" You should define what suitable means.
- P.7/l. 134 "guarantee" Here the information, how the procedure can guarantee the separation of the particles, is necessarily to be mentioned in the text.
- P.7/l. 137-138 Syntax.
- P.8/l. 141-142 Is the placement the same for the first sampling set (DKL-2 sampler, described on p.7) too? If yes, you should make this clearer or add the placement of DKL-2 sampler.
- P.8/l. 160 "Particles in 3-5 grids of each sample were analysed…". It should become clearer how many samples were analysed. How many particles were roughly analyzed on every grid? This information is important to show if and in which way the results are representative (as you point out in p.9/l. 161).
- P.9/l.161-162 "TEM can determine…" here you speak only about TEM. Actually, it is EDS by means you can determine the elemental composition.

**3. Results and Discussion**
- P.10/l.190 Which technique was used here? TEM or SEM?
- P.10/l.200 "number fractions of size resolved aerosol particles" How was this measured/determined? Please outline in experimental part.
- P.12/l.226 "a majority" How representative is Figure 2 a and b for the whole sample set?
- P.13/l.255 "resemble parts of insects" Here is a reference missing? It would be good to describe the features you interpret here.
- P.13/l.257 You should describe in which way the SEM provides "better and more detailed information".
- P13/l.260 "Bacterial particles range from … ". Can this be substantiated with literature?
- P.13/l. 264 "This is because certain hygrophobic secretions of insects (e.g., leafhoppers) are composed of brochosomal particles, and these secretions function in keeping the insect cuticle dry". An explanation/definition of brochosoms should be given earlier in the text (intro or experimental part).
- P.14/l.271-277 Calculations need further clarification.
- P.14/l.286 Please explain what you mean with "differential removal".
- p.16/l. 328-329 "was performed and it showed that bacterial and fungal spores are dominant" This should be clarified in the method section.
- P.16/l. 334 "weak". Please put weak in a context of literature data. If different GFs are compared, the RH of the corresponding the GF should be mentioned for meaningful comparison.
- P.17/l.342-344 "We integrated the morphological, chemical composition and the low growth factor data of individual PBAPs and further concluded that certain hydrophilic organic species might enhance the PBAP size at higher RH". Meaning of sentence nebulous.

**4. Atmospheric implications and conclusion**
- In this section, some aspects are explained too detailed and are therefore redundant at this point. Also here it is important to highlight the main message of the results shortly before you give your conclusion.
- P-17/l. 352-353 "one full database …". This appears to be overstated.
- p.18/l. 360-361 "The growth factor of the bacterial and fungal spores is ~1.09 at 94%, suggesting that some hydrophilic organic species might enhance the size of PBAPs at higher RH". Need clarification.
- P.18/l. 362-366 This statement lacks context here and seems disconnected from the conclusions.
- P.18/l.367-368 "green ocean" This term seems pretty inappropriate for the comparatively small boreal forest area.
- P18/l. 368.369 "they may have an important impact on clouds and climate in Northeast China and in the downwind North Pacific Ocean". This sentence may be true, but seems pure speculation here as it is not related to the results/conclusions of this work.
- P.18/l.369-372 This is another long and nebulous sentence that appears quite speculative. Why speculating about "submicron" particles here?

**Figures:**

**Figure 1**: More precise information may help here to get a feeling for the size of the Khingan area.

**Figure 4**: Where exactly were the EDS spectra obtained?

**Figure 5**: The green framing seems rather confusing/distracting than helpful.

**Figure 6**: Where exactly was the EDS spectrum obtained?

**Figure 7**: Colouring micrographs in this way without any obvious reason seems to violate the widely accepted practise among microscopists to keep the images are raw as possible.

**Figure 8**: Where exactly was the EDS spectrum obtained?

**Figure 9**: See comment on Fig. 7.

**Figure 12**: What exactly do we learn from the ratio of PM10 and PM2.5?

---

## Editor Comment (EC1) · Alex Huffman (Editor) · 13 Sep 2019

Reading the comments from Reviewer #2 again, s/he raises some fundamentally important concerns in the overview of "major points of criticism" starting half-way down the first page. At that point s/he listed four major bullets to concerning the experimental assumptions of the study. S/he also listed four overall comments related to the organization and writing quality in the paragraph before. Lastly, s/he listed several pages worth of detailed comments, some of which are quite significant in themselves. I think all of these comments are relevant and worth carefully considering.

I think it may indeed be possible to revise the manuscript sufficiently to raise it to a

publishable quality, but that will depend on the nature of the revisions. Most fundamentally, Reviewer #2 raised a number of concerns about the methods by which individual particles were classified. The lack of transparency on this issue indeed is one of the major areas of required improvement. After carefully adding specific details about how particles were assigned and categorized, it will be easier to evaluate the observations and conclusions. Without knowing more about the process by which particles were investigated and assigned, it is hard to know if the method itself was sufficient to support the conclusions. The question here is not just about clarifying wording, but that the clarified wording will help evaluate whether the method was sufficient or not. In particular, the question of whether the particle assignments were correct is not sufficiently addressed. Just because a systematic method is established is not a sufficient criterion to determine whether the method leads to correct assignment. For example, consider a skeptical scientist reader. Convince them that your method led to detection and consistent, correct categorization of the particle types you report.

Somewhat independent of the comments from the two reviewers, my suggestions are two-fold:

- The observations and atmospheric implications are relatively similar to works that have used similar techniques in both boreal and tropical areas, but these are not well cited in the manuscript. At a minimum I suggest you to consider additional literature, and make sure to at least briefly compare results with these in mind. I suggest doing a good literature survey of PBAP observations from forests, as well as a search related to ambient studies related to microscopy of PBAP (i.e. SEM and TEM, as you use). Then make sure that the observations you present and the conclusions you draw are put into context of these previous measurements.

- The statement in the manuscript that the work can be used as a "full database" to "be used to identify primary biological particles using single particle techniques" is overstated in my opinion. I think that that study can be revised to show an overview of observations of (bio)-particles in this region, but using the results as a database for

future reference is entirely different. This would require a substantially higher threshold of demonstrated quality, which may or may not have been achieved. To claim these data as basis for a database of atmospheric particles implies that a sufficiently systematic representation of bioparticles has been sampled and correctly analyzed. Further, the database would need to show some sort of independent verification that particles were assigned correctly, similar to above comments. Since most methods of independent identification are well beyond the scope of the work you reported, I do not expect that you would want to argue that the assignments have necessarily been verified as correct. They are merely suggestions, with uncertainties and potential assignment errors to be at least briefly discussed in the revised manuscript. So in that case I would suggest that at a minimum you scale back the conclusions to remove the concept of 'database,' and report in the context of 'observations'. This does not get around the first concerns that Reviewer #2 raised about categorization of particle type, but it will help to re-frame the conclusions a bit.

I strongly suggest keeping all these comments, including those from the two Reviewers, closely in mind as you revise and respond to all comments line-by-line. I look forward to reviewing the revised manuscript when available.

Best regards,

Alex Huffman

---

## Author Comment (AC1) · 4 Nov 2019

**General Response: We thank the Referee#1 for your helpful comments. We have addressed all comments and provided point by point response below. The revised manuscript is presented in below Response**

1) This paper examines the composition of primary biological aerosol particles (PBAP) collected on various substrates for offline analysis, at a mountainous boreal forest site in China. Particles were classified optically based on morphology and composition was determined using a combination of TEM & EDS. The authors report that PBAP were found to contain key, unique compositional markers (e.g., elemental P), which is consistent with previous studies performing similar analysis. A key result of this study was demonstrating that 20% of bacterial particles were internally mixed with non-PBAP, which may have a significant impact on the long range transport of bacteria and aerosol budgets as well as mixed-phase aerosol-cloud interactions. The authors also examined PBAP hygroscopicity, demonstrating that the sampled PBAP display small growth factors and subsequently weak hygroscopicity. Overall the paper is reasonably well written and provides useful information to be absorbed into our general understanding of PBAP emissions and quantifying the fraction of PBAP which are internally mixed is a key result.

Response: We appreciated the referee's positive comments.

2) My only significant criticism is that the paper lacks detail on the sample/substrate handling procedure employed, what procedures were in place to minimise contamination and how any contamination was dealt with during analysis.

Response: We added some explanations about the sample/substrate here. In the section 2.1, we explained the sample storage that the Cu grids and silicon wafers placed in a dry, clean, and airtight container were stored in a desiccator at 25 ℃ and 20±3% RH to minimize exposure to ambient air and preserve them for analysis. Because the TEM can directly observe the dry samples, the procedures before the TEM analysis can guarantee no contamination. The method is widely used in many studies for individual particle analysis. We added one Figure S1 in the revised manuscript as below.

[Figure]

**Figure S1** The sampling procedures of substrate, sampler, storage, and analyzed technique.

To further confirm the procedure about the PBAPs, we obtained *Yeast* and *colibacillus* cultivated in the laboratory (Figure 5 and S2). We found the TEM analysis were no problem to obtain morphology of bacteria and fungi.

[Figure]

**Figure S2** The *Yeast* and the *colibacillus* particles cultivated in laboratory. TEM image showing morphology and EDS showing compositions.

As these two referee's requests, I added more explanations here to provide details about the sampling procedure and detailed analysis. We mainly revised the Method section, please read the RED words.

In context P154-163 "Individual particle samples were collected both on copper (Cu) TEM grids coated with carbon film (carbon type-B, 300-mesh copper; Tianld Co., China) and on silicon membranes (thickness: 500±10 μm, size: 3×3 mm; LIJINGKEJI, China) by a single-stage cascade impactor called the DKL-2 sampler (Genstar Electronic Technology, China). The collection efficiency of the impactor is 50% for particles with an aerodynamic diameter of 0.1 μm when we assume an aerosol particle density of 2 g cm$^{-3}$. We collected individual particles four times each day at 9:00, 15:00, 21:00, and 02:00 local time. At each sampling event, we first collected TEM grids and then changed to silicon wafers in the sampler. The sampling duration at each time varied from 10 min to 25 min depending on the particle distribution on the substrate. The substrates of the carbon film and silicon wafer both have smooth surfaces with no contamination before we use them to collect aerosol particles."

P144-149 "The distribution of aerosol particles on TEM grids was not uniform, with coarser particles occurring near the center and finer particles on the periphery. The quick check by the optical microscopy enabled us to tell whether individual particles were well distributed and whether there was any overlap on the substrate. Whenever the distribution was not even enough or when substantial overlap occurred, we had to discard it and re-collect individual particle samples through adjusting the sampling duration."

"TEM clearly shows the morphology of particles smaller than 2 μm. For some larger particles, we might further carry the scanning electron microscopy (SEM) experiments to determine their morphology. In this study, we did observe one fungi (*Yeast)* and one bacteria (*colibacillus)* sample through TEM, which were prepared in biological laboratories (Figure S2). Microscopic observations from the bacteria and fungi samples prepared in the laboratory were helpful to classify PBAPs emitted from the forest.

Once we clearly obtained electron images of different particles, we could then measure particle size and shape factors. In this study, the area, perimeter, shape factor, and equivalent circle diameter (ECD) of individual particles in TEM images are manually or automatically obtained through an image analysis software (RADUS, EMSIS GmbH, Germany). Based on these measurements, we can classify particle types and determine the diameter and shape factor of individual particles among different particle types. Moreover, we statistically analyze the number fractions in different size bins."

3) I would also have liked to have seen a short section examining any meteorological influence and perhaps some short scale back trajectory analysis to attempt to define source regions.

Response: We added the back trajectories of air mass (Figure 1). Figure S1 listed the meteorological data during the sampling period.

**New section 2.5 "Meterological data and back trajectories**

**"Meteorological data and back trajectories**

Meteorological data, including the relative humidity (RH), temperature, wind speed, and wind direction, were measured and recorded every 5 min by an automated weather meter (Kestrel 5500, USA). During the sampling period, the relative humidity (RH) and temperature varied from 40-70% and 22-28 ℃ during the day and 90-100% and 10-15 ℃ during the night, respectively. The wind speed was 1.5-7.6 m s$^{-1}$ during the day and 0-1 m s$^{-1}$ at night (Figure S4).

To determine the regional transport of air masses, 6-h back trajectories of air masses were generated using a Hybrid Single Particle Lagrangian Integrated Trajectory (HYSPLIT) model at the forest sampling station during 14-21 August, 2016. Based on the sampling times of each day at 09:00, 15:00, 21:00, and 02:00 (midnight) local time, we performed 31 air mass back trajectories. Here we selected an altitude of 500 m as the end point of each back trajectory (Figure 1). Figure 1 shows that all the back trajectories in the past 6-h had been transported over the Lesser Khingan Mountain forest.

[Figure]

**Figure 1** Location of the sampling site and 6-h air mass back trajectories arriving at each sampling time from 14-21 August, 2016 in a boreal forest of the Lesser Khingan Mountain in Northeast China. The map source is Google Earth.

4) L65: Please be cautious of overinterpreting these results. A major criticism of these findings is that it is not possible to separate nucleation processes from scavenging, which should be noted. You may also wish to mention the bioprecipitation hypothesis in this section too, e.g., Morris et al., (2014).

Response: We thank the referee's comments here. We add the possible pathway and reference here. Here we cite this paper Morris et al., (2016), Bioprecipitation: a feedback cycle linking earth history, ecosystem dynamics and land use through biological ice nucleators in the atmosphere. Glob Chang Biol, doi: 10.1111/gcb.12447

The sentence was changed to P70-72 "These studies addressed the hypothesis that PBAPs indeed influence the hydrological cycle and climate by initiating the formation of clouds and precipitation as CCN and IN or the bioprecipitation feedbacks."

5) L105: Whitehead et al., (2016) demonstrated up to 90% of detected particles at a Brasilian rainforest site to be PBAP, and likely fungal spores. They also demonstrated a strong, RH driven, diurnal variation in PBAP, which is consistent with arguments you make later in the paper so I recommend citing this work here.

Response: Thanks to provide such useful paper. Of course we need to cite it.
Whitehead et al., (2016): Biogenic cloud nuclei in the central Amazon during the transition from wet to dry season, Atmos. Chem. Phys., 16, 9727-9743, https://doi.org/10.5194/acp-16-9727-2016

We also revised the part about the RH and PBAPs concentration and added the reference here.
P465-471"A similar phenomenon has been observed in different forests, such as the Amazon rainforest (Huffman et al., 2012;Whitehead et al., 2016), a montane ponderosa pine forest in North American (Crawford et al., 2014), a semi-arid forest in the southern Rocky Mountains of Colorado (Gosselin et al., 2016), and a semi-rural site in southwestern Germany (Toprak and Schnaiter, 2013). These studies above found that a nighttime peak of number concentrations of fluorescent biological aerosol particles is consistent with nocturnal sporulation driven by the increased RH."

6) L119: Please include the altitude of the site.
Response: Added the altitude here
Sentence revised: P125-127 "The sampling site is at the Heilongjiang Liangshui National Nature Reserve (47.32° N, 128.54° E; 350m above sea level) in the center of the Lesser Khingan Mountains of northeast China (Figure 1)."

7) L123: Please state the start and end time and dates of sampling.
Response: Added
Sentence changed to p130 "Because there is less rain in late Auguest, we selected 14-21 August, 2016 to collect the bioaerosol samples."
P137"We collected individual particles four times each day at 9:00, 15:00, 21:00, and 02:00 local time."

8) L139: Please include a description of the sample handling procedure, including any steps taken to minimise contamination, e.g., as in Smith et al., (2018). Were substrate holders and the impactor assembly sterilised in any way prior to sampling? If so, how and with what frequency? I appreciate that you are not performing DNA extraction analysis or any other methods that require strict handling/contamination protocols in this study, but I feel it is a significant weakness to not include this information as it is needed to assess the reliability of your results. Have attempts been made to screen out biological particles introduced by contamination? If so can you quantify the amount of contamination?

Response: We appreciated that the referee carefully provided the reference here. In our experiment, the commercial TEM grids are made in superclean room in a company. The carbon film must be very clean and smooth before we use it. The silicon wafers are also made in superclean room in a company. It should be noted that the silicon wafers are covered by the plastic film to protect the smooth surface. Before we use it, we take the film off. Therefore, the substrates are no possible contamination. The holders and impactor are cleaned using alcohol before and after we use them in the field campaign.

We made one Figure in supplemental Figure S2 to explain our substrate and samplers. We added more steps to explain how to collect and storage the samples. In this study, because we didn't performing DNA extraction analysis, we only observe the particles on the substrate. Therefore, we only need to make sure the clean substrate before we used the substrate. Also, we need to make sure the storage condition in dry and sealed capsule. There is no possible to contact the contamination between sample and air. Also, in the condition, there is no condition for bacteria growth or other activity. In each field campaign, we prepared one blank sample as the background filter. The procedure might quantify the amount of contamination using the background filter through the same procedure. After we check the blank substrate, we didn't found any bacterial particles or fungus on the TEM grid and silicon wafer. Therefore, we can guarantee there is no contamination in the storage conditions. Based on two referee's comments, we further did the standard samples under the same procedure. The storage of PBAPs samples is no problem to obtain morphology of individual PBAPs.

[Figure]

**Figure S1** The sampling procedures of substrate, sampler, storage, and analyzed technique.

9) L202: Here you state that more PBAP were observed at night than during the day. This is not a particularly novel result so I would ask the authors to include some citations to previous studies to contextualise this. Strong, RH driven, diurnal variation in PBAP concentrations at forest sites has previously been demonstrated by Crawford et al., (2014,2015), Gosselin et al., (2016), Toprak and Schnaiter (2013) & Whitehead et al., (2016) for example.

Response: We appreciated the referee to provide such good references. All of them were cited in section 3.2 in the revised manuscript.

P465-471 "A similar phenomenon has been observed in different forests, such as the Amazon rainforest (Huffman et al., 2012; Whitehead et al., 2016), a montane ponderosa pine forest in North American (Crawford et al., 2014), a semi-arid forest in the southern Rocky Mountains of Colorado (Gosselin et al., 2016), and a semi-rural site in southwestern Germany (Toprak and Schnaiter, 2013). These studies above found that a nighttime peak of number concentrations of fluorescent biological aerosol particles is consistent with nocturnal sporulation driven by the increased RH."

10) L202/Fig.3: I would like to see some of the data from the images tabulated here. Would it be possible to provide statistics of the particle size and aspect ratio for each of the PBAP types observed?

Response: Sure, we measured the data about the statistic of the particle size and aspect ratio of the PBAPs.

[Figure]

Figure 6 Size distribution and aspect ratios of rod-like PBAPs, fungal spores, and brochosomes collected in boreal forest air.

11) L210: Can you comment on the possibility of particle misclassification and how this is handled in subsequent analysis.

Response: As the two referee's comments about the classification, we did laboratory experiments to confirm the bacteria and fungi. Indeed, it is difficult to identify them based on their morphology from the electron microscopy. In the revised manuscript, we named "rod-like PBAPs" which contain bacteria and fungi. Besides the fine particles, TEM and SEM can clearly classify fungal spores and brochosomes due to their unique morphology. These two types of PBAPs have been well documented by SEM in many previous studies.

In the revised manuscript, we did more investigations from the literature and communicate more experts who concern the biological molecule, ecology, and bacteria. As the reason, we did additional laboratory experiments to confirm our classification. We believed that the revised classification can precisely deliver the right message for the potential readers.

12) L211: Can you please comment on how the inlet system used may have impacted your ability to detect pollen? If the inlet was fitted with a PM10 head then it would be expected that the majority of pollen would be too large to be sampled.

Response: The inlet is no problem to collect the large particles if pollens do exist in air. We collected coarse particles in desert and ocean air to study dust and sea salt particles before (Chi et al., ACP, 2015; Li et al., JGR, 2013). As the editor's suggestion, we did search more literature. Indeed, there was no pollen in August in boreal forest (Manninen et al., 2014).

In context p269 "Pollen was not found in our samples, which may be because large pollen emissions occur in spring and early summer instead of late summer (August) in boreal forests (Manninen et al., 2014)."

13) L296: Please contextualise this with other results in the literature as suggested earlier.

Response: Thanks. We carefully revised the part.

14) L298: A short section here examining the influence of other meteorological factors (e.g., wind speed/direction) and possibly short time scale back trajectory analysis would strengthen the paper as this would be useful to attempt to define source regions. Are higher counts observed at higher wind speeds or from specific wind sectors for example?

Response: Thanks. We added the back trajectories into the map. The wind speeds were shown in Figure S2. We also made discussion here about the wind speed.

In context p444-459 "It is well documented that meteorological conditions such as RH, wind speed, and temperature can affect PBAPs emission in the forests (Harrison et al., 2005;Whitehead et al., 2016). In particular, the wind speed is especially important in promoting PBAPs emission into air. During the sampling period, the average wind speeds at 5 min intervals had a range from 0 to 7.5 m/s with a mean value of 0.75 m/s. 89% of the measured wind speeds were lower than 2 m/s (Figure S4). Therefore, we conclude that no large consistent wind speeds occurred during the sampling period. Furthermore, we compared all the air mass back trajectories in the past 6-h over the Lesser Khingan Mountain forest at each sampling time (Figure 1). There are similar lengths of these back trajectories, suggesting that wind speeds above the forest canopy had only small changes during the sampling period. Therefore, the result from the ground-based measurements of wind speeds is consistent with air mass back trajectories. Here, we can exclude wind speeds during the sampling period as one important factor to dominate PBAPs emissions during day and night in the boreal forest. High temperatures normally increase the PBAPs emissions from the plants in the daytime (Harrison et al., 2005). However, we observed contrasting results that more PBAPs occurred in nighttime instead of daytime (Figure S4). Therefore, we also exclude temperatures during the sampling period as a cause of the vastly different PBAPs emissions at day and night in the boreal forest.
"

15) L38/L352: I feel that the term full database overstates the work presented here as the particles are only broadly sub-classified and only a few select parameters are presented. Please scale this back. For me, a full database would require deeper classification with comprehensive statistics presented for each phyla or species as appropriate, which is lacking here.

Response: We noticed that we overstated it. We agreed with the referee's comments: The full database might include deeper classification with comprehensive statistics of each phylya or species. Here for electron microscopes, it is impossible to provide any species of PBAPs. Therefore, we deleted such words and made suitable tone here.

16) L45: Too general. Please rephrase. E.g., "At this boreal forest site: : :."
Response: Revised

17) L139: "a diameter of…"
Response: Revised

18) L233/Fig. 6: I'm not sure that bubble is the correct term. Suggest protrusion or protuberance.
Response: Revised

19) L369: Rephase this sentence as it doesn't make sense as it is written. It may need splitting into two or more sentences. Fig.3: Define day and night in the caption.
Response: Revised

**References**
Crawford et al., (2014): Characterisation of bioaerosol emissions from a Colorado pine forest: results from the BEACHON-RoMBAS experiment, Atmos. Chem. Phys., 14, 8559-8578, https://doi.org/10.5194/acp-14-8559-2014
Crawford et al., (2015): Evaluation of hierarchical agglomerative cluster analysis methods for discrimination of primary biological aerosol, Atmos. Meas. Tech., 8, 4979-4991, https://doi.org/10.5194/amt-8-4979-2015
Gosselin et al., (2016): Fluorescent bioaerosol particle, molecular tracer, and fungal spore concentrations during dry and rainy periods in a semi-arid forest, Atmos. Chem. Phys., 16, 15165-15184, https://doi.org/10.5194/acp-16-15165-2016

Morris et al., (2016), Bioprecipitation: a feedback cycle linking earth history, ecosystem dynamics and land use through biological ice nucleators in the atmosphere. Glob Chang Biol, doi: 10.1111/gcb.12447

Smith et al., (2018), Airborne Bacteria in Earth's Lower Stratosphere Resemble Taxa Detected in the Troposphere: Results From a New NASA Aircraft Bioaerosol Collector (ABC), Front. Microbiol., doi:10.3389/fmicb.2018.01752

Toprak and Schnaiter (2013).: Fluorescent biological aerosol particles measured with the Waveband Integrated Bioaerosol Sensor WIBS-4: laboratory tests combined with a one year field study, Atmos. Chem. Phys., 13, 225-243, https://doi.org/10.5194/acp-13-225-2013

Whitehead et al., (2016): Biogenic cloud nuclei in the central Amazon during the transition from wet to dry season, Atmos. Chem. Phys., 16, 9727-9743, https://doi.org/10.5194/acp-16-9727-2016

**Overview** of primary biological aerosol particles from a Chinese boreal forest: insight into morphology, size, and mixing state at microscopic scale

Weijun Li[1], Lei Liu[1], Qi Yuan[1], Liang Xu[1], Yanhong Zhu[1], Bingbing Wang[2], Hua Yu[3], Xiaokun

Ding[4], Jian Zhang[1], Dao Huang[1], Dantong Liu[1], Wei Hu[5], Daizhou Zhang[6], Pingqing Fu[5], Maosheng

Yao[7], Min Hu[7], Xiaoye Zhang[8], Zongbo Shi[9,5]

[1]Department of Atmospheric Sciences, School of Earth Sciences, Zhejiang University, Hangzhou

310027, China

[2]State Key Laboratory of Marine Environmental Science, College of Ocean and Earth Sciences,

Xiamen University, Xiamen 361102, China.

[3]College of Life and Environmental Sciences, Hangzhou Normal University, 310036, Hangzhou, China

[4]Department of Chemistry, Zhejiang University, Hangzhou 310027, China

[5]Institute of Surface-Earth System Science, Tianjin University, 300072, Tianjin, China

[6]Faculty of Environmental and Symbiotic Sciences, Prefectural University of Kumamoto,

Kumamoto 862-8502, Japan

[7]State Key Joint Laboratory of Environmental Simulation and Pollution Control, College of

Environmental Sciences and Engineering, Peking University, Beijing 100871, China

[8]Key Laboratory of Atmospheric Chemistry, Chinese Academy of Meteorological Sciences, Beijing,

China

[9]School of Geography, Earth and Environmental Sciences, University of Birmingham, Birmingham

B15 2TT, UK

*Correspondence to*: Weijun Li (liweijun@zju.edu.cn)

**Abstract:**

Biological aerosols play an important role in atmospheric chemistry, clouds, climate, and public health. Here, we studied the morphology and composition of primary biological aerosol particles (PBAPs) collected in the Lesser Khingan Mountain boreal forest of China in summertime using transmission electron microscopy (TEM) and scanning electron microscopy (SEM). C, N, O, P, K, and Si were detected in most of the PBAPs, and P represented a major marker to discriminate the PBAPs and non-PBAPs. Of all detected particles > 100 nm in diameter, 13% by number were identified as PBAPs. We found that one type of PBAPs mostly appeared as similar rod-like shapes with an aspect ratio > 1.5 and the dominant sizes ranged from 1 μm to 5 μm. The size distribution of the rod-like PBAPs displays two typical peaks at 1.4 μm and 3.5 μm, which likely are bacteria and fungal particles in the forest air. The second most PBAPs were identified as fungal spores with ovoid, sub-globular or elongated shapes with a smooth surface and small protuberances with their dominant size range of 2 - 5 μm. Moreover, we found some large brochosomal clusters containing hundreds of brochosomes with a size range of 200-700 nm and a shape like a truncated icosahedron. The number size distribution of PBAPs coupled with $PM_{2.5}$ and $PM_{10}$ concentrations were used to estimate the total mass concentration of PBAPs, which is approximately 1.9 μg m$^{-3}$ and accounts for 47% of the in situ $PM_{2.5-10}$ mass. Moreover, there is a higher frequency and concentration of PBAPs at night compared with day, suggesting that the relative humidity dramatically enhances the PBAPs emissions in the boreal forest. Our study also showed that the fresh PBAPs displayed weak hygroscopicity with a growth factor of ~1.09 at RH=94%. TEM revealed that about 20% of the rod-like PBAPs were internally mixed with metal, mineral dust, and inorganic salts in the boreal forest air. This work for the first time provides the overview of individual PBAPs from nanoscale to microscale in Chinese boreal forest air.

**Key points**

• Based on morphology, composition, and size of individual PBAPs, rod-like PBAPs (e.g., bacteria and fungi), fungal spores, and brochosomes were identified.

• PBAPs emissions tend to occur with high humidity at night rather than during the day.

• Hygroscopic experiments show that most of the PBAPs displayed weak hygroscopicity, and their growth factor was ~1.09 at RH=94%.

**1. Introduction**

Primary biological aerosol particles (PBAPs) (e.g., bacteria, spores, fungi, viruses, algae, and pollen) are ubiquitous in the Earth's atmosphere and important elements in the life cycle of many organisms and ecosystems (Poschl, 2005;Tunved et al., 2006;Smith et al., 2018). PBAPs are airborne biological materials that are transported from the biosphere to the atmosphere (Huffman et al., 2010), and they can account for a large proportion (25-45%) of the aerosol particle mass in pristine forest air and certain amounts in some rural and marine air (Elbert et al., 2007;Bauer et al., 2008;Hu et al., 2017;May et al., 2018). The growing research interest in PBAPs has one of its goals to better understand how PBAPs or their cell fragments influence cloud condensation nuclei (CCN) and ice nuclei (IN) (Morris et al., 2004;Huffman et al., 2013;Ling et al., 2018). Furthermore, field campaigns have found that abundant biological aerosols occur in cloud ice-crystals, fog/cloud, rain, and snowfall (Amato et al., 2005;Möhler et al., 2007;Christner et al., 2008;Pratt et al., 2009;Prenni et al., 2009;Tobo et al., 2013;Morris et al., 2014;Wilson et al., 2015;Twohy et al., 2016;Hu et al., 2018). These studies addressed the hypothesis that PBAPs indeed influence the hydrological cycle and climate by initiating the formation of clouds and precipitation as CCN and IN or by their bioprecipitation feedbacks.

Previous studies have investigated particle number concentration, size, and composition of primary biological aerosols using online measurement techniques and advanced molecular biological analyses (Wittmaack et al., 2005;Elbert et al., 2007;Fröhlich-Nowoisky et al., 2009;Huffman et al., 2010;Despré s et al., 2012;Crawford et al., 2015;Hu et al., 2017;Therkorn et al., 2017;Zhang et al., 2017;Chen and Yao, 2018). For example, the contribution of fungal spores to total organic carbon was estimated to be approximately 10% in clean and polluted periods in Beijing using an online wideband integrated bioaerosol sensor (WIBS) (Yue et al., 2017); To obtain the organisms of PBAPs in the atmosphere, many studies tend to detect biochemical markers (e.g., proteins, fatty acids, sugars) and nucleic acids (i.e., DNA and RNA) to determine their origins such as plant or animal debris, bacteria, fungi, or viruses (Georgakopoulos et al., 2009;Chen and Yao,

2018;Hu et al., 2018;Ling et al., 2018). Although these previous studies provided comprehensive species or detailed molecular compositions of PBAPs, they still could not reflect the physical properties of individual PBAPs in the atmosphere, such as morphology, size, phase, hygroscopicity, and mixing state. Besides particle composition, the previous studies have proved that the morphology, size, and mixing state of individual particles more or less influence their CCN and IN

activities and optical properties (Spracklen et al., 2008;Fröhlich-Nowoisky et al., 2009;Wilson et al.,

2015;Li et al., 2016;Ault and Axson, 2017;Riemer et al., 2019). Therefore, it is critical to characterize detailed information of different types of individual PBAPs from their natural sources.

In the past decades, several studies have used scanning electron microscopy (SEM) to characterize the morphology and size of individual PBAPs (Nikkels et al., 1996;Wittmaack et al.,

2005;Coz et al., 2010;Tamer Vestlund et al., 2014;Valsan et al., 2015;China et al., 2018). They identified fungal spores, brochosome, pollen, and plant or insect debris larger than 2 μm in the atmosphere. Although the SEM observations adequately characterized the coarse fungal spores, pollen, and plant or insect debris particles, comparable results have not been obtained for fine bacteria and fungal particles, which together account for a large number of suspended particles in ambient air detected by online instruments (Tong and Lighthart, 2000;Després et al., 2012;Afanou et al., 2014;Valsan et al., 2016;Priyamvada et al., 2017;Hu et al., 2018). The reason for this shortfall is likely that SEM could not clearly observe carbonaceous bioaerosols smaller than 1 μm (Li et al.,

2016;Ault and Axson, 2017). Posfai et al. (2003) and Patterson et al. (2016) used transmission electron microscopy (TEM) to detect some fine bacteria in marine air. However, there is no study to characterize the morphology, size, and mixing state of individual PBAPs from nanoscale to microscale. For example, many studies directly used SEM images showing the coarse PBAPs (e.g., fungal spores) in support of their conclusions, but missed large numbers of fine PBAPs (e.g., bacteria) (Shi et al., 2003;Wittmaack et al., 2005;Coz et al., 2009;Shi et al., 2009;Martin et al.,

2010;Huffman et al., 2012;Tamer Vestlund et al., 2014;Afanou et al., 2015;Valsan et al.,

2015;Valsan et al., 2016;Priyamvada et al., 2017;Wu et al., 2019). The result might discourage people considering fine bacteria and fungal particles for their atmospheric effects or for their examination of data from some online instruments. Therefore, it is necessary to integrate SEM and

TEM to characterize the morphology, size, and mixing state of individual PBAPs from nanoscale to microscale.

Forests are important contributors of primary biological aerosols in the atmosphere (Tunved et al., 2006;Spracklen et al., 2008;Després et al., 2012;Whitehead et al., 2016). Aerosols in large forests contain abundant biological particles from plants emitted locally and lesser amounts of anthropogenic pollutants from long-range transport (Tong and Lighthart, 2000;Tunved et al.,

2006;Gabey et al., 2010;Martin et al., 2010). We chose the Lesser Khingan Mountains in northeast

China, which is its second largest boreal forest. In this study, TEM and SEM both have been employed to characterize the morphology, size, and mixing state of various PBAPs collected over the boreal forest. Furthermore, hygroscopic experiments on the primary biological particles have been conducted.

**2.   Methods**

**2.1 Sampling site and sample collection**

The sampling site is at the Heilongjiang Liangshui National Nature Reserve (47.32 $^{\circ}$N, 128.54 $^{\circ}$

E; 350m above sea level) in the center of the Lesser Khingan Mountains of northeast China (Figure

1). The boreal region is characterized by large seasonal variations in temperature, and the flora is dominated by Korean pine and spruce species. There are no anthropogenic sources of pollutants, such as villages, industries and vehicles within 80 km of the sampling site. Boreal forests have the highest emissions of biological aerosols during summer. Because there is less rain in late Auguest, we selected 14-21 August, 2016 to collect the bioaerosol samples.

Individual particle samples were collected both on copper (Cu) TEM grids coated with carbon film (carbon type-B, 300-mesh copper; Tianld Co., China) and on silicon membranes (thickness:

500±10 μm, size: 3×3 mm; LIJINGKEJI, China) by a single-stage cascade impactor called the DKL-

2 sampler (Genstar Electronic Technology, China). The collection efficiency of the impactor is 50%

for particles with an aerodynamic diameter of 0.1 μm when we assume an aerosol particle density of 2 g cm$^{-3}$. We collected individual particles four times each day at 9:00, 15:00, 21:00, and 02:00

local time. At each sampling event, we first collected TEM grids and then changed to silicon wafers in the sampler. The sampling duration at each time varied from 10 min to 25 min depending on the particle distribution on the substrate. The substrates of the carbon film and silicon wafer both have smooth surfaces with no contamination before we use them to collect aerosol particles. After sample collection,  we  immediately  performed  optical  microscopy  (BST60-100,  China)  at  100X

magnification to determine whether the aerosol distribution on the substrate was suitable for electron microscopy analysis. The distribution of aerosol particles on TEM grids was not uniform, with coarser particles occurring near the center and finer particles on the periphery. The quick check by the optical microscopy enabled us to tell whether individual particles were well distributed and whether there was any overlap on the substrate. Whenever the distribution was not even enough or when substantial overlap occurred, we had to discard it and re-collect individual particle samples through adjusting the sampling duration. In a word, this sampling procedure guarantees that the collected particles were adequately separated and did not overlap each other on the substrate (Li et al., 2016). The Cu grids and silicon wafers were placed in a dry, clean, and airtight container with

25 °C and 20±3% RH which minimizes exposure to ambient air and preserves them for subsequent analysis. The detailed sampling and storage procedures are summarized in Figure S1.

The daily $PM_{2.5}$ and $PM_{10}$ samples were collected on quartz-fiber filters with a diameter of 90

mm through two medium-volume samplers (TH-150, Wuhan Tianhong, China) at a constant flow rate of 100 L $min^{-1}$. The samples were changed at 08:00 a.m. each day. The DKL-2 and TH-150

samplers and other monitoring instruments in the field experiment were installed on a building roof

15 m above ground. The quartz filters (Whatman, UK) were put in polyethylene boxes immediately after sampling and stored at −5 °C. They were equilibrated at a constant temperature (20 ±0.5 °C)

and humidity (50 ± 2%) for over 24 h before being weighed with an electronic microbalance (Sartorius-ME5, Germany). This gravimetric procedure provides the mass concentration of $PM_{2.5}$

and $PM_{10}$.

[Figure]

**Figure 1** Location of the sampling site and 6-h air mass back trajectories arriving at each
sampling time from 14-21 August, 2016 in a boreal forest of the Lesser Khingan Mountain in
Northeast China. The map source is Google Earth.

**2.2 Transmission electron microscopy analysis**

Individual aerosol particles collected on Cu grids were analyzed via transmission electron
microscopy (TEM, JEM-2100, JEOL Ltd., Japan) at a 200 kV accelerating voltage. TEM with a
beam of electrons is transmitted through a specimen to form an image. An image is formed from
the interaction of the electrons with the sample as the beam is transmitted through the specimen.
Therefore, TEM images display the inner physical structure of individual particles and the mixing
state of different components. The TEM system is equipped with an energy-dispersive X-ray
spectrometer (EDS, INCA X-Max$^N$ 80T, Oxford Instruments, UK). EDS is an analytical technique
used for the elemental analysis or chemical characterization of a sample. It relies on an interaction
between X-rays and a sample. EDS spectra show the peaks of different elements and the
contribution of each element in the total. EDS semiquantitatively detects the elemental composition
of individual particles with an atomic number greater than six ($Z > 6$). However, Cu peaks in the
EDS spectra were not considered because of interference from the copper substrate of TEM grids.
We determined the morphology, composition, and mixing state of individual particles through the
combination of TEM and EDS. To reduce the damage to particles under the electron beam, the EDS
collection duration was limited to 15 s. Individual particles are distributed on TEM grids, with the
coarser particles in the center of sampling spot and with the finer particles on the periphery.
Therefore, to guarantee that the analyzed particles are representative, five areas are selected from the sampling center to the periphery on each TEM grid. After a labor-intensive operation, we analyzed 150-250 individual particles with diameters of 100 nm-10 μm in each sample. Finally, we successfully analyzed 20 TEM grids in the study. TEM/EDS can determine the internal mixing structure of different aerosol components in fine particles and their specific composition. TEM

clearly shows the morphology of particles smaller than 2 μm. For some larger particles, we might further carry the scanning electron microscopy (SEM) experiments to determine their morphology.

In this study, we did observe one fungi (*Yeast)* and one bacteria (*colibacillus)* sample through TEM, which were prepared in biological laboratories (Figure S2). Microscopic observations from the bacteria and fungi samples prepared in the laboratory were helpful to classify PBAPs emitted from the forest.

Once we clearly obtained electron images of different particles, we could then measure particle size and shape factors. In this study, the area, perimeter, shape factor, and equivalent circle diameter (ECD) of individual particles in TEM images are manually or automatically obtained through an image analysis software (RADUS, EMSIS GmbH, Germany). Based on these measurements, we can classify particle types and determine the diameter and shape factor of individual particles among different particle types. Moreover, we statistically analyze the number fractions in different size bins.

Aspect Ratio is the maximum ratio between the length and width of a bounding box for the measured object. An aspect ratio of 1 (the lowest value) indicates that a particle is not elongated in any direction. The aspect ratio is defined as

$$AR = \frac{L_{max}}{W_{max}}$$

**2.3 Scanning electron microscopy analysis**

SEM is performed using a type of electron microscope that can determine the particle surface by scanning it with a high-energy beam of electrons in a raster scan pattern. An SEM system (Zeiss

Ultra 55) equipped with a field emission gun operating at 5–20 kV was used to obtain detailed information on the surfaces of individual aerosol particles. Moreover, the SEMx was equipped with an energy-dispersive X-ray spectrometry (EDS), which can analyze the chemical composition of individual particles. The SEM/EDS can efficiently obtain the surface morphology, size, and composition of coarse particles without any coating process on the substrate. Finally, we selected six silicon wafers for SEM/EDS analysis (Figure S1). In this study, we used SEM/EDS to observe surface morphology of the coarse particles on silicon wafers and to confirm particle types which cannot be clearly shown in TEM images.

**2.4 Hygroscopic experiments**

A custom-made individual particle hygroscopic (IPH) system was used to observe the hygroscopic properties of individual biological particles at different relative humidity (RH) values (Figure 2). After the hygroscopic experiment, an SEM analysis of the sample was employed to primarily check particle types. This allowed us to further understand how PBAPs particles grow at different RH values ranging from 5% to 94%.

The scheme of the IPH system is shown in Figure 2, which consisted of four steps; (1) Introducing $N_2$ gas with a mass flow controller into a chamber; (2) Setting a TEM grid or silicon wafer on the bottom of an environmental microscopic cell (Gen-RH Mcell, UK), which can change the RH and maintain the temperature at 20 ℃; (3) Taking images at incremental RH values using an optical microscope (Olympus BX51M, Japan) with a camera (Canon 650D); (4) Obtaining through the RADUS software the PBAPs sizes (i.e., D(RH) and $D_0$) in the images taken from the optical microscopy manually or automatically.. The images can be taken at different RHs during hygroscopic experiments and then are input into the RADUS software for size measurement.

This IPH system has been tested and has successfully captured the hygroscopic growth of individual aerosol particles collected on either a silicon wafer or TEM grid in our laboratory (Sun et al., 2018). Before the IPH system is used for ambient samples, it must be checked through standard NaCl particles on a silicon wafer made in the laboratory. Figure S3 shows that the delinquence relitive humidity (DRH) of individual NaCl particles on this silicon wafer is at

76%, similar to the standard DRH at 75±1%. After the procedure, we can replace our collected samples into the IPH system.

The particle growth factor (GF), an important parameter used to describe the hygroscopic growth of individual particles, is defined as follows:

$$GF(RH)=\frac{D(RH)}{D_0}$$

where $D(RH)$ and $D_0$ are the diameters of particles at a given RH and at 5% RH, respectively.

[Figure]

**Figure 2** Scheme of a custom-made individual particle hygroscopic system to observe hygroscopic growth of individual particles

2.5 **Meteorological data and back trajectories**

Meteorological data, including the relative humidity (RH), temperature, wind speed, and wind direction, were measured and recorded every 5 min by an automated weather meter (Kestrel 5500, USA). During the sampling period, the relative humidity (RH) and temperature varied from 40-70% and 22-28 ℃ during the day and 90-100% and 10-15 ℃ during the night, respectively. The wind speed was 1.5-7.6 m s$^{-1}$ during the day and 0-1 m s$^{-1}$ at night (Figure

S4).

To determine the regional transport of air masses, 6-h back trajectories of air masses were generated using a Hybrid Single Particle Lagrangian Integrated Trajectory (HYSPLIT) model at the forest sampling station during 14-21 August, 2016. Based on the sampling times of each day at 09:00, 15:00, 21:00, and 02:00 (midnight) local time, we performed 31 air mass back trajectories. Here we selected an altitude of 500 m as the end point of each back trajectory (Figure 1). Figure 1 shows that all the back trajectories in the past 6-h had been transported over the Lesser Khingan Mountain forest.

**3.    Results and Discussion**

**3.1 Morphology and elemental composition of PBAPs**

Among the 4,122 analyzed aerosol particles with diameters of 100 nm-10 μm analyzed by

TEM/EDS, individual particles are classified into five groups based on their morphology and composition: S-OM (mixture of sulfate (S), organics (OM)), OM, mineral dust, and PBAPs (Figure

3). S can be used to indicate secondary sulfates; abundant C and minor O with transparent color constitute the coating of the sulfate core and represent secondary organic matter; and irregular particles containing Si, Al, Ca, minor Fe, and Ti normally indicate mineral dust particles.

Moreover, previous studies have found that elemental P in individual particles and their associated unique morphologies can be used to identify PBAPs by electron microscopy (Poschl,

2005;Wittmaack et al., 2005). Thirteen percent of particles were PBAPs, and low magnification

TEM and SEM images both revealed that abundant PBAPs occurred in the samples (e.g., Figure 3a- b).

The number fractions of size-resolved aerosol particles show that secondary S-OM and OM

particles were the dominant particle groups in the fine mode (< 1 μm) while PBAPs and mineral particles dominated the coarse mode (≥ 1 μm) (Figure 4a). Moreover, we noticed that the number fractions of PBAPs in each sample collected at night were much higher than those collected during the day. Abundant fine secondary sulfate and organic particles from photochemical formation were observed during the day. Figure 4b shows that the average number fraction of PBAPs was 2.5% in the samples collected during the day and as high as 30.0% at night. If we further calculated the number concentration of PBAPs in Figure 4b, the PBAPs concentration significantly increased by approximately seven times from daytime to nighttime, although the non-PBAPs concentration decreased.

Based on the morphology and size of the PBAPs, we definitely identified fungal spores and brochosomes, and plant or insect debris, all of which have been widely reported before (Wittmaack et al., 2005;Huffman et al., 2012;Afanou et al., 2014;Valsan et al., 2015;Priyamvada et al., 2017).

Besides these PBAPs, we also found many special rod-like PBAPs with a dominant size range of 1

- 5 μm. Pollen was not found in our samples, which may be because large pollen emissions occur in spring and early summer instead of late summer (August) in boreal forests (Manninen et al., 2014).

[Figure]

**Figure 3** Low magnification SEM and TEM images of individual particles collected from the forest air.

(a) low magnification SEM image of rod-like PBAPs (red arrows) and fungal spores (green); (b) low magnification TEM image of rod-like PBAPs particles and secondary sulfate (S-rich) particles; (c) SEM

image of a rod-like particle; (d) TEM image of a mineral dust particle (e) TEM image of an organic matter (OM) particle; and (f) TEM image of OM coating on S-rich particles. The color in (a) was artificially painted on the original SEM images.

[Figure]

**Figure 4** Number fractions of different types of particles in different size bins and their total number fraction (a); and number fractions of primary biological aerosol particles (PBAPs) and non-PBAPs during the day and night (b). The number of analyzed particles is listed above each column. D and N are daytime and nighttime.

Online instruments indicate that large number of fine PBAPs are bacteria and fungi in the forest air (Tong and Lighthart, 2000;Elbert et al., 2007;Huffman et al., 2010;Despr és et al., 2012;Hu et al.,

2018). Although many previous studies reported PBAPs through the SEM, no observations of fine bacteria and fungal particles in forest air were reported (Wittmaack et al., 2005;Shi et al.,

2009;Martin et al., 2010;Huffman et al., 2012;Tamer Vestlund et al., 2014;Valsan et al., 2015;Valsan et al., 2016;Priyamvada et al., 2017;Wu et al., 2019). Posfai et al. (2003) found a few rod-like bacterial particles in marine air using TEM. In this study, we found that the rod-like PBAPs (Figure

5a-e) have a morphology similar to bacteria reported by Posfai et al. (2003). These rod-like PBAPs were stable under the electron beam during the TEM analysis, and they contained C, N, O, P, and K

with minor Mg, Si, S, Ca and Fe (Figure 5a). These rod-like PBAPs have a size range of 300 nm-7

μm with the dominant size range of 1-5 μm with two typical peaks at 1.4 μm and 3.5 μm (Figure

6a). Figure 6b further shows that the aspect ratio of 85% of these particles is larger than 1.5.

In nature, many fine fungi normally displayed similar composition and rod-like shape. To better compare and confirm differences of bacteria and fungi observed in TEM, we cultured *Colibacillus*

and *Yeast* in the laboratory to represent bacteria and fungi. Then we sprayed the solution of

*Colibacillus* and *Yeast* onto TEM grids. After drying these samples, we observed the morphology and size of *Colibacillus* and *Yeast* through the TEM (Figure 5f-g and Figure S5). Indeed, TEM/EDS

show very similar rod-like shape and composition between *Colibacillus* and *Yeast* particles on the substrate, although the *Yeast* particles with a size range at 1-8 μm with a mean diameter at 4.3 μm are larger than *Colibacillus* (300 nm-2.5 μm with mean diameter of 1.3 μm) (Figure S5). It is interesting that the size distribution of the rod-like PBAPs collected in the forest air displays two typical peaks at 1.4 μm and 3.5 μm, which probably represent bacteria and fungi. Després et al.

(2012) stated that bacteria mostly have diameters of 1-2 μm and fungi of 1−10 μm in the atmosphere.

Although we can indicate the bacteria and fungi based on their sizes, the clue could not be used to precisely identify bacteria and fungi through electron microscopy due to their overlapped size range.

Figure 6b shows 85% of particles with larger aspect ratios (> 1.5), suggesting most of these PBAPs particles have typical rod-like shape. Although their identification is tentative, we called all these similar rod-like bacteria and fungal particles "rod-like PBAPs" here.

[Figure]

Figure 5 TEM image of the rod-like PBAPs collected in forest air and the fungi and bacteria cultivated in laboratory. (a) Morphology of a rod-like PBAP and EDS spectra of its core and main part. The red circles indicate where EDS impacted the rod-like PBAP. (b-e) Various rod-like PBAPs collected in forest air. (f) One *Yeast* particle cultivated in laboratory (e) One *colibacillus* particle cultivated in laboratory.

[Figure]

Figure 6 Size distribution and aspect ratios of rod-like PBAPs, fungal spores, and brochosomes collected in boreal forest air.

Fungal spores are microscopic biological particles that allow fungi to reproduce, serving a similar purpose to that of seeds in the plant world (Lacey and West, 2006). Spores can be released as a part of the sexual and/or asexual morph (stage) of the lifecycle of a fungus, and many species are able to produce spores from both stages (Despr és et al., 2012). Fungal spores have been reported in many places in the global air and their morphologies have been well documented (Shi et al.,

2003;Wittmaack et al., 2005;Coz et al., 2009;Shi et al., 2009;Martin et al., 2010;Huffman et al.,

2012;Tamer Vestlund et al., 2014;Afanou et al., 2015;Valsan et al., 2015;Valsan et al.,

2016;Priyamvada et al., 2017;Wu et al., 2019). In this study, the fungal spores generally appeared as ovoid (Figure 7a), sub-globular (Figure 7b-c) or elongated shapes with a smooth surface and small protuberances (apiculus) (Figures 8a-f). Figure 7d shows that their composition mainly consists of C, O and Si, followed by minor N, Mg, P, S, K and Fe. The size range of the observed fungal spores varied roughly between 400 nm and 7 $\mu$m (Figure 6a). The size distribution of fungal spores further showed a dominant size range of 2 - 5 $\mu$m and one peak at 4 $\mu$m. The number fraction of fungal spores at all aspect ratios is generally lower than 0.15, suggesting that there is no typical shape from either roundness or elongation for fungal spores in the boreal forest. SEM images clearly display that several typical fungal spores with diameters of 3.7-6.5 $\mu$m do not have well-defined shapes and that their surfaces have regular strips or regular protuberances (Figure 8). Similar fungal spores have been reported in forest air (Wittmaack et al., 2005;Valsan et al., 2015). Compared with the rod-like PBAPs, fungal spores normally have a rougher surface (Figures 6-7), larger size, and much higher Si and lower N. Therefore, the fungal spores can easily be identified based on their morphology among the PBAPs through the TEM and SEM analysis.

[Figure]

**Figure 7** TEM/EDS showing the morphology and composition of various fungal spores. (a) a spindle fungal spore; (b) a fungal spore with protuberances; (c) a fungal spore with protuberances; and (d) EDS spectrum showing the composition of fungal spore. The red circle indicates where EDS

impacted the particle.

[Figure]

**Figure 8** SEM images showing the shape, size, and surface properties of fungal spores. Size represents the diameter of fungal spores. (a-d) Surfaces of three spindle fungal particles with a layer of strips. (e-f) Surfaces of two fungal spores with protuberances.

Brochosomes are hollow spherical particles produced by leafhoppers (Cicadelliae)

(Wittmaack, 2005). TEM and SEM observations both found abundant brochosomes in the samples.

The low-magnification SEM images showed that there are large brochosomal clusters on the substrate, each containing tens or hundreds of single brochosomes (Figure S6). Wittmaack (2005)

found that most of the brochosomes normally occur as large clusters and reported that each cluster contains up to 100,000 brochosomes. In this study, TEM and SEM both produce clear images showing the structure of the brochosome (Figures 9-10). Interestingly, the outline of each brochosome approximates a truncated icosahedron and the brochosome particles likely have unique inner structures, such as C60 Buckminster fullerenes (Figures 9a-b and 10). Compared with the rod- like PBAPs, chemical composition of the brochosomal particles show extremely high Si and low P

in addition to major C and O and minor N, Na, S, K and Fe (Figure 9c). A single brochosome has a size range of 200-700 nm with a mean diameter of 350 nm. The aspect ratio of individual brochosomes is close to 1, suggesting that they are spherical (Figure 6b). Because the brochosomes might be dispersed from their clusters when they impact on the substrate,, it is not meaningful to compare the number fraction of brochosomes with the rod-like PBAPs and fungal spores.

[Figure]

**Figure 9** TEM images of brochosomes and the composition of (a) a single brochosome and brochosome aggregations; (b) high-resolution TEM image showing the inner structure of one brochosome; (c) EDS

spectrum showing the chemical composition of the brochosomes.

[Figure]

**Figure 10** SEM images of brochosomes. (a) Single brochosome and their aggregations. Some brochosomal particles are associated with primary biological species. (b) High-resolution SEM image showing the surface properties of the brochosomal particles.

The TEM and SEM images both show a few elongated large particles at 8-20 μm among the biological particles. EDS shows that these particles mainly contained C, O, and Si but no detectable

P in some of these biological particles as shown in Figures 11-12. We speculate that these biological particles were plant or insect debris. For example, Wittmaack et al. (2005) suggested that the spaghetti-type biological particles from Figure 11a-d are likely epicuticlar wax fragments of plants.

The SEM images as shown in Figure 12 clearly displayed the surface morphology of the large particles.

[Figure]

**Figure 11** TEM images showing the morphology of the primary biological particles. (a) One elongated particle with thorns; (b) one circular particle; (c-d) two elongated particles; and (e) one long spindle particle

[Figure]

size: 11.4 µm    size: 8.3 µm    size: 10.1 µm

**Figure 12** SEM image showing the morphology and surface properties of three elongated biological particles.

**3.2  Relative abundance of PBAPs**

In this study, we classified PBAPs but also efficiently obtained the number fraction of rod-like

PBAPs and fungal spores in coarse mode particles (> 1 μm). The results from the electron microscopy analysis further estimated that PBAPs, mineral dust, and the remaining particles accounted for 50%, 25%, and 25% of the coarse mode, respectively. Assuming a density of ~1 g cm$^{-3}$ for PBAPs (Elbert et al., 2007), 2 g cm$^{-3}$ for mineral dust particles, and 1.4 g cm$^{-3}$ for the remaining particles (e.g., S-OM, OM, and metal) (Rissler et al., 2006), mass concentrations of the three different types of particles with different size bins can be estimated based on the equation:

$$M_i = \frac{\Pi}{6} D_i{}^3 \rho_i N_i$$

*i*: particle type (PBAPs, mineral dust, and other remaining particle)

D: particle geometrical diameter in a size bin

N: particle number in a size bin

M: total mass of the analyzed particles in a size bin

ρ: particle density (g cm$^{-3}$)

In the equation, $N_i$ and $D_i$ both can be obtained through the measurement of individual particles in TEM images. Finally, we estimated that the mass concentration of PBAPs, mineral dust, and remaining particles accounted for 47%, 43%, and 10% of PM$_{2.5-10}$, respectively. The results suggest that PBAPs significantly contributed to mass concentration of PM$_{2.5-10}$ in summertime in the boreal forest air. During the sampling period, we measured the daily mass concentrations of PM$_{2.5}$ of ~6.0

μg m$^{-3}$ and PM$_{10}$ of ~10.0 μg m$^{-3}$. The number size distribution of PBAPs coupled with the mass concentrations of PM$_{2.5}$ and PM$_{10}$ were used to estimate the total mass concentration of PBAPs using the result from the above equation. We estimated that the PBAPs contributed ~1.9 μg m$^{-3}$ to the concentration of PM$_{2.5-10}$ of 4.0 μg m$^{-3}$.

Thirteen percent of all detected particles by number collected from the boreal forest air are

PBAPs. Such a high fraction of PBAPs has not been reported in urban and rural air in China (Shi et al., 2003;Shi et al., 2009;Li et al., 2016). We noticed that the number concentration of PBAPs was much higher at night than during the day (Figure 3b). A shallow nocturnal boundary layer can lead to a slight increase in the number concentration of coarse particles near the ground (Graham et al.,

2003), but this increase cannot explain the large difference in the relative number fraction of PBAPs (12 times larger at night than during the day) (Figure 3b). Alternately, the relative emission strength of PBAPs from the forest between day and night likely induced the difference of the relative number fractions.

It is well documented that meteorological conditions such as RH, wind speed, and temperature can affect PBAPs emission in the forests (Harrison et al., 2005;Whitehead et al., 2016). In particular, the wind speed is especially important in promoting PBAPs emission into air. During the sampling period, the average wind speeds at 5 min intervals had a range from 0 to 7.5 m/s with a mean value of 0.75 m/s. 89% of the measured wind speeds were lower than 2 m/s (Figure S4). Therefore, we conclude that no large consistent wind speeds occurred during the sampling period. Furthermore, we compared all the air mass back trajectories in the past 6-h over the Lesser Khingan Mountain forest at each sampling time (Figure 1). There are similar lengths of these back trajectories, suggesting that wind speeds above the forest canopy had only small changes during the sampling period. Therefore, the result from the ground-based measurements of wind speeds is consistent with air mass back trajectories. Here, we can exclude wind speeds during the sampling period as one important factor to dominate PBAPs emissions during day and night in the boreal forest. High temperatures normally increase the PBAPs emissions from the plants in the daytime (Harrison et al., 2005). However, we observed contrasting results that more PBAPs occurred in nighttime instead of daytime (Figure S4). Therefore, we also exclude temperatures during the sampling period as a cause of the vastly different PBAPs emissions at day and night in the boreal forest.

Besides wind speed and temperature, RH is an important meteorological variable that influences PBAPs emissions from plants (Harrison et al., 2005;Huffman et al., 2012). In this study, we found large differences of RH between day and night (Figure S4). The elevated RH near 100%

at night (Figure S1) appears to be an important factor that increases the emissions of PBAPs. This result is consistent with the conclusion of Elbert et al. (2007), who showed that PBAPs in a boreal forest are generally most abundant in samples collected at night when the RH is close to 100%. A

similar phenomenon has been  observed in different forests, such as the Amazon rainforest (Huffman et al., 2012;Whitehead et al., 2016), a montane ponderosa pine forest in North American (Crawford et al., 2014), a semi-arid forest in the southern Rocky Mountains of Colorado (Gosselin et al., 2016), and a semi-rural site in southwestern Germany (Toprak and Schnaiter, 2013). These studies above found that a nighttime peak of number concentrations of fluorescent biological aerosol particles is consistent with nocturnal sporulation driven by the increased RH. Moreover, Troutt and

Levetin (2001) explained that the increase in PBAP concentrations is caused by the increase in basidiospores concentrations with RH, and they showed that a clear diurnal rhythm occurs and peaks at 04:00-06:00 LT. Furthermore, the number ratio (4.6 at nighttime and 4.0 at daytime) of rod-like

PBAPs vs fungal spores and their number concentrations increased from daytime to nighttime (Figure S7). These results all suggest that higher RH can promote the emissions of rod-like PBAPs and fungal spores in the boreal forest.

**3.3 Mixing state of rod-like PBAPs**

Our study shows that rod-like PBAPs contain bacteria and fungi in the boreal forest air.

Although approximately 80% of rod-like PBAPs were externally mixed particles in the boreal forest air, we still found that 20% of rod-like PBAPs were internally mixed particles. TEM observations show that the rod-like PBAPs were frequently internally mixed with mineral, metal, organics, and inorganic salts. We noticed that irregular mineral dust particles significantly changed the shape of the rod-like PBAPs (Figure 13a-c). The EDS analysis shows that the internally mixed mineral particles contain certain amounts of C, O, and P in addition to Si, Al, or Ca (Figure 13a-c), suggesting that many rod-like PBAPs were associated with mineral dust particles.

In this study, we found that some nanoscale metal particles were internally mixed with rod-like

PBAPs. Figure 13d-f further shows that these metals were spherical and contained Mn, Si and/or

Fe. As in previous studies, these nanosize metal particles were emitted from industrial activities or power plants instead of natural soil (Li et al., 2017). TEM observations show that these metallic particles were mainly attached to the surface of rod-like PBAPs. Moreover, some rod-like PBAPs were coated by inorganic salts (e.g., K-P in Figure 13g and S-rich in Figure 13i) and organics. The shape of the rod-like PBAPs might change following the aging process during long-range transport (Figure 13), although the elemental P or its associated ionic components ($H_2PO_4^-$ and $PO_3^-$) did not change (Pratt et al., 2009). Pratt et al. (2009) detected $H_2PO_4^-$ and $PO_3^-$ in individual cloud ice- crystal residues to identify PBAPs using aerosol time-of-flight mass spectrometry. Although one study indicates that a few mineral dust or fly ash particles contain trace inorganic P, these particles do not contain abundant organics and their number is low in the air (Zawadowicz et al., 2017).

Therefore, TEM/EDS is an efficient tool to identify fine bacteria or fungi from non-PBAPs collected in the atmosphere. Moreover, it significantly reveals the mixing state of individual PBAPs, a key to understand their possible CCN and IN activity over the boreal forest air in the future.

[Figure]

**Figure 13** TEM showing the internally mixed rod-like PBAPs. (a-c) Internal mixture of mineral and rod- like PBAP; (d-f) Internal mixture of metal and rod-like PBAP; (g) Internal mixture of inorganic salts and rod-like PBAP; (h) Internal mixture of organics and rod-like PBAP; and (i) Internal mixture of S-rich salts and rod-like PBAP.

**3.4 Hygroscopicity of PBAPs**

In this study, we conducted an experiment to observe the hygroscopic growth of fresh PBAPs.

In the hygroscopic experiment, the PBAPs all take up water and grow by up to 88% during hydration, and they lose water and return to the dry particle size (reduction of 83%) during dehydration (Figure

14). The growth factor of the PBAPs is ~1.09 at RH=94% based on the particle diameter change, which is much lower than growth factor of NaCl at ~2.3 (Figure 14). These results show that the fresh PBAPs have extremely weak hygroscopicity.

Recent studies found that fungal fragments collected in Amazon forests displayed strong hygroscopic properties (China et al., 2016;China et al., 2018) and were internally mixed with certain amounts of sodium salts. However, we found weak hygroscopic growth of 1.09, whereas this value was in the range of 1.05-1.3 for bacteria and fungal spores in previous studies (Reponen et al.,

1996;Lee et al., 2002). However, the result is much lower than the value of 2.30 at RH=94% for

NaCl (Figure 2a) and 1.60 at RH 94% for ammonium sulfate (Sun et al., 2018). This comparison suggests that fresh PBAPs display extremely weak hygroscopicity and do not contain any sodium salt in the boreal forest (Figure 2a). Overall, our results indicate that PBAPs from the substantial biological emissions from the Khingan Mountain boreal forest are weakly hygroscopic in nature.

[Figure]

**Figure 14** Hygroscopic growth of NaCl prepared in laboratory and primary biological particles collected in boreal forest air. The up arrows (i.e., RH) represent hydration.

**4. Conclusions**

The TEM and SEM observations both showed that the morphology of PBAPs were unique; they differed markedly from that of the sulfate, mineral, soot, organics, and metal particles in continental air. Our results indicate that significant amounts of PBAPs are emitted from the Khingan

Mountain area. In this study, we establish detailed information that includes the morphology, size, and composition of rod-like PBAPs, fungal spores, and brochosomes. C, N, O, P, K, and Si were detected in most of the PBAPs, and P represented a major marker to discriminate the PBAPs and non-PBAPs. We found that one type of PBAPs mostly appeared as similar rod-like shapes with an aspect ratio > 1.5 and the dominant sizes ranged from 1 μm to 5 μm. The size distribution of the rod-like PBAPs displays two typical peaks at 1.4 μm and 3.5 μm, which likely represent bacteria and fungal particles in the forest air. However, our study shows that there was no clear boundary between bacteria and some fungi from their size because of their size range partly overlapped.

The second most plentiful PBAPs were identified as fungal spores with ovoid, sub-globular or elongated shapes with a smooth surface and small protuberances (apiculus) with size at 400 nm - 7

μm with a mean diameter of 4 μm. Moreover, we found some large brochosomal clusters containing hundreds of brochosomes which have sizes from 200-700 nm and shapes like truncated icosahedrons. We estimated that the mass concentration of PBAPs, mineral dust, and remaining particles accounted for 47%, 43%, and 10% of the $PM_{2.5-10}$ mass concentration, respectively, indicating that large boreal forests might represent a major source of PBAPs in the atmosphere.

Moreover, there is a higher frequency and concentration of PBAPs at night compared with day. This difference could not be explained by wind speed or temperature, but was explicable by RH, which appears to be critical in enhancing PBAPs emissions from plants at night. The hygroscopic experiment shows that the primary bacterial and fungal particles show weak hygroscopicity.

**Author Contributions:** WL designed the study. WL, LL, QY, LX, and HY collected aerosol particles. WL, QL, LL, LX, YZ, BW, XD, and JZ contributed laboratory experiments and data analysis. WL prepared the manuscript with contributions from all the coauthors. BW, DH, DL, WH, DZ, PF, MY, MH, XZ, and ZS commented and edited the paper.

**Competing interests:** The authors declare no competing financial interests

**Acknowledgments**

We appreciate Peter Hyde's comments and proofreading. We appreciate Min Yue and Jiachang Lian (Zhejiang Univ.) providing the *Yeast* and *Colibacillus* cultures. Huijun Xie (Shandong Univ.) is thanked for discussions. This work was funded by the National Natural Science Foundation of China (9184430003, 41622504 and 41575116), Zhejiang Provincial Natural Science Foundation of China (LZ19D050001), and Zhejiang University Education Foundation Global Partnership Fund. All the data are presented in the paper.

**References:**

Afanou, K. A., Straumfors, A., Skogstad, A., Nilsen, T., Synnes, O., Skaar, I., Hjeljord, L., Tronsmo, A., Green, B. J., and Eduard, W.: Submicronic Fungal Bioaerosols: High-Resolution Microscopic Characterization and Quantification, Appl. Environ. Microb., 80, 7122-7130, 10.1128/aem.01740-14, 2014.

Afanou, K. A., Straumfors, A., Skogstad, A., Skaar, I., Hjeljord, L., Skare, Ø., Green, B. J., Tronsmo, A., and Eduard, W.: Profile and Morphology of Fungal Aerosols Characterized by Field Emission Scanning Electron Microscopy (FESEM), Aerosol Sci. and Tech., 49, 423-435, 10.1080/02786826.2015.1040486, 2015.

Amato, P., Ménager, M., Sancelme, M., Laj, P., Mailhot, G., and Delort, A.-M.: Microbial population in cloud water at the Puy de Dôme: Implications for the chemistry of clouds, Atmos. Environ., 39, 4143-4153, https://doi.org/10.1016/j.atmosenv.2005.04.002, 2005.

Ault, A. P., and Axson, J. L.: Atmospheric Aerosol Chemistry: Spectroscopic and Microscopic Advances, Anal. Chem., 89, 430-452, 10.1021/acs.analchem.6b04670, 2017.

Bauer, H., Schueller, E., Weinke, G., Berger, A., Hitzenberger, R., Marr, I. L., and Puxbaum, H.: Significant contributions of fungal spores to the organic carbon and to the aerosol mass balance of the urban atmospheric aerosol, Atmos. Environ., 42, 5542-5549, https://doi.org/10.1016/j.atmosenv.2008.03.019, 2008.

Chen, H., and Yao, M.: A high-flow portable biological aerosol trap (HighBioTrap) for rapid microbial detection, J. Aerosol Sci., 117, 212-223, https://doi.org/10.1016/j.jaerosci.2017.11.012, 2018.

China, S., Wang, B., Weis, J., Rizzo, L., Brito, J., Cirino, G. G., Kovarik, L., Artaxo, P., Gilles, M. K., and Laskin, A.: Rupturing of Biological Spores As a Source of Secondary Particles in Amazonia, Environ. Sci. Technol., 50, 12179-12186, 10.1021/acs.est.6b02896, 2016.

China, S., Burrows, S. M., Wang, B., Harder, T. H., Weis, J., Tanarhte, M., Rizzo, L. V., Brito, J., Cirino, G. G., Ma, P.-L., Cliff, J., Artaxo, P., Gilles, M. K., and Laskin, A.: Fungal spores as a source of sodium salt particles in the Amazon basin, Nat. Commun., 9, 4793, 10.1038/s41467-018-07066-4, 2018.

Christner, B. C., Cai, R., Morris, C. E., McCarter, K. S., Foreman, C. M., Skidmore, M. L., Montross, S. N., and Sands, D. C.: Geographic, Seasonal, and Precipitation Chemistry Influence on the Abundance and Activity of Biological Ice Nucleators in Rain and Snow, P. Natl. Acad. Sci., 105, 18854-18859, 2008.

Coz, E., Gómez-Moreno, F. J., Pujadas, M., Casuccio, G. S., Lersch, T. L., and Artíñano, B.: Individual particle characteristics of North African dust under different long-range transport scenarios, Atmos. Environ., 43, 1850-1863, 10.1016/j.atmosenv.2008.12.045, 2009.

Coz, E., Artíñano, B., Clark, L. M., Hernandez, M., Robinson, A. L., Casuccio, G. S., Lersch, T. L., and Pandis, S. N.: Characterization of fine primary biogenic organic aerosol in an urban area in the northeastern United States, Atmos. Environ., 44, 3952-3962, https://doi.org/10.1016/j.atmosenv.2010.07.007, 2010.

Crawford, I., Robinson, N. H., Flynn, M. J., Foot, V. E., Gallagher, M. W., Huffman, J. A., Stanley, W. R., and Kaye, P. H.: Characterisation of bioaerosol emissions from a Colorado pine forest: results from the BEACHON-RoMBAS experiment, Atmos. Chem. Phys., 14, 8559-8578, 10.5194/acp-14-8559-2014, 2014.

Crawford, I., Ruske, S., Topping, D. O., and Gallagher, M. W.: Evaluation of hierarchical agglomerative cluster analysis methods for discrimination of primary biological aerosol, Atmos. Meas. Tech., 8, 4979-4991, 10.5194/amt-8-4979-2015, 2015.

Després, V., Huffman, J. A., Burrows, S. M., Hoose, C., Safatov, A., Buryak, G., Fröhlich-Nowoisky, J.,

Elbert, W., Andreae, M., Pöschl, U., and Jaenicke, R.: Primary biological aerosol particles in the atmosphere: a review, Tellus B, 64, 15598, 10.3402/tellusb.v64i0.15598, 2012.

Elbert, W., Taylor, P. E., Andreae, M. O., and Pöschl, U.: Contribution of fungi to primary biogenic aerosols in the atmosphere: wet and dry discharged spores, carbohydrates, and inorganic ions, Atmos. Chem. Phys., 7, 4569-4588, 10.5194/acp-7-4569-2007, 2007.

Fröhlich-Nowoisky, J., Pickersgill, D. A., Després, V. R., and Pöschl, U.: High diversity of fungi in air particulate matter, P. Natl. Acad. Sci., 106, 12814-12819, 10.1073/pnas.0811003106, 2009.

Gabey, A. M., Gallagher, M. W., Whitehead, J., Dorsey, J. R., Kaye, P. H., and Stanley, W. R.: Measurements and comparison of primary biological aerosol above and below a tropical forest canopy using a dual channel fluorescence spectrometer, Atmos. Chem. Phys., 10, 4453-4466, 10.5194/acp-10-4453-2010, 2010.

Georgakopoulos, D. G., Després, V., Fröhlich-Nowoisky, J., Psenner, R., Ariya, P. A., Pósfai, M., Ahern, H. E., Moffett, B. F., and Hill, T. C. J.: Microbiology and atmospheric processes: biological, physical and chemical characterization of aerosol particles, Biogeosciences, 6, 721-737, 10.5194/bg-6-721-2009, 2009.

Gosselin, M. I., Rathnayake, C. M., Crawford, I., Pöhlker, C., Fröhlich-Nowoisky, J., Schmer, B., Després, V. R., Engling, G., Gallagher, M., Stone, E., Pöschl, U., and Huffman, J. A.: Fluorescent bioaerosol particle, molecular tracer, and fungal spore concentrations during dry and rainy periods in a semi-arid forest, Atmos. Chem. Phys., 16, 15165-15184, 10.5194/acp-16-15165-2016, 2016.

Graham, B., Guyon, P., Maenhaut, W., Taylor, P. E., Ebert, M., Matthias-Maser, S., Mayol-Bracero, O. L., Godoi, R. H. M., Artaxo, P., Meixner, F. X., Moura, M. A. L., Rocha, C. H. E. D. A., Grieken, R. V., Glovsky, M. M., Flagan, R. C., and Andreae, M. O.: Composition and diurnal variability of the natural Amazonian aerosol, J. Geophy. Res., 108, 4765, doi:10.1029/2003JD004049, 2003.

Harrison, R. M., Jones, A. M., Biggins, P. D. E., Pomeroy, N., Cox, C. S., Kidd, S. P., Hobman, J. L., Brown, N. L., and Beswick, A.: Climate factors influencing bacterial count in background air samples, Int. J. Biometeorol. 49, 167-178, 10.1007/s00484-004-0225-3, 2005.

Hu, W., Murata, K., Fukuyama, S., Kawai, Y., Oka, E., Uematsu, M., and Zhang, D.: Concentration and Viability of Airborne Bacteria Over the Kuroshio Extension Region in the Northwestern Pacific Ocean: Data From Three Cruises, J. Geophy. Res., 122, 12,892-812,905, 10.1002/2017JD027287, 2017.

Hu, W., Niu, H., Murata, K., Wu, Z., Hu, M., Kojima, T., and Zhang, D.: Bacteria in atmospheric waters: Detection, characteristics and implications, Atmos. Environ., 179, 201-221, https://doi.org/10.1016/j.atmosenv.2018.02.026, 2018.

Huffman, J. A., Treutlein, B., and Pöschl, U.: Fluorescent biological aerosol particle concentrations and size distributions measured with an Ultraviolet Aerodynamic Particle Sizer (UV-APS) in Central Europe, Atmos. Chem. Phys., 10, 3215-3233, 10.5194/acp-10-3215-2010, 2010.

Huffman, J. A., Sinha, B., Garland, R. M., Snee-Pollmann, A., Gunthe, S. S., Artaxo, P., Martin, S. T., Andreae, M. O., and Pöschl, U.: Size distributions and temporal variations of biological aerosol particles in the Amazon rainforest characterized by microscopy and real-time UV-APS fluorescence techniques during AMAZE-08, Atmos. Chem. Phys., 12, 11997-12019, 10.5194/acp-12-11997-2012, 2012.

Huffman, J. A., Prenni, A. J., DeMott, P. J., Pöhlker, C., Mason, R. H., Robinson, N. H., Fröhlich-Nowoisky, J., Tobo, Y., Després, V. R., Garcia, E., Gochis, D. J., Harris, E., Müller-Germann, I., Ruzene, C., Schmer, B., Sinha, B., Day, D. A., Andreae, M. O., Jimenez, J. L., Gallagher, M., Kreidenweis, S. M., Bertram, A. K., and Pöschl, U.: High concentrations of biological aerosol particles and ice nuclei during and after rain, Atmos. Chem. Phys., 13, 6151-6164, 10.5194/acp-13-6151-2013, 2013.

Lacey, M. E., and West, J. S.: The Air Spora: A Manual for Catching and Identifying Airborne Biological

Particles, Springer: Dordrecht, The Netherlands, 2006.

Lee, B. U., Kim, S. H., and Kim, S. S.: Hygroscopic growth of E. coli and B. subtilis bioaerosols, J.

Aerosol Sci., 33, 1721-1723, https://doi.org/10.1016/S0021-8502(02)00114-3, 2002.

Li, W., Shao, L., Zhang, D., Ro, C.-U., Hu, M., Bi, X., Geng, H., Matsuki, A., Niu, H., and Chen, J.: A

review of single aerosol particle studies in the atmosphere of East Asia: morphology, mixing state, source, and heterogeneous reactions, J. Clean. Prod., 112, Part 2, 1330-1349, 2016.

Li, W., Xu, L., Liu, X., Zhang, J., Lin, Y., Yao, X., Gao, H., Zhang, D., Chen, J., Wang, W., Harrison, R.

M., Zhang, X., Shao, L., Fu, P., Nenes, A., and Shi, Z.: Air pollution–aerosol interactions produce more bioavailable iron for ocean ecosystems, Sci. Adv., 3, e1601749, 2017.

Ling, M. L., Wex, H., Grawe, S., Jakobsson, J., Löndahl, J., Hartmann, S., Finster, K., Boesen, T., and

Šantl-Temkiv, T.: Effects of Ice Nucleation Protein Repeat Number and Oligomerization Level on Ice

Nucleation Activity, J. Geophy. Res., 123, 1802-1810, 10.1002/2017JD027307, 2018.

Möhler, O., DeMott, P. J., Vali, G., and Levin, Z.: Microbiology and atmospheric processes: the role of biological particles in cloud physics, Biogeosciences, 4, 1059-1071, 10.5194/bg-4-1059-2007, 2007.

Manninen, H. E., Back, J., Sihto-Nissila, S. L., Huffman, J. A., Pessi, A. M., Hiltunen, V., Aalto, P. P.,

Hidalgo, P. J., Hari, P., Saarto, A., Kulmala, M., and Petaja, T.: Patterns in airborne pollen and other primary biological aerosol particles (PBAP), and their contribution to aerosol mass and number in a boreal forest, Boreal Environ. Res,, 19, 383-405, 2014.

Martin, S. T., Andreae, M. O., Artaxo, P., Baumgardner, D., Chen, Q., Goldstein, A. H., Guenther, A.,

Heald, C. L., Mayol-Bracero, O. L., McMurry, P. H., Pauliquevis, T., Pochl, U., Prather, K. A., Roberts,

G. C., Saleska, S. R., Silva Dias, M. A., Spracklen, D. V., Swietlicki, E., and Trebs, I.: Sources and properties of Amazonian aerosol particles, Rev. Geophys., 48, RG2002, 10.1029/2008rg000280, 2010.

May, N. W., Olson, N. E., Panas, M., Axson, J. L., Tirella, P. S., Kirpes, R. M., Craig, R. L., Gunsch, M.

J., China, S., Laskin, A., Ault, A. P., and Pratt, K. A.: Aerosol Emissions from Great Lakes Harmful

Algal Blooms, Environ. Sci. Technol., 52, 397-405, 10.1021/acs.est.7b03609, 2018.

Morris, C. E., Georgakopoulos, D. G., and Sands, D. C.: Ice nucleation active bacteria and their potential role in precipitation, J. Phys. IV France, 121, 87-103, 2004.

Morris, C. E., Conen, F., Alex Huffman, J., Phillips, V., Pöschl, U., and Sands, D. C.: Bioprecipitation: a feedback cycle linking Earth history, ecosystem dynamics and land use through biological ice nucleators in the atmosphere, Global Change Biol., 20, 341-351, 10.1111/gcb.12447, 2014.

Nikkels, A. H., Terstegge, P., and Spieksma, F. T. M.: Ten types of microscopically identifiable airborne fungal spores at Leiden, The Netherlands, Aerobiologia, 12, 107, 10.1007/bf02446602, 1996.

Patterson, J. P., Collins, D. B., Michaud, J. M., Axson, J. L., Sultana, C. M., Moser, T., Dommer, A. C.,

Conner, J., Grassian, V. H., Stokes, M. D., Deane, G. B., Evans, J. E., Burkart, M. D., Prather, K. A., and

Gianneschi, N. C.: Sea Spray Aerosol Structure and Composition Using Cryogenic Transmission

Electron Microscopy, ACS Central Sci., 2, 40-47, 10.1021/acscentsci.5b00344, 2016.

Poschl, U.: Atmospheric aerosols: Composition, transformation, climate and health effects, Angew.

Chem. Int. Edit., 44, 7520-7540, 10.1002/anie.200501122, 2005.

Posfai, M., Li, J., Anderson, J. R., and Buseck, P. R.: Aerosol bacteria over the southern ocean during

ACE-1, Atmos. Res., 66, 231-240, 2003.

Pratt, K. A., DeMott, P. J., French, J. R., Wang, Z., Westphal, D. L., Heymsfield, A. J., Twohy, C. H.,

Prenni, A. J., and Prather, K. A.: In situ detection of biological particles in cloud ice-crystals, Nat. Geosci.,

2, 398-401, 2009.

Prenni, A. J., Petters, M. D., Kreidenweis, S. M., Heald, C. L., Martin, S. T., Artaxo, P., Garland, R. M.,
Wollny, A. G., and Poschl, U.: Relative roles of biogenic emissions and Saharan dust as ice nuclei in the
Amazon basin, Nat. Geosci., 2, 402-405, 2009.

Priyamvada, H., Singh, R. K., Akila, M., Ravikrishna, R., Verma, R. S., and Gunthe, S. S.: Seasonal
variation of the dominant allergenic fungal aerosols – One year study from southern Indian region,
Scientific Reports, 7, 11171, 10.1038/s41598-017-11727-7, 2017.

Reponen, T., Willeke, K., Ulevicius, V., Reponen, A., and Grinshpun, S. A.: Effect of relative humidity
on the aerodynamic diameter and respiratory deposition of fungal spores, Atmos. Environ., 30, 3967-
3974, https://doi.org/10.1016/1352-2310(96)00128-8, 1996.

Riemer, N., Ault, A. P., West, M., Craig, R. L., and Curtis, J. H.: Aerosol Mixing State: Measurements,
Modeling, and Impacts, Rev. Geophys., 57, https://doi.org/10.1029/2018RG000615,
10.1029/2018rg000615, 2019.

Rissler, J., Vestin, A., Swietlicki, E., Fisch, G., Zhou, J., Artaxo, P., and Andreae, M. O.: Size distribution
and hygroscopic properties of aerosol particles from dry-season biomass burning in Amazonia, Atmos.
Chem. Phys., 6, 471-491, 10.5194/acp-6-471-2006, 2006.

Shi, Z., Shao, L., Jones, T. P., Whittaker, A. G., Lu, S., Berube, K. A., He, T., and Richards, R. J.:
Characterization of airborne individual particles collected in an urban area, a satellite city and a clean air
area in Beijing, 2001, Atmos. Environ., 37, 4097-4108, 2003.

Shi, Z., He, K., Xue, Z., Yang, F., Chen, Y., Ma, Y., and Luo, J.: Properties of individual aerosol particles
and their relation to air mass origins in a south China coastal city, J. Geophy. Res., 114,
doi:10.1029/2008JD011221, 2009.

Smith, D. J., Ravichandar, J. D., Jain, S., Griffin, D. W., Yu, H., Tan, Q., Thissen, J., Lusby, T., Nicoll,
P., Shedler, S., Martinez, P., Osorio, A., Lechniak, J., Choi, S., Sabino, K., Iverson, K., Chan, L., Jaing,
C., and McGrath, J.: Airborne Bacteria in Earth's Lower Stratosphere Resemble Taxa Detected in the
Troposphere: Results From a New NASA Aircraft Bioaerosol Collector (ABC), Front. Microbiol. 9,
10.3389/fmicb.2018.01752, 2018.

Spracklen, D. V., Bonn, B., and Carslaw, K. S.: Boreal forests, aerosols and the impacts on clouds and
climate, Philos. T. R. Soc. A., 366, 4613-4626, 10.1098/rsta.2008.0201, 2008.

Sun, J., Liu, L., Xu, L., Wang, Y., Wu, Z., Hu, M., Shi, Z., Li, Y., Zhang, X., Chen, J., and Li, W.: Key
Role of Nitrate in Phase Transitions of Urban Particles: Implications of Important Reactive Surfaces for
Secondary Aerosol Formation, J. Geophy. Res., 123, 1234-1243, 10.1002/2017JD027264, 2018.

Tamer Vestlund, A., Al-Ashaab, R., Tyrrel, S. F., Longhurst, P. J., Pollard, S. J. T., and Drew, G. H.:
Morphological classification of bioaerosols from composting using scanning electron microscopy, Waste
Manage., 34, 1101-1108, https://doi.org/10.1016/j.wasman.2014.01.021, 2014.

Therkorn, J., Thomas, N., Scheinbeim, J., and Mainelis, G.: Field performance of a novel passive
bioaerosol sampler using polarized ferroelectric polymer films, Aerosol Sci. Tech., 51, 787-800,
10.1080/02786826.2017.1316830, 2017.

Tobo, Y., Prenni, A. J., DeMott, P. J., Huffman, J. A., McCluskey, C. S., Tian, G., Pöhlker, C., Pöschl, U.,
and Kreidenweis, S. M.: Biological aerosol particles as a key determinant of ice nuclei populations in a
forest ecosystem, J. Geophy. Res., 118, 10,100-110,110, 10.1002/jgrd.50801, 2013.

Tong, Y., and Lighthart, B.: The Annual Bacterial Particle Concentration and Size Distribution in the
Ambient Atmosphere in a Rural Area of the Willamette Valley, Oregon, Aerosol Sci. Techn., 32, 393-
403, 10.1080/027868200303533, 2000.

Toprak, E., and Schnaiter, M.: Fluorescent biological aerosol particles measured with the Waveband

Integrated Bioaerosol Sensor WIBS-4: laboratory tests combined with a one year field study, Atmos.
Chem. Phys., 13, 225-243, 10.5194/acp-13-225-2013, 2013.

Troutt, C., and Levetin, E.: Correlation of spring spore concentrations and meteorological conditions in
Tulsa, Oklahoma, Int. J. Biometeorol., 45, 64-74, 10.1007/s004840100087, 2001.

Tunved, P., Hansson, H.-C., Kerminen, V.-M., Ström, J., Maso, M. D., Lihavainen, H., Viisanen, Y.,
Aalto, P. P., Komppula, M., and Kulmala, M.: High Natural Aerosol Loading over Boreal Forests, Science,
312, 261-263, 10.1126/science.1123052, 2006.

Twohy, C. H., McMeeking, G. R., DeMott, P. J., McCluskey, C. S., Hill, T. C. J., Burrows, S. M., Kulkarni,
G. R., Tanarhte, M., Kafle, D. N., and Toohey, D. W.: Abundance of fluorescent biological aerosol
particles at temperatures conducive to the formation of mixed-phase and cirrus clouds, Atmos. Chem.
Phys., 16, 8205-8225, 10.5194/acp-16-8205-2016, 2016.

Valsan, A. E., Priyamvada, H., Ravikrishna, R., Després, V. R., Biju, C. V., Sahu, L. K., Kumar, A.,
Verma, R. S., Philip, L., and Gunthe, S. S.: Morphological characteristics of bioaerosols from contrasting
locations    in    southern    tropical    India   –   A    case    study,    Atmos.    Environ.,   122,    321-331,
https://doi.org/10.1016/j.atmosenv.2015.09.071, 2015.

Valsan, A. E., Ravikrishna, R., Biju, C. V., Pöhlker, C., Després, V. R., Huffman, J. A., Pöschl, U., and
Gunthe, S. S.: Fluorescent biological aerosol particle measurements at a tropical high-altitude site in
southern India during the southwest monsoon season, Atmos. Chem. Phys., 16, 9805-9830, 10.5194/acp-
16-9805-2016, 2016.

Whitehead, J. D., Darbyshire, E., Brito, J., Barbosa, H. M. J., Crawford, I., Stern, R., Gallagher, M. W.,
Kaye, P. H., Allan, J. D., Coe, H., Artaxo, P., and McFiggans, G.: Biogenic cloud nuclei in the central
Amazon during the transition from wet to dry season, Atmos. Chem. Phys., 16, 9727-9743, 10.5194/acp-
16-9727-2016, 2016.

Wilson, T. W., Ladino, L. A., Alpert, P. A., Breckels, M. N., Brooks, I. M., Browse, J., Burrows, S. M.,
Carslaw, K. S., Huffman, J. A., Judd, C., Kilthau, W. P., Mason, R. H., McFiggans, G., Miller, L. A.,
Nájera, J. J., Polishchuk, E., Rae, S., Schiller, C. L., Si, M., Temprado, J. V., Whale, T. F., Wong, J. P. S.,
Wurl, O., Yakobi-Hancock, J. D., Abbatt, J. P. D., Aller, J. Y., Bertram, A. K., Knopf, D. A., and Murray,
B. J.: A marine biogenic source of atmospheric ice-nucleating particles, Nature, 525, 234,
10.1038/nature14986, 2015.

Wittmaack, K.: Brochosomes produced by leafhoppers-a widely unknown, yet highly abundant species
of    bioaerosols    in    ambient    air,    Atmos.    Environ.,    39,    1173-1180,
https://doi.org/10.1016/j.atmosenv.2004.11.003, 2005.

Wittmaack, K., Wehnes, H., Heinzmann, U., and Agerer, R.: An overview on bioaerosols viewed by
scanning    electron    microscopy,    Sci.    Total    Environ.,    346,    244-255,
https://doi.org/10.1016/j.scitotenv.2004.11.009, 2005.

Wu, L., Li, X., Kim, H., Geng, H., Godoi, R. H. M., Barbosa, C. G. G., Godoi, A. F. L., Yamamoto, C.
I., de Souza, R. A. F., Pöhlker, C., Andreae, M. O., and Ro, C. U.: Single-particle characterization of
aerosols collected at a remote site in the Amazonian rainforest and an urban site in Manaus, Brazil, Atmos.
Chem. Phys., 19, 1221-1240, 10.5194/acp-19-1221-2019, 2019.

Yue, S., Ren, H., Fan, S., Wei, L., Zhao, J., Bao, M., Hou, S., Zhan, J., Zhao, W., Ren, L., Kang, M., Li,
L., Zhang, Y., Sun, Y., Wang, Z., and Fu, P.: High Abundance of Fluorescent Biological Aerosol Particles
in Winter in Beijing, China, ACS Earth Space Chem., 1, 493-502, 10.1021/acsearthspacechem.7b00062,
2017.

Zawadowicz, M. A., Froyd, K. D., Murphy, D. M., and Cziczo, D. J.: Improved identification of primary biological aerosol particles using single-particle mass spectrometry, Atmos. Chem. Phys., 17, 7193-7212,

10.5194/acp-17-7193-2017, 2017.

Zhang, D., Murata, K., Hu, W., Yuan, H., Li, W., Matsusaki, H., and Kakikawa, M.: Concentration and

Viability of Bacterial Aerosols Associated with Weather in Asian Continental Outflow: Current

Understanding, Aerosol Sci. Engin., 1-12, 10.1007/s41810-017-0008-y, 2017.

**Supplemental Materials**

**Overview of primary biological aerosol particles from a Chinese boreal forest: insight into morphology, size, and mixing state at microscopic scale**

Weijun Li[1], Lei Liu[1], Qi Yuan[1], Liang Xu[1], Yanhong Zhu[1], Bingbing Wang[2], Hua Yu[3], Xiaokun Ding[4], Jian Zhang[1], Dao Huang[1], Dantong Liu[1], Wei Hu[5], Daizhou Zhang[6], Pingqing Fu[5], Maosheng Yao[7], Min Hu[7], Xiaoye Zhang[8], Zongbo Shi[9,5]

[1]Department of Atmospheric Sciences, School of Earth Sciences, Zhejiang University, Hangzhou 310027, China

[2]State Key Laboratory of Marine Environmental Science, College of Ocean and Earth Sciences, Xiamen University, Xiamen 361102, China.

[3]College of Life and Environmental Sciences, Hangzhou Normal University, 310036, Hangzhou, China

[4]Department of Chemistry, Zhejiang University, Hangzhou 310027, China

[5]Institute of Surface-Earth System Science, Tianjin University, 300072, Tianjin, China

[6]Faculty of Environmental and Symbiotic Sciences, Prefectural University of Kumamoto, Kumamoto 862-8502, Japan

[7]State Key Joint Laboratory of Environmental Simulation and Pollution Control, College of Environmental Sciences and Engineering, Peking University, Beijing 100871, China

[8]Key Laboratory of Atmospheric Chemistry, Chinese Academy of Meteorological Sciences, Beijing, China

9School of Geography, Earth and Environmental Sciences, University of Birmingham,

Birmingham B15 2TT, UK

*Correspondence to*: Weijun Li (liweijun@zju.edu.cn)

[Figure]

**Figure S1** The sampling procedures of substrate, sampler, storage, and analyzed technique.

[Figure]

**Figure S2** The *Yeast* and the *colibacillus* particles cultivated in laboratory. TEM image showing morphology and EDS showing compositions.

[Figure]

**Figure S3** Hygroscopic growth of NaCl generated in laboratory

[Figure]

**Figure S4** Meteorological data during the sampling including Wind speed and direction, Temperature, and relative humidity (RH).

[Figure]

**Figure S5** Size distribution of Yeast and Colibacillus cultivated in laboratory.

[Figure]

**Figure S6** SEM images of brochosomes.

[Figure]

**Figure S7** Particle number and relative abundance of rod-like PBAPs and fungal spores in the samples collected in daytime and nighttime.

---

## Author Comment (AC2) · 4 Nov 2019

**General Response: We thank the Referee#2 for your helpful comments. We have addressed all comments and provided point by point response below. The revised manuscript is presented in below Response**

(1) The manuscript "Morphology, mixing state, and hygroscopicity of primary biological aerosol particles from Chinese boreal forest" from Li et al. presents a physical and chemical characterisation of aerosol particles collected at a boreal forest site in China. The authors (i) derive an identification of large taxonomic classes (i.e., bacteria and fungi) from the particle's morphology and chemical composition, (ii) analyse the relative abundance of large particle classes as a function of day and night cycles, and (iii) analyse the hygroscopic growth of the collected particles. These results were obtained from transmission electron microscopy (TEM) and scanning electron microscopy (SEM) with energy-dispersive x-ray spectroscopy (EDS) analyses. Ultimately, the authors derive quantitative concentrations of certain bioaerosol classes and speculate on their potential roles in clouds and in precipitation formation.

Most of the paper is based on rather established concepts of bioaerosol cycling and techniques for bioaerosol analysis (i.e., SEM and TEM, hygroscopic growth studies, etc.). In my view, the really new aspects are the analysis of bioaerosol samples from this particular Chinese boreal forest site, which may allow interesting comparisons with other (boreal) forest sites worldwide as well as the quantification of bacterial and fungal spore concentrations. Thus, the aim and focus of the study is clearly a useful one.

However, I am very concerned about the overall quality of the manuscript – formally as well as scientifically. Formally, the paper is (i) not well structured, (ii) the introduction is just a loose collection of previous literature without really motivating the present work, (iii) the summary rather lists speculations than provides rigorous conclusions, and (iv) the language should be improved. Scientifically, crucial aspects of the analysis are poorly or even not at all explained. Moreover, I am sceptical if certain key results of the study are correct. My major points of criticism can be summarized as follows:

Response: We really appreciated the referee's comments. We carefully made the major revisions as the detailed comments as below. (1) We rewrote some parts indicated by red words and added one new experiment based on the referee's comments. (2) We rewrote the introduction part. (3) We rewrote conclusion part replacing the summary. (3) One native speaker was invited to publish the English writing. (4) We specifically explain the experimental procedure. (5) For the suspected part about bacteria, we added one new laboratory experiment to correct it. In the revised manuscript, we carefully draw the conclusions as two referee's and editor's comments.

(2)- The meaning and use of "bioaerosol identification" seems very problematic in this study. The authors state for example "*As a result, P derived from the particle EDS analysis coupled with the morphological features can be used to identify the PBAPs.*" First of all, it is not clear

what the authors exactly mean by "identification". In some case this seems to mean discrimination of biological and non-biological particles, whereas in other cases it seems to mean taxonomic determination.

Response: Thank you to point out this issue. Here P only can classify the biological and non-biological particles. We revised the statement here.

In abstract p28-30 "C, N, O, P, K, and Si were detected in most of the PBAPs, and P represented a major marker to discriminate the PBAPs and non-PBAPs."

(3) Moreover, a fundamental question of this work, which remains unanswered, is to what extent SEM/TEM analysis allows an identification of certain (taxonomic) groups within the total bioaerosol population and which uncertainty this involves. I don't doubt that several aerosol particles can be recognized as biological based on their morphology, surface texture and so on. Also, certain fungal spores (the characteristic ones) can be identified taxonomically based on their appearance as shows in previous studies. However, I am sceptical if any clear discrimination between bacteria and fungal spores (as stated in this study) can be obtained.

Response: Thank you to point out the problem. When we received your comments, we contacted several colleagues who study on the molecular biology, ecology, or disease. Indeed, there is one problem about bacteria classification. In textbook, the bacteria and some fungi might have very similar shape and composition. If there is no any molecular information, they should not be determined. As this reason, I also asked my colleagues to help cultivate one normal bacteria and fungi in laboratory and then we generated them into our TEM grids. Finally, we obtained their morphology and chemical compositions of *Colibacillus* and *Yeast*. Interestingly, we found that they have very similar shape and composition and different size range. The TEM observations from *Colibacillus* and *Yeast* fit our expectation. As the reason, we could have clear boundary to identify bacteria and fungi cell. In the revised manuscript, we named them as the rod-like PBAPs.

Figure 5 TEM image of the rod-like PBAPs collected in forest air and the fungi and bacteria cultivated in laboratory. (a) Morphology of a rod-like PBAP and EDS spectra of its core and main part. The red circles indicate where EDS obtained on rod-like PBAP. (b-e) Various rod-like PBAPs collected in forest air. (f) One *Yeast* particle cultivated in laboratory (e) One *colibacillus* particle cultivated in laboratory.

(4) In both classes, the morphological diversity is large. Many of the "bacteria" that the authors show (e.g., Fig. 2, 4, 5) are pretty large, which rather advocates for fungal spores. In fact, I have the impression that many fungal spores are 'sold' here as bacteria (i.e., see increase of bacteria fraction towards 10  $\mu$ m in Fig. 12).

Response: We really appreciated the referee's comments. Please see the above response. Indeed, we made mistake here. We re-analyzed the data and analyzed size distribution and aspect ratio of PBAPs. It is interesting that size distribution of the rod-like PBAPs collected in the forest air displays two typical peaks at 1.4  $\mu$ m and 3.5  $\mu$ m which likely represent bacteria and fungi.

---

## Author Comment (AC3) · 4 Nov 2019

1) Reading the comments from Reviewer #2 again, s/he raises some fundamentally important concerns in the overview of "major points of criticism" starting half-way down the first page. At that point s/he listed four major bullets to concerning the experimental assumptions of the study. S/he also listed four overall comments related to the organization and writing quality in the paragraph before. Lastly, s/he listed several pages worth of detailed comments, some of which are quite significant in themselves. I think all of these comments are relevant and worth carefully considering. I think it may indeed be possible to revise the manuscript sufficiently to raise it to a publishable quality, but that will depend on the nature of the revisions.

Response: We appreciated the editor to summary the comments. We carefully considered all the comments and revised manuscript. We almost rewrote the manuscript marked by red color.

Besides we considered the comments from the two referee's comments, one additional experiment which concerns the bacteria and fungi cultivated in the laboratory was added. During the revision period, we contacted several professors who are experts on bacteria molecular, ecology, and biology. Indeed, the classification on fine bacteria and fungi could be problem because some of fungi have almost same morphology with bacteria. Although bacteria normally have size at <2 μm than fungi with size at 1-10 μm, the particle size could not be used to classify bacteria and fungi. Therefore, we carefully revised the particle classification here to guarantee the right contents for the potential readers.

As the reviewer's comments and literatures, fungal spores have typical morphology and larger size (> 2μm). These particles easily have been identified from TEM and SEM images. Some previous studies well documented the fungal spores (Valsan et al., 2015; Wittmaack et al., 2005).

2) Most fundamentally, Reviewer #2 raised a number of concerns about the methods by which individual particles were classified. The lack of transparency on this issue indeed is one of the major areas of required improvement. After carefully adding specific details about how particles were assigned and categorized, it will be easier to evaluate the observations and conclusions. Without knowing more about the process by which particles were investigated and assigned, it is hard to know if the method itself was sufficient to support the conclusions. The question here is not just about clarifying wording, but that the clarified wording will help evaluate whether the method was sufficient or not. In particular, the question of whether the particle assignments were correct is not sufficiently addressed. Just because a systematic method is established is not a sufficient criterion to determine whether the method leads to correct assignment. For example, consider a skeptical scientist reader. Convince them that your method led to detection and consistent, correct categorization of the particle types you report.

Response: In the revised manuscript, we mainly revised the classification part. We carefully revised the method part and added more details. We tried to clearly make methods of particle classification. After we seriously considered their comments, we decided to prepare the standard samples of bacteria and fungi on the TEM grids. The laboratory samples can further confirm our mythology that TEM and SEM can well observe the bacteria or fungi. Moreover, the TEM and SEM observations of the standard bacteria and fungi are helpful to further classify the ambient PBAPs collected in the boreal forest air. We found that bacteria and fungi have almost same morphology and composition. Although bacteria normally have smaller size than fungi, particle size could not be acted as one key element to classify the bacteria and fungi. We also noticed that most of bacteria and fungi have rod-like shape in our samples. As the reason, we named the rod-like PBAPs for this type of particles. Furthermore, we classified fungal spores and brochosomes based on the unique morphology of individual PBAPs. Here we added Figure 2 and Figure 6 and revised Figure 5.

As the revised part, we made major revisions on the particle classification. We precisely delivered the information for the potential reader. We had the detailed responses of two referee's comments.

3) - The observations and atmospheric implications are relatively similar to works that have used similar techniques in both boreal and tropical areas, but these are not well cited in the manuscript. At a minimum I suggest you to consider additional literature, and make sure to at least briefly compare results with these in mind. I suggest doing a good literature survey of PBAP observations from forests, as well as a search related to ambient studies related to microscopy of PBAP (i.e. SEM and TEM, as you use). Then make sure that the observations you present and the conclusions you draw are put into context of these previous measurements.

Response: We appreciated the editor's suggestion. Indeed, we missed some important literatures. In the revised manuscript, we added 23 papers to support our results. Please see our revised manuscript with red word part.

References:

Afanou, K. A., Straumfors, A., Skogstad, A., Nilsen, T., Synnes, O., Skaar, I., Hjeljord, L., Tronsmo, A., Green, B. J., and Eduard, W.: Submicronic Fungal Bioaerosols: High-Resolution Microscopic Characterization and Quantification, Applied and Environmental Microbiology, 80, 7122-7130, 10.1128/aem.01740-14, 2014.

Afanou, K. A., Straumfors, A., Skogstad, A., Skaar, I., Hjeljord, L., Skare, Ø., Green, B. J., Tronsmo, A., and Eduard, W.: Profile and Morphology of Fungal Aerosols Characterized by Field Emission Scanning Electron Microscopy (FESEM), Aerosol Science and Technology, 49, 423-435, 10.1080/02786826.2015.1040486, 2015.

Ault, A. P., and Axson, J. L.: Atmospheric Aerosol Chemistry: Spectroscopic and Microscopic Advances, Analytical Chemistry, 89, 430-452, 10.1021/acs.analchem.6b04670, 2017.

Coz, E., Artíñano, B., Clark, L. M., Hernandez, M., Robinson, A. L., Casuccio, G. S., Lersch, T. L., and Pandis, S. N.: Characterization of fine primary biogenic organic aerosol in an urban area in the northeastern United States, Atmospheric Environment, 44, 3952-3962, https://doi.org/10.1016/j.atmosenv.2010.07.007, 2010.

Coz, E., Gómez-Moreno, F. J., Pujadas, M., Casuccio, G. S., Lersch, T. L., and Artíñano, B.: Individual particle characteristics of North African dust under different long-range transport scenarios, Atmospheric Environment, 43, 1850-1863, 10.1016/j.atmosenv.2008.12.045, 2009.

Crawford, I., Robinson, N. H., Flynn, M. J., Foot, V. E., Gallagher, M. W., Huffman, J. A., Stanley, W. R., and Kaye, P. H.: Characterisation of bioaerosol emissions from a Colorado pine forest: results from the BEACHON-RoMBAS experiment, Atmos. Chem. Phys., 14, 8559-8578, 10.5194/acp-14-8559-2014, 2014.

Crawford, I., Ruske, S., Topping, D. O., and Gallagher, M. W.: Evaluation of hierarchical agglomerative cluster analysis methods for discrimination of primary biological aerosol, Atmos. Meas. Tech., 8, 4979-4991, 10.5194/amt-8-4979-2015, 2015.

Gosselin, M. I., Rathnayake, C. M., Crawford, I., Pöhlker, C., Fröhlich-Nowoisky, J., Schmer, B., Després, V. R., Engling, G., Gallagher, M., Stone, E., Pöschl, U., and Huffman, J. A.: Fluorescent bioaerosol particle, molecular tracer, and fungal spore concentrations during dry and rainy periods in a semi-arid forest, Atmos. Chem. Phys., 16, 15165-15184, 10.5194/acp-16-15165-2016, 2016.

Harrison, R. M., Jones, A. M., Biggins, P. D. E., Pomeroy, N., Cox, C. S., Kidd, S. P., Hobman, J. L., Brown, N. L., and Beswick, A.: Climate factors influencing bacterial count in background air samples, International Journal of Biometeorology, 49, 167-178, 10.1007/s00484-004-0225-3, 2005.

Lacey, M. E., and West, J. S.: The Air Spora: A Manual for Catching and Identifying Airborne Biological Particles, Springer: Dordrecht, The Netherlands, 2006.

Manninen, H. E., Back, J., Sihto-Nissila, S. L., Huffman, J. A., Pessi, A. M., Hiltunen, V., Aalto, P. P., Hidalgo, P. J., Hari, P., Saarto, A., Kulmala, M., and Petaja, T.: Patterns in airborne pollen and other primary biological aerosol particles (PBAP), and their contribution to aerosol mass and number in a boreal forest, Boreal Environment Research, 19, 383-405, 2014.

Martin, S. T., Andreae, M. O., Artaxo, P., Baumgardner, D., Chen, Q., Goldstein, A. H., Guenther, A., Heald, C. L., Mayol-Bracero, O. L., McMurry, P. H., Pauliquevis, T., Pochl, U., Prather, K. A., Roberts, G. C., Saleska, S. R., Silva Dias, M. A., Spracklen, D. V., Swietlicki, E., and Trebs, I.: Sources and properties of Amazonian aerosol particles, Rev. Geophys., 48, RG2002, 10.1029/2008rg000280, 2010.

Morris, C. E., Conen, F., Alex Huffman, J., Phillips, V., Pöschl, U., and Sands, D. C.: Bioprecipitation: a feedback cycle linking Earth history, ecosystem dynamics and land use through biological ice nucleators in the atmosphere, Global Change Biol., 20, 341-351, 10.1111/gcb.12447, 2014.

Nikkels, A. H., Terstegge, P., and Spieksma, F. T. M.: Ten types of microscopically identifiable airborne fungal spores at Leiden, The Netherlands, Aerobiologia, 12, 107, 10.1007/bf02446602, 1996.

Priyamvada, H., Singh, R. K., Akila, M., Ravikrishna, R., Verma, R. S., and Gunthe, S. S.: Seasonal variation of the dominant allergenic fungal aerosols – One year study from southern Indian region, Scientific Reports, 7, 11171, 10.1038/s41598-017-11727-7, 2017.

Riemer, N., Ault, A. P., West, M., Craig, R. L., and Curtis, J. H.: Aerosol Mixing State: Measurements, Modeling, and Impacts, Rev. Geophys., 57, https://doi.org/10.1029/2018RG000615, 10.1029/2018rg000615, 2019.

Smith, D. J., Ravichandar, J. D., Jain, S., Griffin, D. W., Yu, H., Tan, Q., Thissen, J., Lusby, T., Nicoll, P., Shedler, S., Martinez, P., Osorio, A., Lechniak, J., Choi, S., Sabino, K., Iverson, K., Chan, L., Jaing, C., and McGrath, J.: Airborne Bacteria in Earth's Lower Stratosphere Resemble Taxa Detected in the Troposphere: Results From a New NASA Aircraft Bioaerosol Collector (ABC), Frontiers in Microbiology, 9, 10.3389/fmicb.2018.01752, 2018.

Tamer Vestlund, A., Al-Ashaab, R., Tyrrel, S. F., Longhurst, P. J., Pollard, S. J. T., and Drew, G. H.: Morphological classification of bioaerosols from composting using scanning electron microscopy, Waste Management, 34, 1101-1108, https://doi.org/10.1016/j.wasman.2014.01.021, 2014.

Tong, Y., and Lighthart, B.: The Annual Bacterial Particle Concentration and Size Distribution in the Ambient Atmosphere in a Rural Area of the Willamette Valley, Oregon, Aerosol Science and Technology, 32, 393-403, 10.1080/027868200303533, 2000.

Toprak, E., and Schnaiter, M.: Fluorescent biological aerosol particles measured with the Waveband Integrated Bioaerosol Sensor WIBS-4: laboratory tests combined with a one year field study, Atmos.

Chem. Phys., 13, 225-243, 10.5194/acp-13-225-2013, 2013.

Valsan, A. E., Ravikrishna, R., Biju, C. V., Pöhlker, C., Després, V. R., Huffman, J. A., Pöschl, U., and Gunthe, S. S.: Fluorescent biological aerosol particle measurements at a tropical high-altitude site in southern India during the southwest monsoon season, Atmos. Chem. Phys., 16, 9805-9830, 10.5194/acp-16-9805-2016, 2016.

Whitehead, J. D., Darbyshire, E., Brito, J., Barbosa, H. M. J., Crawford, I., Stern, R., Gallagher, M. W., Kaye, P. H., Allan, J. D., Coe, H., Artaxo, P., and McFiggans, G.: Biogenic cloud nuclei in the central Amazon during the transition from wet to dry season, Atmos. Chem. Phys., 16, 9727-9743, 10.5194/acp-16-9727-2016, 2016.

Wilson, T. W., Ladino, L. A., Alpert, P. A., Breckels, M. N., Brooks, I. M., Browse, J., Burrows, S. M., Carslaw, K. S., Huffman, J. A., Judd, C., Kilthau, W. P., Mason, R. H., McFiggans, G., Miller, L. A., Nájera, J. J., Polishchuk, E., Rae, S., Schiller, C. L., Si, M., Temprado, J. V., Whale, T. F., Wong, J. P. S., Wurl, O., Yakobi-Hancock, J. D., Abbatt, J. P. D., Aller, J. Y., Bertram, A. K., Knopf, D. A., and Murray, B. J.: A marine biogenic source of atmospheric ice-nucleating particles, Nature, 525, 234, 10.1038/nature14986, 2015.

4) - The statement in the manuscript that the work can be used as a "full database" to "be used to identify primary biological particles using single particle techniques" is overstated in my opinion. I think that that study can be revised to show an overview of observations of (bio)-particles in this region, but using the results as a database for future reference is entirely different. This would require a substantially higher threshold of demonstrated quality, which may or may not have been achieved. To claim these data as basis for a database of atmospheric particles implies that a sufficiently systematic representation of bioparticles has been sampled and correctly analyzed. Further, the database would need to show some sort of independent verification that particles were assigned correctly, similar to above comments. Since most methods of independent identification are well beyond the scope of the work you reported, I do not expect that you would want to argue that the assignments have necessarily been verified as correct. They are merely suggestions, with uncertainties and potential assignment errors to be at least briefly discussed in the revised manuscript. So in that case I would suggest that at a minimum you scale back the conclusions to remove the concept of 'database,' and report in the context of 'observations'. This does not get around the first concerns that Reviewer #2 raised about categorization of particle type, but it will help to re-frame the conclusions a bit. I strongly suggest keeping all these comments, including those from the two Reviewers, closely in mind as you revise and respond to all comments line-by-line. I look forward to reviewing the revised manuscript when available.

Response: We noticed that we made overstate. In the revised manuscript, we deleted such words and carefully gave the conclusions. About the classification part, we made one major revision. We also modified the title. We all carefully response the comments raised by the two referees.

**Overview of primary biological aerosol particles from a Chinese boreal forest: insight into morphology, size, and mixing state at microscopic scale**

Weijun Li[1], Lei Liu[1], Qi Yuan[1], Liang Xu[1], Yanhong Zhu[1], Bingbing Wang[2], Hua Yu[3], Xiaokun Ding[4], Jian Zhang[1], Dao Huang[1], Dantong Liu[1], Wei Hu[5], Daizhou Zhang[6], Pingqing Fu[5], Maosheng Yao[7], Min Hu[7], Xiaoye Zhang[8], Zongbo Shi[9,5]

[1]Department of Atmospheric Sciences, School of Earth Sciences, Zhejiang University, Hangzhou 310027, China

[2]State Key Laboratory of Marine Environmental Science, College of Ocean and Earth Sciences, Xiamen University, Xiamen 361102, China.

[3]College of Life and Environmental Sciences, Hangzhou Normal University, 310036, Hangzhou, China

[4]Department of Chemistry, Zhejiang University, Hangzhou 310027, China

[5]Institute of Surface-Earth System Science, Tianjin University, 300072, Tianjin, China

[6]Faculty of Environmental and Symbiotic Sciences, Prefectural University of Kumamoto, Kumamoto 862-8502, Japan

[7]State Key Joint Laboratory of Environmental Simulation and Pollution Control, College of Environmental Sciences and Engineering, Peking University, Beijing 100871, China

[8]Key Laboratory of Atmospheric Chemistry, Chinese Academy of Meteorological Sciences, Beijing, China

[9]School of Geography, Earth and Environmental Sciences, University of Birmingham, Birmingham B15 2TT, UK

*Correspondence to*: Weijun Li (liweijun@zju.edu.cn)

**Abstract:**

Biological aerosols play an important role in atmospheric chemistry, clouds, climate, and public health. Here, we studied the morphology and composition of primary biological aerosol particles (PBAPs) collected in the Lesser Khingan Mountain boreal forest of China in summertime using transmission electron microscopy (TEM) and scanning electron microscopy (SEM). C, N, O, P, K, and Si were detected in most of the PBAPs, and P represented a major marker to discriminate the PBAPs and non-PBAPs. Of all detected particles > 100 nm in diameter, 13% by number were identified as PBAPs. We found that one type of PBAPs mostly appeared as similar rod-like shapes with an aspect ratio > 1.5 and the dominant sizes ranged from 1 μm to 5 μm. The size distribution of the rod-like PBAPs displays two typical peaks at 1.4 μm and 3.5 μm, which likely are bacteria and fungal particles in the forest air. The second most PBAPs were identified as fungal spores with ovoid, sub-globular or elongated shapes with a smooth surface and small protuberances with their dominant size range of 2 - 5 μm. Moreover, we found some large brochosomal clusters containing hundreds of brochosomes with a size range of 200-700 nm and a shape like a truncated icosahedron. The number size distribution of PBAPs coupled with $PM_{2.5}$ and $PM_{10}$ concentrations were used to estimate the total mass concentration of PBAPs, which is approximately 1.9 μg m$^{-3}$ and accounts for 47% of the in situ $PM_{2.5-10}$ mass. Moreover, there is a higher frequency and concentration of PBAPs at night compared with day, suggesting that the relative humidity dramatically enhances the PBAPs emissions in the boreal forest. Our study also showed that the fresh PBAPs displayed weak hygroscopicity with a growth factor of ~1.09 at RH=94%. TEM revealed that about 20% of the rod-like PBAPs were internally mixed with metal, mineral dust, and inorganic salts in the boreal forest air. This work for the first time provides the overview of individual PBAPs from nanoscale to microscale in Chinese boreal forest air.

**Key points**

• Based on morphology, composition, and size of individual PBAPs, rod-like PBAPs (e.g., bacteria and fungi), fungal spores, and brochosomes were identified.

• PBAPs emissions tend to occur with high humidity at night rather than during the day.

• Hygroscopic experiments show that most of the PBAPs displayed weak hygroscopicity, and their growth factor was ~1.09 at RH=94%.

**1. Introduction**

Primary biological aerosol particles (PBAPs) (e.g., bacteria, spores, fungi, viruses, algae, and pollen) are ubiquitous in the Earth's atmosphere and important elements in the life cycle of many organisms and ecosystems (Poschl, 2005;Tunved et al., 2006;Smith et al., 2018). PBAPs are airborne biological materials that are transported from the biosphere to the atmosphere (Huffman et al., 2010), and they can account for a large proportion (25-45%) of the aerosol particle mass in pristine forest air and certain amounts in some rural and marine air (Elbert et al., 2007;Bauer et al., 2008;Hu et al., 2017;May et al., 2018). The growing research interest in PBAPs has one of its goals to better understand how PBAPs or their cell fragments influence cloud condensation nuclei (CCN) and ice nuclei (IN) (Morris et al., 2004;Huffman et al., 2013;Ling et al., 2018). Furthermore, field campaigns have found that abundant biological aerosols occur in cloud ice-crystals, fog/cloud, rain, and snowfall (Amato et al., 2005;Möhler et al., 2007;Christner et al., 2008;Pratt et al., 2009;Prenni et al., 2009;Tobo et al., 2013;Morris et al., 2014;Wilson et al., 2015;Twohy et al., 2016;Hu et al., 2018). These studies addressed the hypothesis that PBAPs indeed influence the hydrological cycle and climate by initiating the formation of clouds and precipitation as CCN and IN or by their bioprecipitation feedbacks.

Previous studies have investigated particle number concentration, size, and composition of primary biological aerosols using online measurement techniques and advanced molecular biological analyses (Wittmaack et al., 2005;Elbert et al., 2007;Fröhlich-Nowoisky et al., 2009;Huffman et al., 2010;Després et al., 2012;Crawford et al., 2015;Hu et al., 2017;Therkorn et al., 2017;Zhang et al., 2017;Chen and Yao, 2018). For example, the contribution of fungal spores to total organic carbon was estimated to be approximately 10% in clean and polluted periods in Beijing using an online wideband integrated bioaerosol sensor (WIBS) (Yue et al., 2017); To obtain the organisms of PBAPs in the atmosphere, many studies tend to detect biochemical markers (e.g., proteins, fatty acids, sugars) and nucleic acids (i.e., DNA and RNA) to determine their origins such as plant or animal debris, bacteria, fungi, or viruses (Georgakopoulos et al., 2009;Chen and Yao,

2018;Hu et al., 2018;Ling et al., 2018). Although these previous studies provided comprehensive species or detailed molecular compositions of PBAPs, they still could not reflect the physical properties of individual PBAPs in the atmosphere, such as morphology, size, phase, hygroscopicity, and mixing state. Besides particle composition, the previous studies have proved that the morphology, size, and mixing state of individual particles more or less influence their CCN and IN

activities and optical properties (Spracklen et al., 2008;Fröhlich-Nowoisky et al., 2009;Wilson et al.,

2015;Li et al., 2016;Ault and Axson, 2017;Riemer et al., 2019). Therefore, it is critical to characterize detailed information of different types of individual PBAPs from their natural sources.

In the past decades, several studies have used scanning electron microscopy (SEM) to characterize the morphology and size of individual PBAPs (Nikkels et al., 1996;Wittmaack et al.,

2005;Coz et al., 2010;Tamer Vestlund et al., 2014;Valsan et al., 2015;China et al., 2018). They identified fungal spores, brochosome, pollen, and plant or insect debris larger than 2 µm in the atmosphere. Although the SEM observations adequately characterized the coarse fungal spores, pollen, and plant or insect debris particles, comparable results have not been obtained for fine bacteria and fungal particles, which together account for a large number of suspended particles in ambient air detected by online instruments (Tong and Lighthart, 2000;Després et al., 2012;Afanou et al., 2014;Valsan et al., 2016;Priyamvada et al., 2017;Hu et al., 2018). The reason for this shortfall is likely that SEM could not clearly observe carbonaceous bioaerosols smaller than 1 µm (Li et al.,

2016;Ault and Axson, 2017). Posfai et al. (2003) and Patterson et al. (2016) used transmission electron microscopy (TEM) to detect some fine bacteria in marine air. However, there is no study to characterize the morphology, size, and mixing state of individual PBAPs from nanoscale to microscale. For example, many studies directly used SEM images showing the coarse PBAPs (e.g., fungal spores) in support of their conclusions, but missed large numbers of fine PBAPs (e.g., bacteria) (Shi et al., 2003;Wittmaack et al., 2005;Coz et al., 2009;Shi et al., 2009;Martin et al., 2010;Huffman et al., 2012;Tamer Vestlund et al., 2014;Afanou et al., 2015;Valsan et al., 2015;Valsan et al., 2016;Priyamvada et al., 2017;Wu et al., 2019). The result might discourage people considering fine bacteria and fungal particles for their atmospheric effects or for their examination of data from some online instruments. Therefore, it is necessary to integrate SEM and TEM to characterize the morphology, size, and mixing state of individual PBAPs from nanoscale to microscale.

Forests are important contributors of primary biological aerosols in the atmosphere (Tunved et al., 2006;Spracklen et al., 2008;Despr $\acute{\text{e}}$ s et al., 2012;Whitehead et al., 2016). Aerosols in large forests contain abundant biological particles from plants emitted locally and lesser amounts of anthropogenic pollutants from long-range transport (Tong and Lighthart, 2000;Tunved et al., 2006;Gabey et al., 2010;Martin et al., 2010). We chose the Lesser Khingan Mountains in northeast China, which is its second largest boreal forest. In this study, TEM and SEM both have been employed to characterize the morphology, size, and mixing state of various PBAPs collected over the boreal forest. Furthermore, hygroscopic experiments on the primary biological particles have been conducted.

**2.    Methods**

**2.1 Sampling site and sample collection**

The sampling site is at the Heilongjiang Liangshui National Nature Reserve (47.32 °N, 128.54 °

E; 350m above sea level) in the center of the Lesser Khingan Mountains of northeast China (Figure

1). The boreal region is characterized by large seasonal variations in temperature, and the flora is dominated by Korean pine and spruce species. There are no anthropogenic sources of pollutants, such as villages, industries and vehicles within 80 km of the sampling site. Boreal forests have the highest emissions of biological aerosols during summer. Because there is less rain in late Auguest, we selected 14-21 August, 2016 to collect the bioaerosol samples.

Individual particle samples were collected both on copper (Cu) TEM grids coated with carbon film (carbon type-B, 300-mesh copper; Tianld Co., China) and on silicon membranes (thickness:

500±10 μm, size: 3×3 mm; LIJINGKEJI, China) by a single-stage cascade impactor called the DKL-

2 sampler (Genstar Electronic Technology, China). The collection efficiency of the impactor is 50%

for particles with an aerodynamic diameter of 0.1 μm when we assume an aerosol particle density of 2 g cm$^{-3}$. We collected individual particles four times each day at 9:00, 15:00, 21:00, and 02:00

local time. At each sampling event, we first collected TEM grids and then changed to silicon wafers in the sampler. The sampling duration at each time varied from 10 min to 25 min depending on the particle distribution on the substrate. The substrates of the carbon film and silicon wafer both have smooth surfaces with no contamination before we use them to collect aerosol particles. After sample collection, we immediately performed optical microscopy (BST60-100, China) at 100X

magnification to determine whether the aerosol distribution on the substrate was suitable for electron microscopy analysis. The distribution of aerosol particles on TEM grids was not uniform, with coarser particles occurring near the center and finer particles on the periphery. The quick check by the optical microscopy enabled us to tell whether individual particles were well distributed and whether there was any overlap on the substrate. Whenever the distribution was not even enough or when substantial overlap occurred, we had to discard it and re-collect individual particle samples through adjusting the sampling duration. In a word, this sampling procedure guarantees that the collected particles were adequately separated and did not overlap each other on the substrate (Li et al., 2016). The Cu grids and silicon wafers were placed in a dry, clean, and airtight container with

25 ℃ and 20±3% RH which minimizes exposure to ambient air and preserves them for subsequent analysis. The detailed sampling and storage procedures are summarized in Figure S1.

The daily $PM_{2.5}$ and $PM_{10}$ samples were collected on quartz-fiber filters with a diameter of 90

mm through two medium-volume samplers (TH-150, Wuhan Tianhong, China) at a constant flow rate of 100 L $min^{-1}$. The samples were changed at 08:00 a.m. each day. The DKL-2 and TH-150

samplers and other monitoring instruments in the field experiment were installed on a building roof

15 m above ground. The quartz filters (Whatman, UK) were put in polyethylene boxes immediately after sampling and stored at −5 °C. They were equilibrated at a constant temperature (20 ±0.5 ℃)

and humidity (50 ± 2%) for over 24 h before being weighed with an electronic microbalance (Sartorius-ME5, Germany). This gravimetric procedure provides the mass concentration of $PM_{2.5}$

and $PM_{10}$.

[Figure]

**Figure 1** Location of the sampling site and 6-h air mass back trajectories arriving at each sampling time from 14-21 August, 2016 in a boreal forest of the Lesser Khingan Mountain in

Northeast China. The map source is Google Earth.

**2.2 Transmission electron microscopy analysis**

Individual aerosol particles collected on Cu grids were analyzed via transmission electron microscopy (TEM, JEM-2100, JEOL Ltd., Japan) at a 200 kV accelerating voltage. TEM with a beam of electrons is transmitted through a specimen to form an image. An image is formed from the interaction of the electrons with the sample as the beam is transmitted through the specimen.

Therefore, TEM images display the inner physical structure of individual particles and the mixing state of different components. The TEM system is equipped with an energy-dispersive X-ray spectrometer (EDS, INCA X-Max$^N$ 80T, Oxford Instruments, UK). EDS is an analytical technique used for the elemental analysis or chemical characterization of a sample. It relies on an interaction between X-rays  and a sample. EDS spectra show the peaks of different elements and the contribution of each element in the total. EDS semiquantitatively detects the elemental composition of individual particles with an atomic number greater than six (Z > 6). However, Cu peaks in the

EDS spectra were not considered because of interference from the copper substrate of TEM grids.

We determined the morphology, composition, and mixing state of individual particles through the combination of TEM and EDS. To reduce the damage to particles under the electron beam, the EDS

collection duration was limited to 15 s. Individual particles are distributed on TEM grids, with the coarser particles in the center of sampling spot and with the finer particles on the periphery.

Therefore, to guarantee that the analyzed particles are representative, five areas are selected from the sampling center to the periphery on each TEM grid. After a labor-intensive operation, we analyzed 150-250 individual particles with diameters of 100 nm-10 μm in each sample. Finally, we successfully analyzed 20 TEM grids in the study. TEM/EDS can determine the internal mixing structure of different aerosol components in fine particles and their specific composition. TEM clearly shows the morphology of particles smaller than 2 μm. For some larger particles, we might further carry the scanning electron microscopy (SEM) experiments to determine their morphology. In this study, we did observe one fungi (*Yeast)* and one bacteria (*colibacillus)* sample through TEM, which were prepared in biological laboratories (Figure S2). Microscopic observations from the bacteria and fungi samples prepared in the laboratory were helpful to classify PBAPs emitted from the forest.

Once we clearly obtained electron images of different particles, we could then measure particle size and shape factors. In this study, the area, perimeter, shape factor, and equivalent circle diameter (ECD) of individual particles in TEM images are manually or automatically obtained through an image analysis software (RADUS, EMSIS GmbH, Germany). Based on these measurements, we can classify particle types and determine the diameter and shape factor of individual particles among different particle types. Moreover, we statistically analyze the number fractions in different size bins.

Aspect Ratio is the maximum ratio between the length and width of a bounding box for the measured object. An aspect ratio of 1 (the lowest value) indicates that a particle is not elongated in any direction. The aspect ratio is defined as

$$AR = \frac{L_{max}}{W_{max}}$$

**2.3 Scanning electron microscopy analysis**

SEM is performed using a type of electron microscope that can determine the particle surface by scanning it with a high-energy beam of electrons in a raster scan pattern. An SEM system (Zeiss Ultra 55) equipped with a field emission gun operating at 5–20 kV was used to obtain detailed information on the surfaces of individual aerosol particles. Moreover, the SEMx was equipped with an energy-dispersive X-ray spectrometry (EDS), which can analyze the chemical composition of individual particles. The SEM/EDS can efficiently obtain the surface morphology, size, and composition of coarse particles without any coating process on the substrate. Finally, we selected six silicon wafers for SEM/EDS analysis (Figure S1). In this study, we used SEM/EDS to observe surface morphology of the coarse particles on silicon wafers and to confirm particle types which cannot be clearly shown in TEM images.

**2.4 Hygroscopic experiments**

A custom-made individual particle hygroscopic (IPH) system was used to observe the hygroscopic properties of individual biological particles at different relative humidity (RH) values (Figure 2). After the hygroscopic experiment, an SEM analysis of the sample was employed to primarily check particle types. This allowed us to further understand how PBAPs particles grow at different RH values ranging from 5% to 94%.

The scheme of the IPH system is shown in Figure 2, which consisted of four steps; (1) Introducing $N_2$ gas with a mass flow controller into a chamber; (2) Setting a TEM grid or silicon wafer on the bottom of an environmental microscopic cell (Gen-RH Mcell, UK), which can change the RH and maintain the temperature at 20 ℃; (3) Taking images at incremental RH values using an optical microscope (Olympus BX51M, Japan) with a camera (Canon 650D); (4) Obtaining through the RADUS software the PBAPs sizes (i.e., D(RH) and $D_0$) in the images taken from the optical microscopy manually or automatically.. The images can be taken at different RHs during hygroscopic experiments and then are input into the RADUS software for size measurement.

This IPH system has been tested and has successfully captured the hygroscopic growth of individual aerosol particles collected on either a silicon wafer or TEM grid in our laboratory (Sun et al., 2018). Before the IPH system is used for ambient samples, it must be checked through standard NaCl particles on a silicon wafer made in the laboratory. Figure S3 shows that the delinquence relitive humidity (DRH) of individual NaCl particles on this silicon wafer is at

76%, similar to the standard DRH at 75±1%. After the procedure, we can replace our collected samples into the IPH system.

The particle growth factor (GF), an important parameter used to describe the hygroscopic growth of individual particles, is defined as follows:

$$GF(RH)=\frac{D(RH)}{D_0}$$

where $D(RH)$ and $D_0$ are the diameters of particles at a given RH and at 5% RH, respectively.

[Figure]

**Figure 2** Scheme of a custom-made individual particle hygroscopic system to observe hygroscopic growth of individual particles

2.5 **Meteorological data and back trajectories**

Meteorological data, including the relative humidity (RH), temperature, wind speed, and wind direction, were measured and recorded every 5 min by an automated weather meter (Kestrel 5500, USA). During the sampling period, the relative humidity (RH) and temperature varied from 40-70% and 22-28 ℃ during the day and 90-100% and 10-15 ℃ during the night, respectively. The wind speed was 1.5-7.6 m s$^{-1}$ during the day and 0-1 m s$^{-1}$ at night (Figure

S4).

To determine the regional transport of air masses, 6-h back trajectories of air masses were generated using a Hybrid Single Particle Lagrangian Integrated Trajectory (HYSPLIT) model at the forest sampling station during 14-21 August, 2016. Based on the sampling times of each day at 09:00, 15:00, 21:00, and 02:00 (midnight) local time, we performed 31 air mass back trajectories. Here we selected an altitude of 500 m as the end point of each back trajectory (Figure 1). Figure 1 shows that all the back trajectories in the past 6-h had been transported over the Lesser Khingan Mountain forest.

**3.    Results and Discussion**

**3.1 Morphology and elemental composition of PBAPs**

Among the 4,122 analyzed aerosol particles with diameters of 100 nm-10 μm analyzed by

TEM/EDS, individual particles are classified into five groups based on their morphology and composition: S-OM (mixture of sulfate (S), organics (OM)), OM, mineral dust, and PBAPs (Figure

3). S can be used to indicate secondary sulfates; abundant C and minor O with transparent color constitute the coating of the sulfate core and represent secondary organic matter; and irregular particles containing Si, Al, Ca, minor Fe, and Ti normally indicate mineral dust particles.

Moreover, previous studies have found that elemental P in individual particles and their associated unique morphologies can be used to identify PBAPs by electron microscopy (Poschl,

2005;Wittmaack et al., 2005). Thirteen percent of particles were PBAPs, and low magnification

TEM and SEM images both revealed that abundant PBAPs occurred in the samples (e.g., Figure 3a- b).

The number fractions of size-resolved aerosol particles show that secondary S-OM and OM

particles were the dominant particle groups in the fine mode (< 1 μm) while PBAPs and mineral particles dominated the coarse mode (≥ 1 μm) (Figure 4a). Moreover, we noticed that the number fractions of PBAPs in each sample collected at night were much higher than those collected during the day. Abundant fine secondary sulfate and organic particles from photochemical formation were observed during the day. Figure 4b shows that the average number fraction of PBAPs was 2.5% in the samples collected during the day and as high as 30.0% at night. If we further calculated the number concentration of PBAPs in Figure 4b, the PBAPs concentration significantly increased by approximately seven times from daytime to nighttime, although the non-PBAPs concentration decreased.

Based on the morphology and size of the PBAPs, we definitely identified fungal spores and brochosomes, and plant or insect debris, all of which have been widely reported before (Wittmaack et al., 2005;Huffman et al., 2012;Afanou et al., 2014;Valsan et al., 2015;Priyamvada et al., 2017).

Besides these PBAPs, we also found many special rod-like PBAPs with a dominant size range of 1

- 5 μm. Pollen was not found in our samples, which may be because large pollen emissions occur in spring and early summer instead of late summer (August) in boreal forests (Manninen et al., 2014).

[Figure]

**Figure 3** Low magnification SEM and TEM images of individual particles collected from the forest air.

(a) low magnification SEM image of rod-like PBAPs (red arrows) and fungal spores (green); (b) low magnification TEM image of rod-like PBAPs particles and secondary sulfate (S-rich) particles; (c) SEM

image of a rod-like particle; (d) TEM image of a mineral dust particle (e) TEM image of an organic matter (OM) particle; and (f) TEM image of OM coating on S-rich particles. The color in (a) was artificially painted on the original SEM images.

[Figure]

**Figure 4** Number fractions of different types of particles in different size bins and their total number fraction (a); and number fractions of primary biological aerosol particles (PBAPs) and non-PBAPs during the day and night (b). The number of analyzed particles is listed above each column. D and N are daytime and nighttime.

Online instruments indicate that large number of fine PBAPs are bacteria and fungi in the forest air (Tong and Lighthart, 2000;Elbert et al., 2007;Huffman et al., 2010;Despr és et al., 2012;Hu et al.,

2018). Although many previous studies reported PBAPs through the SEM, no observations of fine bacteria and fungal particles in forest air were reported (Wittmaack et al., 2005;Shi et al.,

2009;Martin et al., 2010;Huffman et al., 2012;Tamer Vestlund et al., 2014;Valsan et al., 2015;Valsan et al., 2016;Priyamvada et al., 2017;Wu et al., 2019). Posfai et al. (2003) found a few rod-like bacterial particles in marine air using TEM. In this study, we found that the rod-like PBAPs (Figure

5a-e) have a morphology similar to bacteria reported by Posfai et al. (2003). These rod-like PBAPs were stable under the electron beam during the TEM analysis, and they contained C, N, O, P, and K

with minor Mg, Si, S, Ca and Fe (Figure 5a). These rod-like PBAPs have a size range of 300 nm-7

μm with the dominant size range of 1-5 μm with two typical peaks at 1.4 μm and 3.5 μm (Figure

6a). Figure 6b further shows that the aspect ratio of 85% of these particles is larger than 1.5.

In nature, many fine fungi normally displayed similar composition and rod-like shape. To better compare and confirm differences of bacteria and fungi observed in TEM, we cultured *Colibacillus*

and *Yeast* in the laboratory to represent bacteria and fungi. Then we sprayed the solution of

*Colibacillus* and *Yeast* onto TEM grids. After drying these samples, we observed the morphology and size of *Colibacillus* and *Yeast* through the TEM (Figure 5f-g and Figure S5). Indeed, TEM/EDS

show very similar rod-like shape and composition between *Colibacillus* and *Yeast* particles on the substrate, although the *Yeast* particles with a size range at 1-8 μm with a mean diameter at 4.3 μm are larger than *Colibacillus* (300 nm-2.5 μm with mean diameter of 1.3 μm) (Figure S5). It is interesting that the size distribution of the rod-like PBAPs collected in the forest air displays two typical peaks at 1.4 μm and 3.5 μm, which probably represent bacteria and fungi. Després et al.

(2012) stated that bacteria mostly have diameters of 1-2 μm and fungi of 1−10 μm in the atmosphere.

Although we can indicate the bacteria and fungi based on their sizes, the clue could not be used to precisely identify bacteria and fungi through electron microscopy due to their overlapped size range.

Figure 6b shows 85% of particles with larger aspect ratios (> 1.5), suggesting most of these PBAPs particles have typical rod-like shape. Although their identification is tentative, we called all these similar rod-like bacteria and fungal particles "rod-like PBAPs" here.

[Figure]

Figure 5 TEM image of the rod-like PBAPs collected in forest air and the fungi and bacteria cultivated in laboratory. (a) Morphology of a rod-like PBAP and EDS spectra of its core and main part. The red circles indicate where EDS impacted the rod-like PBAP. (b-e) Various rod-like PBAPs collected in forest air. (f) One *Yeast* particle cultivated in laboratory (e) One *colibacillus* particle cultivated in laboratory.

[Figure]

Figure 6 Size distribution and aspect ratios of rod-like PBAPs, fungal spores, and brochosomes collected in boreal forest air.

Fungal spores are microscopic biological particles that allow fungi to reproduce, serving a similar purpose to that of seeds in the plant world (Lacey and West, 2006). Spores can be released as a part of the sexual and/or asexual morph (stage) of the lifecycle of a fungus, and many species are able to produce spores from both stages (Despr és et al., 2012). Fungal spores have been reported in many places in the global air and their morphologies have been well documented (Shi et al.,

2003;Wittmaack et al., 2005;Coz et al., 2009;Shi et al., 2009;Martin et al., 2010;Huffman et al.,

2012;Tamer Vestlund et al., 2014;Afanou et al., 2015;Valsan et al., 2015;Valsan et al.,

2016;Priyamvada et al., 2017;Wu et al., 2019). In this study, the fungal spores generally appeared as ovoid (Figure 7a), sub-globular (Figure 7b-c) or elongated shapes with a smooth surface and small protuberances (apiculus) (Figures 8a-f). Figure 7d shows that their composition mainly consists of C, O and Si, followed by minor N, Mg, P, S, K and Fe. The size range of the observed fungal spores varied roughly between 400 nm and 7 μm (Figure 6a). The size distribution of fungal spores further showed a dominant size range of 2 - 5 μm and one peak at 4 μm. The number fraction of fungal spores at all aspect ratios is generally lower than 0.15, suggesting that there is no typical shape from either roundness or elongation for fungal spores in the boreal forest. SEM images clearly display that several typical fungal spores with diameters of 3.7-6.5 μm do not have well-defined shapes and that their surfaces have regular strips or regular protuberances (Figure 8). Similar fungal spores have been reported in forest air (Wittmaack et al., 2005;Valsan et al., 2015). Compared with the rod-like PBAPs, fungal spores normally have a rougher surface (Figures 6-7), larger size, and much higher Si and lower N. Therefore, the fungal spores can easily be identified based on their morphology among the PBAPs through the TEM and SEM analysis.

[Figure]

**Figure 7** TEM/EDS showing the morphology and composition of various fungal spores. (a) a spindle fungal spore; (b) a fungal spore with protuberances; (c) a fungal spore with protuberances; and (d) EDS spectrum showing the composition of fungal spore. The red circle indicates where EDS

impacted the particle.

[Figure]

size: 5.6 μm        size: 3.8 μm        size: 4.3 μm        size: 3.7 μm size: 6.5 μm        size: 4.0 μm

**Figure 8** SEM images showing the shape, size, and surface properties of fungal spores. Size represents the diameter of fungal spores. (a-d) Surfaces of three spindle fungal particles with a layer of strips. (e-f) Surfaces of two fungal spores with protuberances.

Brochosomes are hollow spherical particles produced by leafhoppers (Cicadelliae)

(Wittmaack, 2005). TEM and SEM observations both found abundant brochosomes in the samples.

The low-magnification SEM images showed that there are large brochosomal clusters on the substrate, each containing tens or hundreds of single brochosomes (Figure S6). Wittmaack (2005)

found that most of the brochosomes normally occur as large clusters and reported that each cluster contains up to 100,000 brochosomes. In this study, TEM and SEM both produce clear images showing the structure of the brochosome (Figures 9-10). Interestingly, the outline of each brochosome approximates a truncated icosahedron and the brochosome particles likely have unique inner structures, such as C60 Buckminster fullerenes (Figures 9a-b and 10). Compared with the rod- like PBAPs, chemical composition of the brochosomal particles show extremely high Si and low P

in addition to major C and O and minor N, Na, S, K and Fe (Figure 9c). A single brochosome has a size range of 200-700 nm with a mean diameter of 350 nm. The aspect ratio of individual brochosomes is close to 1, suggesting that they are spherical (Figure 6b). Because the brochosomes might be dispersed from their clusters when they impact on the substrate,, it is not meaningful to compare the number fraction of brochosomes with the rod-like PBAPs and fungal spores.

[Figure]

**Figure 9** TEM images of brochosomes and the composition of (a) a single brochosome and brochosome aggregations; (b) high-resolution TEM image showing the inner structure of one brochosome; (c) EDS

spectrum showing the chemical composition of the brochosomes.

[Figure]

**Figure 10** SEM images of brochosomes. (a) Single brochosome and their aggregations. Some brochosomal particles are associated with primary biological species. (b) High-resolution SEM image showing the surface properties of the brochosomal particles.

The TEM and SEM images both show a few elongated large particles at 8-20 μm among the biological particles. EDS shows that these particles mainly contained C, O, and Si but no detectable

P in some of these biological particles as shown in Figures 11-12. We speculate that these biological particles were plant or insect debris. For example, Wittmaack et al. (2005) suggested that the spaghetti-type biological particles from Figure 11a-d are likely epicuticlar wax fragments of plants.

The SEM images as shown in Figure 12 clearly displayed the surface morphology of the large particles.

[Figure]

**Figure 11** TEM images showing the morphology of the primary biological particles. (a) One elongated particle with thorns; (b) one circular particle; (c-d) two elongated particles; and (e) one long spindle particle

[Figure]

size: 11.4 µm    size: 8.3 µm    size: 10.1 µm

**Figure 12** SEM image showing the morphology and surface properties of three elongated biological particles.

**3.2  Relative abundance of PBAPs**

In this study, we classified PBAPs but also efficiently obtained the number fraction of rod-like

PBAPs and fungal spores in coarse mode particles (> 1 μm). The results from the electron microscopy analysis further estimated that PBAPs, mineral dust, and the remaining particles accounted for 50%, 25%, and 25% of the coarse mode, respectively. Assuming a density of ~1 g cm$^{-3}$ for PBAPs (Elbert et al., 2007), 2 g cm$^{-3}$ for mineral dust particles, and 1.4 g cm$^{-3}$ for the remaining particles (e.g., S-OM, OM, and metal) (Rissler et al., 2006), mass concentrations of the three different types of particles with different size bins can be estimated based on the equation:

$$M_i = \frac{\Pi}{6} D_i{}^3 \rho_i N_i$$

*i*: particle type (PBAPs, mineral dust, and other remaining particle)

D: particle geometrical diameter in a size bin

N: particle number in a size bin

M: total mass of the analyzed particles in a size bin

ρ: particle density (g cm$^{-3}$)

In the equation, $N_i$ and $D_i$ both can be obtained through the measurement of individual particles in TEM images. Finally, we estimated that the mass concentration of PBAPs, mineral dust, and remaining particles accounted for 47%, 43%, and 10% of PM$_{2.5-10}$, respectively. The results suggest that PBAPs significantly contributed to mass concentration of PM$_{2.5-10}$ in summertime in the boreal forest air. During the sampling period, we measured the daily mass concentrations of PM$_{2.5}$ of ~6.0

μg m$^{-3}$ and PM$_{10}$ of ~10.0 μg m$^{-3}$. The number size distribution of PBAPs coupled with the mass concentrations of PM$_{2.5}$ and PM$_{10}$ were used to estimate the total mass concentration of PBAPs using the result from the above equation. We estimated that the PBAPs contributed ~1.9 μg m$^{-3}$ to the concentration of PM$_{2.5-10}$ of 4.0 μg m$^{-3}$.

Thirteen percent of all detected particles by number collected from the boreal forest air are

PBAPs. Such a high fraction of PBAPs has not been reported in urban and rural air in China (Shi et al., 2003;Shi et al., 2009;Li et al., 2016). We noticed that the number concentration of PBAPs was much higher at night than during the day (Figure 3b). A shallow nocturnal boundary layer can lead to a slight increase in the number concentration of coarse particles near the ground (Graham et al.,

2003), but this increase cannot explain the large difference in the relative number fraction of PBAPs (12 times larger at night than during the day) (Figure 3b). Alternately, the relative emission strength of PBAPs from the forest between day and night likely induced the difference of the relative number fractions.

It is well documented that meteorological conditions such as RH, wind speed, and temperature can affect PBAPs emission in the forests (Harrison et al., 2005;Whitehead et al., 2016). In particular, the wind speed is especially important in promoting PBAPs emission into air. During the sampling period, the average wind speeds at 5 min intervals had a range from 0 to 7.5 m/s with a mean value of 0.75 m/s. 89% of the measured wind speeds were lower than 2 m/s (Figure S4). Therefore, we conclude that no large consistent wind speeds occurred during the sampling period. Furthermore, we compared all the air mass back trajectories in the past 6-h over the Lesser Khingan Mountain forest at each sampling time (Figure 1). There are similar lengths of these back trajectories, suggesting that wind speeds above the forest canopy had only small changes during the sampling period. Therefore, the result from the ground-based measurements of wind speeds is consistent with air mass back trajectories. Here, we can exclude wind speeds during the sampling period as one important factor to dominate PBAPs emissions during day and night in the boreal forest. High temperatures normally increase the PBAPs emissions from the plants in the daytime (Harrison et al., 2005). However, we observed contrasting results that more PBAPs occurred in nighttime instead of daytime (Figure S4). Therefore, we also exclude temperatures during the sampling period as a cause of the vastly different PBAPs emissions at day and night in the boreal forest.

Besides wind speed and temperature, RH is an important meteorological variable that influences PBAPs emissions from plants (Harrison et al., 2005;Huffman et al., 2012). In this study, we found large differences of RH between day and night (Figure S4). The elevated RH near 100%

at night (Figure S1) appears to be an important factor that increases the emissions of PBAPs. This result is consistent with the conclusion of Elbert et al. (2007), who showed that PBAPs in a boreal forest are generally most abundant in samples collected at night when the RH is close to 100%. A

similar phenomenon has been   observed in different forests, such as the Amazon rainforest (Huffman et al., 2012;Whitehead et al., 2016), a montane ponderosa pine forest in North American (Crawford et al., 2014), a semi-arid forest in the southern Rocky Mountains of Colorado (Gosselin et al., 2016), and a semi-rural site in southwestern Germany (Toprak and Schnaiter, 2013). These studies above found that a nighttime peak of number concentrations of fluorescent biological aerosol particles is consistent with nocturnal sporulation driven by the increased RH. Moreover, Troutt and

Levetin (2001) explained that the increase in PBAP concentrations is caused by the increase in basidiospores concentrations with RH, and they showed that a clear diurnal rhythm occurs and peaks at 04:00-06:00 LT. Furthermore, the number ratio (4.6 at nighttime and 4.0 at daytime) of rod-like

PBAPs vs fungal spores and their number concentrations increased from daytime to nighttime (Figure S7). These results all suggest that higher RH can promote the emissions of rod-like PBAPs and fungal spores in the boreal forest.

**3.3  Mixing state of rod-like PBAPs**

Our study shows that rod-like PBAPs contain bacteria and fungi in the boreal forest air.

Although approximately 80% of rod-like PBAPs were externally mixed particles in the boreal forest air, we still found that 20% of rod-like PBAPs were internally mixed particles. TEM observations show that the rod-like PBAPs were frequently internally mixed with mineral, metal, organics, and inorganic salts. We noticed that irregular mineral dust particles significantly changed the shape of the rod-like PBAPs (Figure 13a-c). The EDS analysis shows that the internally mixed mineral particles contain certain amounts of C, O, and P in addition to Si, Al, or Ca (Figure 13a-c), suggesting that many rod-like PBAPs were associated with mineral dust particles.

In this study, we found that some nanoscale metal particles were internally mixed with rod-like

PBAPs. Figure 13d-f further shows that these metals were spherical and contained Mn, Si and/or

Fe. As in previous studies, these nanosize metal particles were emitted from industrial activities or power plants instead of natural soil (Li et al., 2017). TEM observations show that these metallic particles were mainly attached to the surface of rod-like PBAPs. Moreover, some rod-like PBAPs were coated by inorganic salts (e.g., K-P in Figure 13g and S-rich in Figure 13i) and organics. The shape of the rod-like PBAPs might change following the aging process during long-range transport (Figure 13), although the elemental P or its associated ionic components ($H_2PO_4^-$ and $PO_3^-$) did not change (Pratt et al., 2009). Pratt et al. (2009) detected $H_2PO_4^-$ and $PO_3^-$ in individual cloud ice- crystal residues to identify PBAPs using aerosol time-of-flight mass spectrometry. Although one study indicates that a few mineral dust or fly ash particles contain trace inorganic P, these particles do not contain abundant organics and their number is low in the air (Zawadowicz et al., 2017).

Therefore, TEM/EDS is an efficient tool to identify fine bacteria or fungi from non-PBAPs collected in the atmosphere. Moreover, it significantly reveals the mixing state of individual PBAPs, a key to understand their possible CCN and IN activity over the boreal forest air in the future.

[Figure]

**Figure 13** TEM showing the internally mixed rod-like PBAPs. (a-c) Internal mixture of mineral and rod- like PBAP; (d-f) Internal mixture of metal and rod-like PBAP; (g) Internal mixture of inorganic salts and rod-like PBAP; (h) Internal mixture of organics and rod-like PBAP; and (i) Internal mixture of S-rich salts and rod-like PBAP.

**3.4 Hygroscopicity of PBAPs**

In this study, we conducted an experiment to observe the hygroscopic growth of fresh PBAPs.

In the hygroscopic experiment, the PBAPs all take up water and grow by up to 88% during hydration, and they lose water and return to the dry particle size (reduction of 83%) during dehydration (Figure

14). The growth factor of the PBAPs is ~1.09 at RH=94% based on the particle diameter change, which is much lower than growth factor of NaCl at ~2.3 (Figure 14). These results show that the fresh PBAPs have extremely weak hygroscopicity.

Recent studies found that fungal fragments collected in Amazon forests displayed strong hygroscopic properties (China et al., 2016;China et al., 2018) and were internally mixed with certain amounts of sodium salts. However, we found weak hygroscopic growth of 1.09, whereas this value was in the range of 1.05-1.3 for bacteria and fungal spores in previous studies (Reponen et al.,

1996;Lee et al., 2002). However, the result is much lower than the value of 2.30 at RH=94% for

NaCl (Figure 2a) and 1.60 at RH 94% for ammonium sulfate (Sun et al., 2018). This comparison suggests that fresh PBAPs display extremely weak hygroscopicity and do not contain any sodium salt in the boreal forest (Figure 2a). Overall, our results indicate that PBAPs from the substantial biological emissions from the Khingan Mountain boreal forest are weakly hygroscopic in nature.

[Figure]

**Figure 14** Hygroscopic growth of NaCl prepared in laboratory and primary biological particles collected in boreal forest air. The up arrows (i.e., RH) represent hydration.

**4.  Conclusions**

The TEM and SEM observations both showed that the morphology of PBAPs were unique; they differed markedly from that of the sulfate, mineral, soot, organics, and metal particles in continental air. Our results indicate that significant amounts of PBAPs are emitted from the Khingan

Mountain area. In this study, we establish detailed information that includes the morphology, size, and composition of rod-like PBAPs, fungal spores, and brochosomes. C, N, O, P, K, and Si were detected in most of the PBAPs, and P represented a major marker to discriminate the PBAPs and non-PBAPs. We found that one type of PBAPs mostly appeared as similar rod-like shapes with an aspect ratio > 1.5 and the dominant sizes ranged from 1 μm to 5 μm. The size distribution of the rod-like PBAPs displays two typical peaks at 1.4 μm and 3.5 μm, which likely represent bacteria and fungal particles in the forest air. However, our study shows that there was no clear boundary between bacteria and some fungi from their size because of their size range partly overlapped.

The second most plentiful PBAPs were identified as fungal spores with ovoid, sub-globular or elongated shapes with a smooth surface and small protuberances (apiculus) with size at 400 nm - 7

μm with a mean diameter of 4 μm. Moreover, we found some large brochosomal clusters containing hundreds of brochosomes which have sizes from 200-700 nm and shapes like truncated icosahedrons. We estimated that the mass concentration of PBAPs, mineral dust, and remaining particles accounted for 47%, 43%, and 10% of the $PM_{2.5-10}$ mass concentration, respectively, indicating that large boreal forests might represent a major source of PBAPs in the atmosphere.

Moreover, there is a higher frequency and concentration of PBAPs at night compared with day. This difference could not be explained by wind speed or temperature, but was explicable by RH, which appears to be critical in enhancing PBAPs emissions from plants at night. The hygroscopic experiment shows that the primary bacterial and fungal particles show weak hygroscopicity.

**Author Contributions:** WL designed the study. WL, LL, QY, LX, and HY collected aerosol particles. WL, QL, LL, LX, YZ, BW, XD, and JZ contributed laboratory experiments and data analysis. WL prepared the manuscript with contributions from all the coauthors. BW, DH, DL, WH, DZ, PF, MY, MH, XZ, and ZS commented and edited the paper.

**Competing interests:** The authors declare no competing financial interests

**Acknowledgments**

We appreciate Peter Hyde's comments and proofreading. We appreciate Min Yue and Jiachang Lian (Zhejiang Univ.) providing the *Yeast* and *Colibacillus* cultures. Huijun Xie (Shandong Univ.) is thanked for discussions. This work was funded by the National Natural Science Foundation of China (9184430003, 41622504 and 41575116), Zhejiang Provincial Natural Science Foundation of China (LZ19D050001), and Zhejiang University Education Foundation Global Partnership Fund. All the data are presented in the paper.

**Supplemental Materials**

**Overview of primary biological aerosol particles from a Chinese boreal forest: insight into morphology, size, and mixing state at microscopic scale**

Weijun Li[1], Lei Liu[1], Qi Yuan[1], Liang Xu[1], Yanhong Zhu[1], Bingbing Wang[2], Hua Yu[3], Xiaokun Ding[4], Jian Zhang[1], Dao Huang[1], Dantong Liu[1], Wei Hu[5], Daizhou Zhang[6], Pingqing Fu[5], Maosheng Yao[7], Min Hu[7], Xiaoye Zhang[8], Zongbo Shi[9,5]

[1]Department of Atmospheric Sciences, School of Earth Sciences, Zhejiang University, Hangzhou 310027, China

[2]State Key Laboratory of Marine Environmental Science, College of Ocean and Earth Sciences, Xiamen University, Xiamen 361102, China.

[3]College of Life and Environmental Sciences, Hangzhou Normal University, 310036, Hangzhou, China

[4]Department of Chemistry, Zhejiang University, Hangzhou 310027, China

[5]Institute of Surface-Earth System Science, Tianjin University, 300072, Tianjin, China

[6]Faculty of Environmental and Symbiotic Sciences, Prefectural University of Kumamoto, Kumamoto 862-8502, Japan

[7]State Key Joint Laboratory of Environmental Simulation and Pollution Control, College of Environmental Sciences and Engineering, Peking University, Beijing 100871, China

[8]Key Laboratory of Atmospheric Chemistry, Chinese Academy of Meteorological Sciences, Beijing, China

[9]School of Geography, Earth and Environmental Sciences, University of Birmingham, Birmingham B15 2TT, UK

*Correspondence to*: Weijun Li (liweijun@zju.edu.cn)

[Figure]

**Figure S1** The sampling procedures of substrate, sampler, storage, and analyzed technique.

[Figure]

**Figure S2** The *Yeast* and the *colibacillus* particles cultivated in laboratory. TEM image showing morphology and EDS showing compositions.

[Figure]

**Figure S3** Hygroscopic growth of NaCl generated in laboratory

[Figure]

**Figure S4** Meteorological data during the sampling including Wind speed and direction, Temperature, and relative humidity (RH).

[Figure]

**Figure S5** Size distribution of Yeast and Colibacillus cultivated in laboratory.

[Figure]

**Figure S6** SEM images of brochosomes.

[Figure]

**Figure S7** Particle number and relative abundance of rod-like PBAPs and fungal spores in the samples collected in daytime and nighttime.